# Chlorine oxidation of VOCs at a semi-rural site in Beijing: Significant chlorine liberation from ClNO₂ and subsequent gas and particle phase Cl-VOC production

Michael Le Breton[1], Åsa M Hallquist[2], Ravi Kant Pathak[1], David Simpson[3,4], Yujue Wang[5], John Johansson[3], Jing Zheng[5], Yudong Yang[5], Dongjie Shang[5], Haichao Wang[5], Qianyun Liu[6], Chak Chan[7], Tao Wang[8], Thomas J. Bannan[9], Michael Priestley[9], Carl J. Percival[9*], Dudley E. Shallcross[10,11], Keding Lu[5], Song Guo[5], Min Hu[5] and Mattias Hallquist[1]

[1]Department of Chemistry and Molecular Biology, University of Gothenburg, Gothenburg, Sweden
[2]IVL Swedish Environmental Research Institute, Gothenburg, Sweden
[3]Earth and Space Sciences, Chalmers University of Technology, Gothenburg, Sweden
[4] Norwegian Meteorological Institute, Oslo, Norway
[5]State Key Joint Laboratory of Environmental Simulation and Pollution Control, College of Environmental Sciences and Engineering, Peking University, Beijing, China
[6]Division of Environment and Sustainability, The Hong Kong University of Science and Technology, Clearwater Bay, Kowloon, Hong Kong
[7]School of Energy and Environment, City University of Hong Kong, Hong Kong
[8]Department of Civil and Environmental Engineering, The Hong Kong Polytechnic University, Hong Kong, China
[9]Centre for Atmospheric Science, School of Earth, Atmospheric and Environmental Science, University of Manchester, Manchester, UK
[10]School of Chemistry, University of Bristol, Cantock's Close, Bristol, BS8 1TS, UK
[11]Department of Chemistry, University of the Western Cape, Bellville, Cape Town, South Africa.
* Now at Jet Propulsion ~~laboratory~~Laboratory, California Institute of Technology, 4800 Oak Grove Drive, Pasadena, California, USA.

Correspondence to: M. le Breton (Michael.le.breton@gu.se)

**Abstract.** Nitryl Chloride (ClNO₂) accumulation at night-time acts as a significant reservoir for active chlorine and impacts the following day´s photochemistry when the chlorine atom is liberated at sunrise. Here, we report simultaneous measurements of N₂O₅ and a suite of inorganic halogens including ClNO₂ and Cl-VOCs in the gas and particle phase utilizing the FIGAERO-ToF-CIMS during an intensive measurement campaign 40 km Northwest of Beijing in May and June 2016. A maximum mixing ratio of 2900 ~~pptV~~ppt of ClNO₂ was observed with a mean campaign night-time mixing ratio of 487 ppt, appearing to have an anthropogenic source supported by correlation with SO₂, CO and benzene, which often persisted at high levels after sunrise until midday. This was attributed to such high mixing ratios persisting after numerous e-folding times of the photolytic lifetime enabling the chlorine atom production to reach $2.3 \times 10^5$ molecules cm⁻³ from ClNO₂ alone, peaking at 9:30 am and up to $8.4 \times 10^5$ molecules cm⁻³ when including the supporting inorganic halogen measurements.

Cl-VOCs were ~~measured~~observed in the particle and gas phase for the first time at high time resolution and illustrate how the iodide ToF-CIMS can detect unique markers of chlorine atom chemistry in ambient air from both biogenic and anthropogenic sources. Their presence and abundance can be explained via time series of their measured and steady state calculated precursors, enabling the assessment of competing OH and chlorine atom oxidation via measurements of products from both of these mechanisms and their relative contribution to SOA formation.

## 1. Introduction

NO and $NO_2$ ($NO_x$) are important catalysts in the photochemical production of ozone ($O_3$) playing a significant
role in the oxidation of volatile organic compounds (VOCs) and subsequently have an adverse effect on air quality.
In the daytime $NO_x$ is primarily removed by the hydroxyl radical (OH) to form nitric acid ($HNO_3$), which is
subsequently lost by wet deposition, becoming a major component of acid rain. At night-time, the OH radical is
not a significant oxidant as photolysis stops, enabling the reaction between $NO_2$ and $O_3$ to form significant levels
of the nitrate radical ($NO_3$) (Atkinson, 2000). $NO_3$ can accumulate at night or further react with $NO_2$ leading to the
formation of $N_2O_5$ (Brown *et al.,* 2003b, Brown and Stutz, 2012). This equilibrium can lead to the reaction of $NO_3$
with VOCs at night forming organic nitrates or act as an important intermediate for heterogeneous reaction on
aerosols as $N_2O_5$ produces $NO_3^-$ and $NO_2^+$ in the aqueous phase (Hallquist *et al* 1999, Hallquist *et al*, 2000, Wagner
*et al.,* 2013). In the presence of chlorine, which is assumed in models to predominantly come from sea salt (Baker
*et al.*, 2016), nitryl chloride ($ClNO_2$) can be formed and released into the gas phase from the aerosol surface
(Osthoff *et al.*, 2008). $ClNO_2$ formation thereafter acts as a night-time radical reservoir due to its stability at night.
At sunrise $ClNO_2$ is rapidly photolysed, liberating the highly reactive chlorine atom subsequently converting it
into ~~Cl,~~ ClO, HOCl and $ClONO_2$ depending on the available sunlight, $O_3$, $HO_x$ and $NO_x$ levels via the following
reaction pathways (R1-R11).
**R1.** $ClNO_2 + h\nu \rightarrow Cl + NO_2$
**R2.** $Cl + O_3 \rightarrow ClO + O_2$
**R3.** $ClO + NO_2 \rightarrow ClONO_2$
**R4.** $ClO + HO_2 \rightarrow HOCl + O_2$
**R5**. $ClONO_2 + h\nu \rightarrow Cl + NO_3$
**R6.** $ClONO_2 + h\nu \rightarrow ClO + NO_2$
**R7.** $HOCl + h\nu \rightarrow Cl + OH$
**R8.** $ClONO_2 + H^+ + Cl^- \rightarrow Cl_2 + HNO_3$
**R9.** $HOCl + H^+ + Cl^- \rightarrow Cl_2 + H_2O$
**R10.** $ClO + NO \rightarrow Cl + NO_2$
**R11.** $OH + HCl \rightarrow Cl + H_2O$
The liberated chlorine will predominantly react with VOCs, with the ~~above~~ pathways listed (R2-R11) representing
alternative routes to loss of the chloride ~~atom~~radical, and contribute to daytime photochemical oxidation,

competing with OH and perturbing standard organic peroxy radical abundance (ROx = OH + HO$_2$ + RO$_2$), O$_3$ production rate, NOx lifetime and partitioning between reactive forms of nitrogen (Riedel et al., 2014). The ~~chlorine atom possesses~~ rate constants for the reaction of chlorine atoms with a number of VOCs is round 200 times larger than the equivalent reaction with OH (Tanaka *et al*., 2003); therefore, its abundance, fate and cycling can significantly alter standard daytime oxidation pathways. The oxidation of VOCs by chlorine atoms is thought to be significant in the early hours of the day while OH mixing ~~ratio~~ratios are low and chlorine atom production is high through the photolysis of ClNO$_2$, as well as feeding into the standard HO$_x$/NO$_x$ cycles via production of peroxy radicals from reactions with alkanes. Additional Cl$_2$ photolysis and HCl reaction with OH can also produce chlorine atoms throughout the day but at lower rates.

~~The oxidation mechanism of saturated hydrocarbon (R12-R13) is initiated~~Saturated hydrocarbons are usually oxidised by reaction with OH or chlorine atom to form an organic peroxy radical (RO$_2$), and H$_2$O or HCl depending on the oxidant, (R12 and R13), which is the dominant pathway for chloride-VOC reactions. In a heavily polluted environment such as Beijing, the RO$_2$ favours further reactions with NO to form an oxygenated volatile organic compound, HO$_2$ and NO$_2$ or an alkyl nitrate RONO$_2$. Specifically, acyl peroxy radicals can also react with NO$_2$ to form acyl peroxy nitrates (APN) such as peroxy acetyl nitrate (PAN).

$$\textbf{R12}.\ RH + OH \xrightarrow{O_2} RO_2 + H_2O$$

$$\textbf{R13.}\ RH + Cl \xrightarrow{O_2} RO_2 + HCl$$

$$\textbf{R14a.}\ RO_2 + NO \rightarrow OVOC + HO_2 + NO_2$$

$$\textbf{R14b.}\ RO_2 + NO \rightarrow RONO_2$$

$$\textbf{R15.}\ RO_2 + NO_2 \leftrightarrow APN$$

Addition of the chlorine atom to unsaturated VOC can also occur and then continue on the similar reaction pathway as denoted by R12 – R15. These pathways result in the production of unique chlorine atom chemistry markers which have been previously investigated to indicate the extent of chlorine atom oxidation reactions (Riemer *et al*., 2008, Keil and Shepson, 2006). The utilization of these compounds, such as 2-chloroperoxypropionyl nitrate (2-Cl PPN) and 1-chloro-3-methyl-3butene-2-one (CMBO) as chlorine atom chemistry markers relies on the abundance of the chlorine atom, the VOC precursor; HO$_x$, NO$_x$ and O$_3$ and competing pathways for chlorine atom reactions. Riedel *et al.* (2014) calculated that up to ~~tens of~~ten parts per trillion (ppt) Cl-VOCs are formed as a result of chlorine atom addition to alkenes and can therefore provide a number of potential periods of dominating active Cl chemistry (Wang *et al*., 2001).

The production of chloroperoxy radicals via chlorine atom addition can lead to the formation of semi volatile oxidation products which have been observed for both biogenic (Cai and Griffin *et al*., 2006) and anthropogenic emissions (Huang *et al*., 2014, Riva *et al.,* 2015) in controlled laboratory studies. Chlorine initiated oxidation of isoprene could also represent a significant oxidation pathway due to its rapid reaction rate compared ~~to~~with OH (Orlando *et al*., 2003) resulting in gas phase products such as chloroacetaldehyde and CMBO, a unique tracer for atmospheric chlorine atom chemistry (Nordmeyer *et al*., 1997). Furthermore, reactions of the chlorine atom with

isoprene or its SOA derived products could serve as an atmospheric chlorine sink (Ofner *et al*., 2012). Wang *et al*. (2017) revealed chlorine initiated oxidation of isoprene can produce SOA yields up to 36%, with products similar to that of OH isoprene oxidation, compared ~~to~~with the 15% yield from standard oxidation calculated by Liu *et al*. (2016), although this is known to be a factor of 2 higher than utilised in standard climate models. This SOA formation from chlorine initiated oxidation presents a large knowledge gap in the literature, which to date is limited by measurement capabilities.

This complex system results in a large uncertainty in the global budget of chlorine atoms $\sim 15-40$ Tg Cl yr$^{-1}$ calculated by indirect means (Allan *et al*., 2007; Platt *et al*., 2004), which is further limited by the ability of measurement techniques to accurately quantify short lived species at low mixing ratios. Our knowledge of the Cl budget therefore depends on the accurate measurement of its precursors, namely $ClNO_2$ and major reaction pathways of the chlorine atom upon liberation in the daytime. Measurements to date show that the mixing ratio of $ClNO_2$ vary geographically from below limits of detection to hundreds of ppt (Mielke *et al*., 2015, Phillips *et al*., 2012, Bannan *et al*., 2015) and up to 3 parts per billion (ppb) (Tham *et al*., 2014, Riedel *et al* 2014, Liu *et al*., 2017) in heavily polluted urban areas. To date, the majority of these measurements have been performed in the United States, ~~although~~with more recent research ~~globally and~~ in Europe, China ~~have recently been published~~etc. (Tham *et al*., 2014, T Wang *et al.,* 2016, X. Wang *et al.,* 2017, Z. Wang, Liu *et al*. 2017). A major factor in the variation of $ClNO_2$ mixing ratios is the availability and abundance of aerosol chloride which can vary significantly, although is predominantly present as sodium chloride from sea salt.

Iodide adduct ionization has previously been applied to measure inorganic halogens in ambient air (Osthoff *et al*., 2008, Riedel *et al*., 2012, Thornton *et al*., 2010, Le Breton *et al*., 2017a) using mass spectrometers with quadrupole mass analysers. This technique involves periodically changing the tuning of the spectrometer to allow transmission of a particular mass ion to the detector. Several species are therefore often "chosen" for detection in order to achieve high enough time resolution. Recent developments and availabilities of a Time of Flight Chemical Ionisation Mass Spectrometer (ToF-CIMS) have enabled the simultaneous measurement of all detectable ions by an ionization technique via high frequency full mass spectral collection. The high resolving power (3500) of this technique also enables much lower limits of detection for species which may have the similar mass to a compound that is much more abundant via multi peak fitting. This technique has previously been applied for the measurement of $ClNO_2$ and $Cl_2$ (Faxon *et al* 2015) and recently for Cl-VOCs (Wang *et al.,* 2017) in the gas phase. In this study, a ToF-CIMS utilizing the FIGAERO (Filter Inlet for Gas and AEROsols) is deployed at a site in semi-rural Beijing, China to measure the gas and particle phase precursor ($ClNO_2$, $N_2O_5$) and selective halogen containing species at high time frequency and resolving power to further our understanding of the chlorine atom budget in this region and its potential fate.

## 2. Experimental

### 2.1 Site description

The data presented here ~~was~~were collected during the inter-collaborative field campaign, within the framework of a Sino-Sweden research project "Photochemical Smog in China" aimed to further our understanding of the episodic

pollution events in China through gas and particle phase measurements with numerous analytical instruments. The
laboratory setup in the Changping University Campus of PKU was situated at a semi-rural site 40 km North West
of Beijing close to Changping town (40.2207° N, 116.2312° E). The general setup has previously been described
by Le Breton *et al*., 2017b.
All instruments sampled from inlets setup in a laboratory 12 metres high from the 13[th] May 2016 to 23[rd] June 2016.
The site has a small town within its vicinity and some small factories within 5 kilometers. A High Resolution Time
of Flight Aerosol Mass Spectrometer (HR-ToF-AMS) was utilized to measure the mass mixing ratios and size
distributions of non-refractory species in submicron aerosols, including organics, sulfate, ammonium and chloride
(DeCarlo *et al, 2006*, Hu *et al.,* 2013). The setup of this instrument has been previously described by Hu *et al.,*
(2016). Photolysis rates were measured by a commercial spectradiometer for O3, NO2, HCHO, HONO and H2O2.
(Metcon UF CCD), the instrument was calibrated by high power halogen lamp after the field campaign. The
photolysis rate of ~~any given~~other  related species ~~was calculated by normalizing to the cross section and quantum~~
~~yields taken from the recommendations~~were scaled by the recommendation of the Jet Propulsion Laboratory (JPL)
kinetic evaluation report (Burkholder et al., 2015). Before the campaign the was instrument calibrated through
comparison with a chemical actinomter in 2014 (Zou et al., 2016).
An Ionicon Analytik high sensitivity PTR-MS (Proton TRansfer Mass Spectrometer) as described by de Gouw
and Warneke *et al*, (2007) provided supporting precursor VOC measurements. Detailed information about the PTR
MS measurements can be found in Yuan et al 2012 and 2013. In brief, 28 masses are measured ~~for~~throughout the
campaign at 1 Hz. Zero air, which was produced by ambient air passing through a platinum catalytic converter at
350 °C (Shimadzu Inc., Japan), was measured for 15 min every 2.5 hours to determine the background. ~~used to~~
~~measure background Aromatics~~Aromatic masses (m/z 79 for benzene, m/z 93 for toluene, m/z 105 for styrene,
m/z 107 for C8 aromatics and m/z 121 for C9 aromatics), oxygenated masses (m/z 33 for methanol, m/z 45 for
acetaldehyde, m/z 59 for acetone, m/z 71 for MVK+MACR and m/z 73 for MEK), isoprene (m/z 69) and
acetonitrile (m/z 42) were calibrated by using EPA TO15 standard from Apel-Riemer Environmental Inc., USA.
Formic acid (m/z 47), acetic acid (m/z 61), formaldehyde (m/z 31), and monoterpenes (m/z 81 and m/z 137) were
calibrated by permeation tubes (VICI, USA). The uncertainties of most species are below 10%, which is detailed
in the previous work (Liu, Y. 2015, ACP).
**2.2 ToF-CIMS setup**
Gas and particle phase ambient species were measured using an iodide ToF-CIMS (resolving power of 3500)
coupled to the FIGAERO inlet (Lopez-Hilfiker *et al*., 2014). The setup for this campaign has previously been
described by ~~le~~Le Breton *et al*. (2017b). Briefly, the iodide ionization scheme was utilised to acquire non-
fragmented ions of interest by passing UHP $N_2$ over a permeation tube containing liquid $CH_3I$ (Alfa Aesar, 99%),
and through a Tofwerk X-Ray Ion Source type P (operated at 9.5 kV and 150 μA) to produce the iodide reagent
ions. The ionized gas was then carried out of the ion source and into the Ion-Molecule Reaction (IMR) chamber,
which was heated to 40 degrees Celsius to reduce wall loss, through an orifice (Ø = 1 μm). The inlet lines were 2
metres long and composed of copper tubing (12 mm) for the aerosol inlet and Teflon tubing (12 mm) for the gas
sample line. Particles were collected onto a Zefluor® PTFE membrane filter at the same rate as the gas inlet line
sampling, 2 SLM. The FIGAERO was operated in a cyclic pattern; 25 minutes of gas phase measurement and
simultaneous particle collection, followed by a 20 minute period during which the filter was shifted into position
over the IMR inlet and the collected particle mass was desorbed.

## 2.3 Calibration

In the field formic acid calibrations were performed daily utilising a permeation source maintained at 40 °C. A dry
$N_2$ flow (200 sccm) was passed over the permeation source and joined a 2 SLM $N_2$ flow line directed towards the
inlet. The mixing ratio of the flow was determined by mass loss in the laboratory after the campaign. The sensitivity
of the ToF-CIMS to formic acid was found to be 3.4 ion counts per ppt $Hz^{-1}$ for $1x10^5$ iodide ion counts.
$N_2O_5$ was synthesized by mixing 20 ppm $O_3$ with pure $NO_2$ (98%, AGA Gas) in a glass vessel and then passing
the mixture through a cold trap held at -78.5 °C by dry ice. The $N_2O_5$ was transferred to a diffusion vial fitted with
a capillary tube (i.d. 2 mm). The $N_2O_5$ diffusion source was held at a constant temperature (-23 °C), and the mass
loss rate was characterized gravimetrically for a flow rate of 100 sccm. The same flow was added to a dry nitrogen
inlet dilution flow of 2 SLM to calibrate the CIMS. $ClNO_2$ measurements were quantified by passing the $N_2O_5$
over a wetted NaCl bed to produce $ClNO_2$. The decrease in $N_2O_5$ from the reaction with NaCl was assumed to be
equal to the mixing ratio of $ClNO_2$ produced (i.e., a 100% yield). Conversion of $N_2O_5$ to $ClNO_2$ can be as efficient
as 100% on sea salt, but it can also be lower, for example if $ClNO_2$ were to convert to $Cl_2$ (Roberts et al., 2008).
For NaCl the conversion efficiency has however been as low as 60% (Hoffman et al., 2003). In this calibration we
have followed the accepted methods of Osthoff et al., (2008) and Kercher et al., (2009) that show a conversion
yield of 100% and have assumed this yield in the calibrations of this study.  The lower detection limit of the CIMS
to $N_2O_5$ and $ClNO_2$ was found to be 9.5 and 1.2 ppt respectively for 1minute averaged data. ~~Using the~~The error in
the individual slope of the calibrations results ~~in~~yields a total uncertainty of 30% for both $N_2O_5$ and $ClNO_2$. These
sensitivities for $N_2O_5$ and $ClNO_2$ (9.8 and 1.6 ion counts per ppt $Hz^{-1}$ for ~~$1x10^5$~~$1 \times 10^5$ iodide ion counts) were
applied relatively to that of formic acid. The other inorganic halogens reported in this work are ~~assumed to have~~
~~the same sensitivity as ClNO2. This is in line with that Le Breton et al. (2017) reported many inorganic halogens~~
~~possess a similar, if not the same, sensitivity, which is also supported by our chloroacetic acid calibration.~~reported
in ion counts. Other acids identified by CIMS which are reported in the literature are given the sensitivity of $N_2O_5$
to provide a minimum concentration so no concentrations are over estimated.
A post campaign calibration of chloroacetic acid (99%, Sigma Aldrich) was utilised to ~~apply~~characterise a
sensitivity factor for ~~all~~a Cl ~~VOCs measured during the campaign~~VOC. The calibration was performed using the
same method as for formic acid and gave a sensitivity of 1.02 ion counts $ppt^{-1}$ Hz when normalized to ~~$1x10^5$~~$1 \times$
$10^5$ $I^-$ ion counts. ~~Using~~ This similar sensitivity to that of the Cl VOC to that of $ClNO_2$ could imply a relative
~~sensitivities will increase the uncertainties, but is a commonly applied method within~~sensitivity may be appropriate
to constrain the ~~CIMS community~~mixing ratios of all Cl VOCs, although ~~in~~further work is required to confirm
this ~~specific case it is very likely that~~ and therefore the ~~sensitivity is similar for all inorganic/organic halogens, as~~
~~demonstrated by Le Breton et al. (2017a).~~manuscript reports all Cl VOC measurements in units of ion counts.

**2.4 Model setup**
The EMEP MSC-W chemical transport model (Simpson et al., 2012, Simpson et al., 2017) driven by meteorology
from the WRF-ARW model (Skamarock et al., 2008) was utilised to support source analysis of the particulate
chloride. The model was run on two nested domains (0.5˚ and 0.1667˚ resolution respectively) with biomass
burning emissions from the two databases FINN and GFAS, and anthropogenic emissions from the MEIC
inventory (http://meicmodel.org/). Two versions of the model, one getting emissions from open biomass burning
from the Fire Inventory from NCAR (FINN) (Wiedinmyer et al., 2011) and one getting them from the Global Fire
Assimilation System (GFAS) (Kaiser et al., 2012), were run for the entire period of the Changping measurement
campaign.
**Results and Discussion**
**3.1 Peak identification and quantification**
Peak fitting was performed utilizing the Tofware peak fitting software for molecular weights up to 620 AMU. The
standard peak shape was fitted a peak on the spectra until the residual was less than 5%. Each unknown peak was
assigned a chemical formula using the peaks exact mass maxima to 5 decimal places and also isotopic ratios of
subsequent minor peaks. An accurate fitting was characterized by a ppm error of less than 5 and subsequent
accurate fitting of isotopic peaks. The analysis here focuses on species identified in the mass spectra considered to
possibly play important roles with respect to the night-time chlorine reservoir and several other key night-time
oxidants; $ClNO_2$, HCl, $Cl_2$, ClO, HOCl, OClO, $ClONO_2$, $N_2O_5$ and Cl-VOCs. Figure 1 displays the average mass
spectra for the measurement campaign and the peak fitting applied for ClO and $ClNO_2$. All species were
represented by a dominant peak with a multi peak fit, although a number of co-existing peaks were present for
much of the campaign. This signifies the importance of high resolution fit data and the need for high resolution
measurements. A quadrupole CIMS may not be able to resolve the peak adjacent to ClO at m/z 178 (dominant
peak is $IC_6F_3HO_3^-$) and the second dominant peak for the $ClNO_2$ fit (cluster of $HNO_3$ with water) would result in
a 10% over estimation.
**3.2 $N_2O_5$ measurements**
The CIMS and a Cavity Enhanced Absorption Spectrometer (CEAS) measured $N_2O_5$ (Wang *et al*., 2017)
simultaneously from the 13th May 2016 to the 6th June 2016. However, given the use of the FIGAERO, the CIMS
alternated measurements between gas and particle phases so did not generate a completely continuous gas phase
time series. Here, the CEAS is utilised to validate the CIMS $N_2O_5$ (at m/z 235) measurements and also instrument
stability. The CEAS utilised a dynamic source by mixing $NO_2$ and $O_3$ to generate stable $N_2O_5$ for calibration (Wang
*et al*., 2017). The source was used to calibrate the ambient sampling loss of $N_2O_5$ in the sampling line, filter, the
preheater cavity and optical cavity. This was performed pre and post campaign. During the campaign the
reflectivity of the high reflectivity mirror was calibrated daily and filter changed hourly. The simultaneous
measurements of $N_2O_5$ can be shown in Figure 2 for one minute averaged data. The time series ~~show a~~shows good
agreement for both background mixing ratios during the day (sub 10 ppt) and high night-time mixing ratios (up to
800 ppt), excluding one night. The highest $N_2O_5$ levels observed by both the CEAS and CIMS were observed on
the 3rd June although the CEAS reports 880 ppt whereas the CIMS reports 580 ppt. If included in the analysis the
$R^2$ is 0.71 and when excluded it is 0.76. To date the reason for this deviation during that night is not known but it
should be stressed that $N_2O_5$ measurements are delicate and highly ~~depending~~dependent on sampling condition,
e.g. the RH. Nevertheless, excluding this night from the comparison, a slope of 0.85 is observed and ~~a y~~an offset
of 0.9 ppt. The diurnal profile in Figure 2 represented the difference between the two measurements throughout
the campaign. The largest error between the two measurements occurs at night during the higher levels of $N_2O_5$,
although averaging at 4 ppt (representing 11% error on the average campaign concentration). Differences could
arise from a number of various factors. Inlet differences such as the CIMS heated IMR (to 40 ˚C to reduce wall
loss), residence time and ambient $NO_2$ can all change thermal decomposition and wall loss rates between the
instruments, which is determined for the CEAS in Wang *et al.* (2017) but not for the CIMS in this work. Also, the
separate inlets were facing in different directions within the same laboratory, possibly enabling local wind patterns
to affect the mixing ratios reaching each instrument.
The CEAS data was further utilised to assess any sensitivity changes for the CIMS that daily carboxylic acid
calibrations did not account for. A time series of hourly factor differences between the CIMS and CEAS was
implemented into ~~the~~these data to weight the measurements to a normalised sensitivity. The high level of
agreement (R$^2$ of 0.76) from low mixing ratio measurements and a species with a short lifetime from different
inlets confirms the accuracy and reliability of the CIMS measurements for this campaign.
Generally, $N_2O_5$ was detected throughout the campaign with a clear diurnal variation peaking at night-time and
rapidly falling to below limits of detection in the daytime as a result of photolysis of $N_2O_5$ and $NO_3$. The campaign
mean night-time mixing ratio was 121 ppt with a standard deviation of 76 ppt. The maximum mixing ratio of $N_2O_5$
observed was 880 ppt on the 3rd June. This range of mixing ratios lie within the recently reported values in the
literature, but not at the extreme mixing ratios as observed in Germany (2.5 ppb) (Phillips *et al.*, 2016) or Hong
Kong (7.7 ppb) by Wang *et al.* (2016) and Brown *et al.* (2017). Although the mean mixing ratios do not increase
significantly during the pollution episodes, the maximum mixing ratios detected overnight increase by up to a
factor of 4. Further analysis of $N_2O_5$ nighttime chemistry was performed by Wang et al (2018) who calculated an
average steady state lifetime of 310 + 240 s and mean uptake coefficient of 0.034 ± 0.018.
**3.3 Inorganic chlorine: Abundance, profiles and source**
**3.3.1 Abundance and profiles**
Mean diurnal profiles of HCl, $Cl_2$, $ClONO_2$, HOCl, ClO and $ClNO_2$ are displayed in Figure 3 from data between
the 23rd May and the 6th June. HCl exhibited a standard diurnal profile increasing in mixing ratio throughout the
day and peaking at 4 pm which then fell off slowly at night. The mean HCl campaign mixing ratio was 510 ppt
(standard deviation (σ) 270 ppt) and the maximum HCl mixing ratio was 1360 ppt on the 30th June. $Cl_2$ exhibited
a diurnal profile peaking at both ~~the~~ night-time and daytime. High mixing ratios were observed at night followed
by a sharp loss at sunrise and a general build-up throughout the day. The campaign mean mixing ratio was 0.65
ppt (σ 0.5 ppt) and the maximum mixing ratio was 4.2 ppt on the 4th June just before midnight. This agrees well

| 1 | with recent urban measurements of $Cl_2$ in the USA where Faxon et al. (2015) observed a maximum of 3.5 ppt and |
| 2 | Finley et al. (2006) observed up to 20 ppt in California. Up to 500 ppt $Cl_2$ has recently been reported in the Wangdu |
| 3 | County, South West of Beijing (Liu et al., 2017). Although the mixing ratios we report here are significantly lower, |
| 4 | as detailed later, their source ~~maybe~~may be of similar origin, which is indicated to be from power plant emissions. |

The diurnal profile of HOCl peaked during the daytime via its main formation pathways are via reaction of ClO
and $HO_2$ and Cl with OH. Interestingly, the ClO in this work exhibits a night-time diurnal peak, contradicting
known formation pathways via Cl reaction with $O_3$ and the photolysis of $ClONO_2$. The complexity continues as
$ClONO_2$ also peaks during the night, given that its main known formation pathway is via reaction of ClO (produced
at sunrise via $ClNO_2$ photolysis) with $NO_2$. The misidentification of $ClONO_2$ and ClO is not thought to be a
possible reason for these discrepancies due to the low number of mass spectral peaks that have maxima at night
and the mass defect of chlorine making the peak position unique to chlorine containing molecules. IMR chemistry
is also not a possible source as these reactions would occur throughout the day, therefore skewing all of the data
and not just the night-time levels, although there is a possibility that $ClONO_2$ can be formed in the IMR by reactions
between ClO and $NO_2$. It is hypothesized that in extremely high OH and $HO_2$ mixing ratios, all ClO is rapidly
converted to HOCl, limiting the formation on significant levels of ClO and subsequently $ClONO_2$. Khan et al
(2008) suggest that Cl atoms of around ~~2x1042~~ $2 \times 10^4$ molecules $cm^{-3}$ could be present at night via analysis of
alkane relative abundance. Although a formation mechanism is not proposed, it provides further evidence that ClO
formation at night-time is possible and may represent an unknown reaction pathway, which would agree with the
measurements presented in this work.
$ClNO_2$ exhibited a similar diurnal profile as $N_2O_5$, peaking at night-time and lost during daylight due to photolysis.
The campaign mean night-time mixing ratio was 487 ppt. The maximum mixing ratio observed was 2900 ppt on
the 31st May, similar to that previously measured at semi-rural site in Wangdu (up to 1500 ppt) (Liu et al., 2017),
Mount Tai (2000 ppt) (Wang et al., 2017), but lower than that in Hong Kong (4 ppb) (Wang et al 2016).
**3.3.2 Source of chloride**
The high levels of $ClNO_2$ indicate a local significant source of chlorine to support these observations. The
dominant source of chlorine atoms for $ClNO_2$ production within models, such as the Master Chemical Mechanism
(MCM), is from sea salt. However, the site is situated 200 km from the Yellow Sea and therefore this origin would
have a low probability. The mean AMS chloride mass loading was 0.05 $\mu g\ m^{-3}$ for the campaign with a maximum
of 1.7 $\mu g\ m^{-3}$. The $Cl^-$ from the AMS appears to be correlated strongly with CO and $SO_2$, possibly originating from
power plants or combustion sources. It should be noted that the AMS data does not include refractory aerosol and
also has a cut off size larger than ~~anticipate~~the anticipated size of sea salt particles. Instead, the high $Cl^-$ observed
appears to originate from mainland areas to the site (Figure 4) rather from the nearest coast, further supporting a
~~strong~~ anthropogenic source. Tham et al., (2016) observed a strong correlation of aerosol chloride with $SO_2$ and
potassium from measurements done during the same season in 2014 at Wangdu (semi-rural site 160 km south
West of Beijing) and suggested ~~contribution~~contributions to fine chloride from burning of coal and crop residues.
The latter was also supported by satellite fire spot count data (Tham et al., 2016). Riedel et al. (2013) have
previously reported high $ClNO_2$ mixing ratios observed from urban and power plant plumes measuring high mixing
ratios of gas phase $Cl_2$. The correlation with $SO_2$ indicates coal burning as a potential source of particulate chlorine
which is known to be a significant source of PM in the Beijing region (Ma et al., 2017), and the correlation with
CO and benzene could be an indicator of biomass burning (Wang et al., 2002). To support this analysis, figure S1
displays a wind rose plot in which radial and tangential axes represent the wind direction and speed (km h$^{-1}$). The
colour bar represents the PM2.5 concentration. We could see that during the campaign, the severe pollution was
from the south and southwest, with little contribution from the east part. Therefore, ~~we could deduce~~it is likely that
little contribution of the chloride was from the ocean.
In order to test the hypothesis of biomass burning as a source of particulate chlorine, biomass burning emissions
and transport utilising the EMEP MSC-W chemical transport model driven by meteorology from the WRF-ARW
model (Skamarock et al., 2008) were used. Neither of the two biomass burning databases used (FINN and GFAS)
contained data on chlorine emissions, so instead the biomass burning emissions of CO (CObb) were tracked and
compared ~~to~~with the total mixing ratio of CO (COt) at the Changping site. CO was chosen since the measurements
at Changping had shown a strong correlation between CO and $ClNO_2$ and because CO could be expected to be co-
emitted with chlorine for both biomass burning and industrial combustion.
Figure S2 (supplementary) shows the time series of the measured $ClNO_2$ mixing ratios at the Changping site, as
well as the modelled mixing ratios of COt and CObb. CObb is shown for calculations using either the FINN or the
GFAS data base, while for clarity the COt is only shown using the FINN data base. ~~From this figure it~~It is clear
that mixing ratios of CObb are very low compared ~~to~~with COt~~.~~ (figure S2). The two pollution episodes on May
18-May 23 and May 28-June 5, are to some extent visible in all time series, but for the biomass burning CO series,
the second episode is much less pronounced. Night-time averages of the mixing ratios shown in figure S2 were
calculated for each night for the time period 18:00 to 08:00 local time (UTC+8), roughly corresponding to the
period when $ClNO_2$ is not destroyed by photolysis. Nights with significant amount of missing data for the
measurements were excluded. Figure S2 shows scatter plots of these averages of $ClNO_2$ against the averages of
the other species including their linear fits. The $R^2$ value for these fits were 0.48, 0.04, and 0.21 for COt, CObb
FINN, and CObb GFAS respectively. The fact that mixing ratios of CObb ~~is~~are so much smaller than COt
according to the model, combined with the much better correlation for COt than for CObb strongly suggests that
industrial emissions are the dominant source of chlorine, rather than biomass burning. To further investigate the
source of chloride, the model was also run to calculate sea salt levels instead of CO. This resulted in a poor
correlation between sea salt and the $ClNO_2$ (figure S4). The absolute levels of sea salt calculated by the model
were also very low, unlikely to be able to produce the observed mixing ratios of $ClNO_2$ as observed by CIMS.

## 3.4 Particle phase $ClNO_2$

A particle desorption profile was observed in the high resolution data for $ClNO_2$. The count increase at this 1 AMU
mass can be attributed to two sources; $SO_3$ and $ClNO_2$ as shown in Figure 5. The $SO_3$ peak is predominantly found
in the particle phase and is below the limit of detection (LOD) in the gas phase. During initial analysis of ~~the~~these
data, $SO_3$ interfered with the $ClNO_2$ peak fitting and attributed its counts to $ClNO_2$ in the particle phase as its $^{33}$S
ion is only 0.005 AMU away from the $ClNO_2$ peak. Upon its inclusion into the peak list and utilisation of the
Tofware feature which constrains isotopes and reallocates the signal appropriately, $ClNO_2$ remains to indicate a

strong desorption profile. The diurnal cycle of these desorptions correlate well with the $ClNO_2$ gas phase profile, indicating a correct assignment of the counts to particle phase $ClNO_2$. The desorption profiles with respect to temperature also exhibit a thermogram structure and not ~~e.g.~~for example a gas phase leak into the system which could have accounted for the correlation with the gas phase time series. This suggests the possible presence of $ClNO_2$ in the particle phase. Another possible explanation could be the deposition of $ClNO_2$ from the gas phase onto the filter as the ambient air flows through the FIGAERO.

If we assume the analysis and collection technique is correct, we see an average particle to gas phase partitioning of 0.07, with a maximum of 0.33 and a minimum of 0.009. The average mixing ratio of $ClNO_2$ collected onto the filter during desorption is 13 ppt with a maximum of 120 ppt. Previous modelling studies assume all $ClNO_2$ is in the gas phase due to the low Henry's law constant e.g. for the TexAQS II campaign they calculated that 0.1 ppb in the gas phase would yield 0.54 ppt in the particle phase (Simon *et al*., 2008). However, ~~this~~these data ~~indicates~~suggests that a non-negligible amount of the chlorine associated with $ClNO_2$ is not liberated from the particle phase, assuming that no additional $ClNO_2$ is formed by thermally driven reactions. The slope of the particle to gas phase CIMS data is calculated to be 0.048, a factor of 96 higher than using the Henry's law coefficient to estimate the particle mixing ratio.

### 3.5 $ClNO_2$ daytime persistence and Cl liberation

Both $ClNO_2$ and $N_2O_5$ are photolytically unstable, with studies reporting lifetimes on the order of hours for $ClNO_2$ depending on the solar strength (e.g. Ganske et al., 1992, Ghosh et al., 2011). Nocturnal $ClNO_2$ removal pathways have generally been reported to be negligible, with $ClNO_2$ being assumed to be relatively inert (Wilkins et al., 1974; Frenzel et al., 1998; Rossi, 2003; Osthoff et al., 2008), but the work of Roberts et al., (2008) and Kim et al., (2014) would suggest that this may not be strictly true. However, given that the average diurnal profile does not show the importance of nocturnal removal pathways in this study, observed losses are attributed solely to photolysis, with $J(ClNO_2)$ controlling the lifetime.

Rapid photolysis can be observed for $N_2O_5$ in Figure 6 showing a near instant drop below LOD, whereas the $ClNO_2$ mixing ratio not only persists for up to 7 hours, but also shows evidence of an increase in mixing ratio at 7 am (Figure 6). This is observed throughout the campaign and has been frequently observed in the previous study at Wangdu (Tham *et al*., 2016). The breakdown of the nocturnal boundary layer and inflow of air masses from above, carrying pollution from nearby industry/ies is a likely cause of this persistence of possible increase of $ClNO_2$. Liu *et al*. (2017) also observed high daytime mixing ratios of $ClNO_2$ (60 ppt) at the Wangdu site which they attribute to a possible oxidation mechanism due its correlation with $O_3$ and $Cl_2$ providing a daytime formation pathway to maintain mixing ratios against its rapid photolysis.

Consistent with past measurements and the measurements of this study, $ClNO_2$ is expected to provide a significant source of Cl during day time hours, presenting a potentially significant source of the reactive Cl atom during the day. Its rapid photolysis rate and elevated mixing ratios enables Cl to compete with OH oxidation chemistry, the known dominant daytime radical source. Here, a simple steady state calculation will be used to determine the Cl atom mixing ratio summarised ~~below~~as follows, but detailed within the supplementary:

| | | |
|---|---|---|
| 1 | $Cl_2 + hv \rightarrow Cl + Cl$ | (~~1~~ss1) |
| 2 | $ClNO_2 + hv \rightarrow Cl + NO_2$ | (~~2~~ss2) |
| 3 | $ClONO_2 + hv \rightarrow ClO + NO_2$ | (~~3~~ss3) |
| 4 | $HOCl + hv \rightarrow OH + Cl$ | (~~4~~ss4) |
| 5 | $OClO + hv \rightarrow O + ClO$ | (~~5~~ss5) |
| 6 | $OH + HCl \rightarrow Cl + H_2O$ | (~~6~~ss6) |
| 7 | $Cl + O_3 \rightarrow ClO + O_2$ | (~~7~~ss7) |
| 8 | $Cl + CH_{4\ equivalent} \rightarrow HCl + products$ | (~~8~~ss8) |

$[Cl]_{SS} = \{2J_1[Cl_2] + J_2[ClNO_2] + J_3[ClONO_2] + J_4[HOCl] + J_5[OClO] + k_7[OH][HCl]\} / \{k_7[O_3] + k_8[CH_4]_{equivalent}\}$

11  (9)

Where $[CH_4]_{equivalent}$ represents the reactive VOC present as if it were reacting as $CH_4$
Bannan et al., (2105), were able to use this steady state approach to compare the relative loss via reaction with OH
compared with Cl atoms. Although this approach is an estimation, it was shown to produce results comparable ~~to~~
~~results~~ with that of the more rigorous MCM approach~~.~~ although we do acknowledge large errors will be present in
the radical species calculations, which is detailed in the supporting information. Steady state calculations reveal a
sharp rise of chlorine atoms produced at sunrise peaking at 1.~~6x10⁵~~6 x 10⁵ molecules ~~cm³~~cm⁻³ around 7 am which
then gradually decreases, contributing to Cl atom production until 2 pm (Figure 7a). Supporting $Cl_2$, $ClONO_2$,
OClO, HOCl and HCl measurements by CIMS report that chlorine atoms can sustain a relatively high production
rate until 3 pm as evidenced by the daytime build-up of HCl and $Cl_2$. $ClNO_2$ on average contributes ~~to~~ 78% of the
chlorine atoms produced from inorganic halogens with 13% from $Cl_2$. $ClNO_2$ also represents over 50% of the
chlorine atoms until midday. After ca. 3 pm $Cl_2$ and HCl becomes the more dominant Cl atom ~~source~~sources. On
the night where the highest $ClNO_2$ mixing ratios were measured, 90% of the chlorine atoms originated from $ClNO_2$
photolysis until 2 pm and HCl and $Cl_2$ then become main contributors ~~sustaining~~ until 4 pm (up to 80%). $ClONO_2$,
HOCl and OClO appear to be insignificant contributors to chlorine atom production throughout the campaign
compared with $ClNO_2$, HCl and $Cl_2$.
To put these chlorine atom mixing ratios into a more global perspective, data collected by the University of
Manchester from a marine site and an urban European site have been compared in Figure 7b. Bannan et al., (2015)
and (2017) previously utilised a box model to calculate Cl atom mixing ratios during the campaign so that the rate
of oxidation of VOCs by Cl atoms could be compared with oxidation by measured OH and measured ozone. The
simple steady state calculation described previously will be used to determine the Cl atom mixing ratio for ~~both~~
this measurement study. The results show that both at the UK marine and urban site ~~max~~maximum chlorine atom
mixing ratios are more than an order of magnitude lower than the mean of Beijing. It should ~~however~~ be noted that
the only source of Cl in the UK studies was $ClNO_2$, but given the dominance of $ClNO_2$ in this study the
measurements presented here suggest a high importance of the chlorine chemistry for the Asian air chemistry.
Studies of ~~chloride~~chlorine radical production in Los Angeles by Riedel *et al.* (2012) and Young *et al.* (2014)
indicate that the high production rate in Beijing is somewhat typical of urban sites, although HCl and ClNO$_2$
contribution to radical production is the same, whereas here we see very little chloride radical production from
HCl in comparison ~~to~~with ClNO$_2$.
Although this study does not reach the scope of characterising O$_3$ and RO$_x$ production from chlorine atom
chemistry, statistics are often reported with ClNO$_2$ morning chemistry via modelling simulations, we can put into
perspective the mean and maximum mixing ratios relative to other studies. Tham *et al*. (2016) recorded a maximum
ClNO$_2$ mixing ratio of 2070 ppt from a plume originating from Tianjin, the closest megacity to Beijing, and report
a 30% increase in RO$_x$ production and up to 13% of O$_3$ production. Liu *et al* (2017) observed peak mixing ratios
up to 3 ppb and similar diurnal mixing ratios which they calculated contributes to a 15% enhancement of peroxy
radicals and 19% O$_3$ production. Wang *et al*. (2016) report up to 4.7 ppb of ClNO$_2$ in Hong Kong and calculated
a maximum increase of 106% of HO$_x$ in the morning and an enhancement of ~~following daytime~~ O$_3$ production the
next day by up to 41%. ~~It~~Therefore, it is ~~therefore~~ evident that this work supports similar studies in Asia that
conclude that chlorine atom oxidation significantly contributes to atmospheric oxidation via RO$_x$ and O$_3$
production. Although several studies have demonstrated a non-negligible impact of chlorine oxidation chemistry
(e.g. Oshoff et al., 2008, Riedel et al., 2014 and Sarwar et al., 2014), the impact of Cl chemistry varies significantly
between various areas and atmospheric conditions, e.g. Bannan et al., (2015, 2017) deemed the impact from
chlorine atom chemistry to be relatively low with respect to O$_3$ production and competing with OH radicals for
VOC oxidation[22].
**3.6 VOC oxidation by chlorine atoms**
Steady state calculations of OH (as described by Whalley *et al*., 2010) estimate that campaign average maximum
mixing ratio was 7 x 10$^6$ molecules cm$^3$ (Figure 7b), 6 times greater than the maximum chlorine atom mixing ratio
and 14 times higher than the average chlorine atom mixing ratio. Pszenny *et al*. (2007) report estimated OH to
chlorine atom ratios, from VOC lifetime variability relationships, of 45 to 199 along the East Coast of the United
States. Although the ratio appears much larger than calculated in this work, here we present not only significantly
~~high~~higher mixing ratios of ClNO$_2$ which are appearing to be a consistent conclusion from measurements in Asia,
but also the chlorine within this study appears to originate from an anthropogenic origin rather than marine,
possessing the ability to supply a much larger reservoir of halogens to be liberated through photolysis.
The relative oxidation rate of the chlorine atom and OH to VOCs can vary greatly. Rate coefficients for reaction
of Cl atoms with some volatile organic compounds have been shown to be up to 200 times faster than the
comparable reaction with OH.  The ratio reported here is significantly less than this each day, Cl can subsequently
dominate VOC oxidation for some fraction of the day. Here, the diurnal maxima of the chlorine atom and OH
differs by 5 hours, enabling chlorine atoms to ~~clearly~~ dominate VOC oxidation earlier in the day before OH mixing
ratios have built up. The relative oxidation rate of VOCs to OH and the chlorine atom also varies greatly, creating
a difference for various VOCs. If an average reaction rate for alkenes and alkanes to Cl and OH is calculated, it is
possible to generalise the significance of each oxidation pathway to qualitatively asses the contribution chlorine
atoms have on oxidation chemistry. It can be seen in Figure 8 that alkenes are much more likely to be oxidised by
OH than Cl, although a significant contribution (15%) is attributed to chlorine chemistry. Although significant if
evaluated on a global level, Liu *et al.*, (2017) estimated that Cl atoms oxidize slightly more alkanes than OH
radicals in a similar region of China, implying the increased scale of chlorine oxidation in China. Alkanes are
known to have a much higher Cl to OH relative reaction rate than alkenes and Cl contribution to oxidation is higher
than OH until midday. The contribution to oxidation remains almost equal for the remainder of the day due to the
persistence of $ClNO_2$ and also relatively high levels of $Cl_2$ and HCl. This analysis is representative of that by
Bannan *et al.* (2015) who report contributions of alkene and alkane oxidation by Cl up to 3 and 15% respectively
from $ClNO_2$ mixing ratios peaking at 724 ppt.
This significant oxidation of VOCs by chlorine atoms will result in different products to that of OH oxidation as
illustrated and that neglecting the contributions made by Cl atoms will significantly underestimate the degree of
chemical processing of VOCs in this study, and other environments where there is a source of Cl atoms. Evidence
of the proposed Cl oxidation of VOCs is validated through detection of selected Cl induced oxidation products by
the ToF-CIMS, all of which are displayed in Table 1.

**3.6.1 Isoprene oxidation by the chlorine atom**

1-Chloro-3-methyl-3-butene-2-one (CMBO, $C_5H_6ClO$), a unique marker of chlorine chemistry, has previously
been measured at mixing ratios up to 9 ppt by offline gas chromatography in Houston Texas (Tanaka *et al.*, 2003)
and in laboratory studies of chlorine-isoprene oxidation (Wang *et al.* (2017)). CMBO exhibited a campaign
maximum of ~~13.2 ppt~~21 and mean of ~~5.16 ppt~~34 ion counts (near similar ppt mixing ratio if the chloroacetic acid
calibration sensitivity is applied) exhibiting a near typical diurnal profile with ~~mixing ratios~~abundance rising
sharply after sunrise, at the same rate as the chlorine atom production but maintaining mixing ratios past noon
longer than that of isoprene and the chlorine atom.
The daily maxima of CMBO varied throughout the campaign and can be explained by the relative mixing ratios
of its precursors; the chlorine atom and isoprene. Its mixing ratio throughout the campaign followed similar
intensities to its precursors and figure 9 highlights its dependence on both Cl atom and isoprene mixing ratios. The
production rate of Cl and mixing ratio of isoprene were relatively low ~~form~~from the 24th to the 27th of May
(1.~~6x10^5~~6 x 10^5 molecules cm$^{-3}$ s$^{-1}$ Cl and 0.5 ppb isoprene), which resulted in relatively low CMBO mixing ratios.
An increase in isoprene and Cl on the 28th to the 30th May was subsequently mirrored by the CMBO levels as
qualitatively expected. On closer inspection of the 30th and 31st May, the mixing ratio of CMBO was lower than
expected on the 30th due to higher chlorine atom and isoprene mixing ratios compared ~~to~~with the ~~31th~~31st. This
could be explained by anticipated higher OH mixing ratio as calculated by the steady state model, which is also
further represented by higher mixing ratios of IEPOX (isoprene epoxydiols, i.e. OH oxidation products) on the
30th. This illustrates how the ToF-CIMS can identify isoprene oxidation products of two competing oxidation
pathways. The high levels of IEPOX on the 28th May can also possibly describe the relatively high levels of CMBO
in the particle phase due to an already well oxidised air mass. CMBO may also not be unique to only isoprene-
chloride reactions and therefore have alternative sources not represented in this data set.
Further daily oxidation rates can be probed via analysis of the related isoprene oxidation products observed by the
CIMS. Figure 10 depicts the diurnal time series of the precursor itself and several Cl-VOC products and IEPOX.

CMBO mixing ratios rise rapidly after sunrise due to the low mixing ratio of OH and high production rate of the chlorine atom. The secondary and tertiary products, $C_5H_9ClO_2$ and $C_5H_9ClO_3$ (also measured in the laboratory by Wang *et al*., 2017) increased in mixing ratio at a much slower rate, but appear to peak later in the day (4 pm) whereas CMBO peaked around 10 am (similar to the $ClNO_2$ peak time) and fall off, due to its further oxidation to form the secondary and tertiary products. IEPOX mixing ratios increased slowly after sunrise and peaked later in the day, as expected due to the availability of OH and competition from the chlorine atom chemistry. The similar time series of the secondary and tertiary products to IEPOX was also reported by Wang *et al.,* (2017) and were suggested to be ideal tracers of SOA production.

### 3.6.2 Anthropogenic Cl-VOC production

A similar unique chlorine oxidation marker in urban coastal areas, has been reported in the literature for 1, 3 butadiene; 4-chlorocrotonaldehyde (CCA) (Wang *et al*., 2000). No measurements of 1, 3 butadiene were made during this field campaign, although due to its common source to benzene (automobile exhausts (Ye *et al*., 1998), we present a comparison of the CCA measured by CIMS and benzene measurements made by the PTR-MS. The intensity of CCA in both the gas and particle phase ~~with mixing ratio in the gas phase up to 13 ppt~~ reflect well the ~~mixing ratios~~abundance of its precursors. The maximum mixing ratio of the chlorine atom coincides with a high mixing ratio of benzene and subsequently CCA on the 30[th] May whereas very low levels of CCA were observed for the beginning of the campaign (Figure 11).

The diurnal time series of benzene (Figure 12) indicates high mixing ratios in the early hours of the day, possibly associated with high anthropogenic activity or an inflow of urban air masses from downtown Beijing. The mixing ratio falls off throughout the day and almost perfectly anti correlates with the CCA gas phase diurnal profile which increases from sunrise and peaks at 3 pm. The particle phase CCA diurnal time series steadily builds up throughout the day and ~~do~~does not peak until late in the evening, providing evidence of SOA production from the chlorine oxidation of anthropogenic pollutants.

### 4. Conclusions

A FIGAERO ToF-CIMS was utilised in Beijing to assess the liberation of chlorine atoms via inorganic halogen photolysis. A suite of inorganic halogens were detected, namely $ClNO_2$ reaching mixing ratios up to 2900 ppt, which is suggested to have an anthropogenic origin due to the particulate chlorine correlation with $SO_2$, benzene and CO. $ClNO_2$ was identified in the particle phase at higher ratios with respect to its gas phase component than expected, which may only prove to be significant at such elevated mixing ratios as observed in East Asia. $ClNO_2$ mixing ratios above LOD persisted up to 7 hours past sunrise, attributed to the lifetime of $ClNO_2$ at these high mixing ratios and a possible in-flow of heavily polluted air masses from the downtown urban area. Supporting $Cl_2$ and HCl mixing ratios proved to be significant contributors to chlorine atom production via steady state calculations ~~enabling an average daytime peak mixing ratio of chlorine atoms of 1.6 x10$^5$ molecules cm$^{-3}$~~.

Compared with data attained from European based campaigns, these mixing ratios exceed marine and urban environments by at least an order of magnitude.

This high mixing ratio of chlorine atoms resulted in a steady state calculated OH:Cl ratios down to a factor of 6, ~~enabling~~suggesting Cl chemistry may be able to ~~not only~~ dominate alkane oxidation until midday but contribute significantly to alkene oxidation throughout the day (15% on average). This enabled significant mixing ratios of Cl-VOCs to be formed providing the first ambient high time resolution measurements of specific Cl-VOC species simultaneously measured in the gas and particle phase. The measured unique markers of chlorine chemistry for both biogenic and anthropogenic precursors provides quantitative and qualitative data to probe the extent of chlorine atom chemistry and how they compete with OH. Simultaneous measurements of the VOC precursors via PTR-MS, and IEPOX, Cl-VOCs with the CIMS provides rich information on SOA formation pathways via both OH and chlorine atom oxidation. Multistep oxidation products of Cl-VOCs were also identified and can provide partitioning information and SOA formation rates and lifetimes.

The results highlight deficiency in chlorine atom chemistry descriptions within models possibly due to a lack in quantification and identification of Cl-VOC products in gas and particle phase. This work provides instrumental capability to probe the competition between OH and Cl oxidation chemistry and quantify their effect on ozone and SOA formation.

**Acknowledgement:**

The work was ~~done~~carried out under the framework research program on 'Photochemical smog in China" financed by Swedish Research Council (639-2013-6917). The National Natural Science Foundation of China (21677002) and the National Key Research and Development Program of China (2016YFC0202003) also helped fund this work.

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

**Table 1. Identified Cl-VOC reaction products, nomenclature of Cl-VOC and precursor compound.**

| Cl-VOC | Potential nomenclature | Precursor |
|---|---|---|
| $CHClO$ | formyl chloride | formaldehyde |
| $C_2H_3ClO$ | chloroacetaldehyde | acetaldehyde |
| $C_3H_5ClNO_5$ | Chloro PPN | PPN |
| $C_2H_3ClNO_5$ | chloro PAN | PAN |
| $C_3H_5ClO$ | chloroacetone | acetone |
| $C_2H_3ClO_2$ | chloroacetic acid | acetic acid |
| $CHClO_2$ | chloroformic acid | formic acid |
| $C_4H_7ClO$ | chloro MEK or butanal | isoprene |
| $C_5H_6ClO$ | CMBO - chloro 3-methyl-3-butene-2-one | isoprene |
| $C_5H_9ClO_2$ | - | isoprene |
| $C_5H_9ClO_3$ | - | isoprene |
| $C_3H_5ClO$ | propanoyl chloride | 1, 3 butadiene |
| $C_8H_9Cl$ | chloroethyl benzene | aromatic |

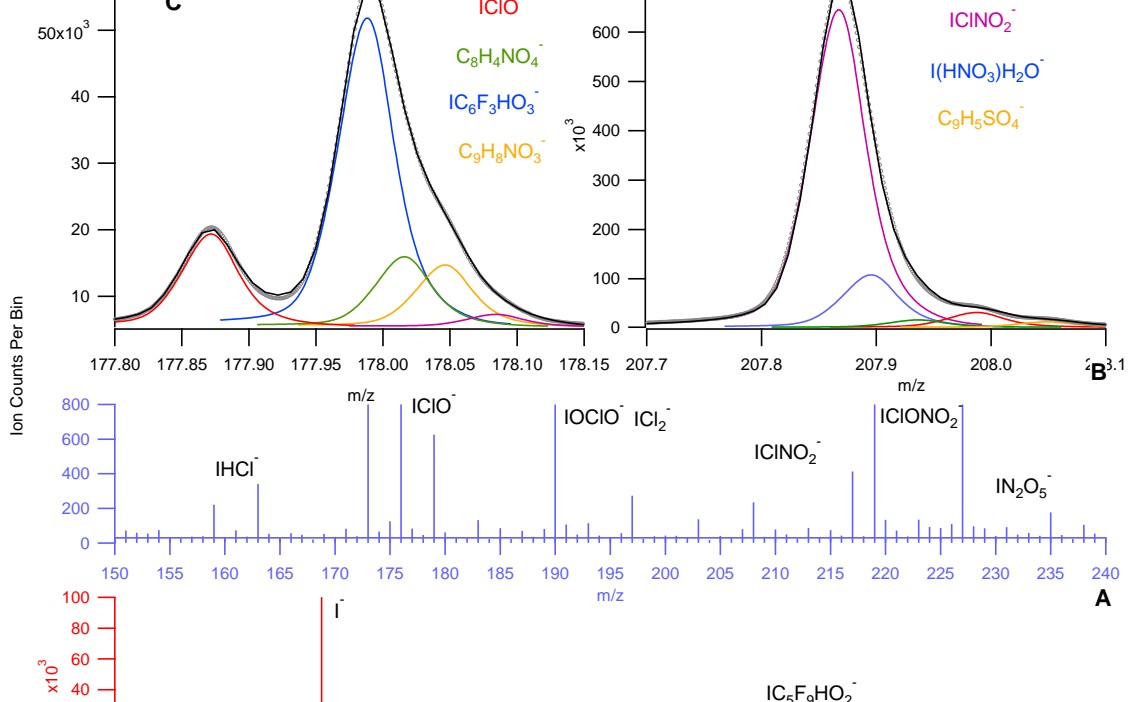

Figure 1. Panel A: Average mass spectrum for the whole measured range. Panel B: Average mass spectrum

for the region that contains all gas phase night time species utilised in this work. A high resolution spectral

1    **fit for ClO and ClNO₂ are displayed with corresponding multi peaks with 0.5 AMU (panels C and D). The**

2    **black line represents the total fit from all peaks. The grey line represents the mass spectral raw data.**

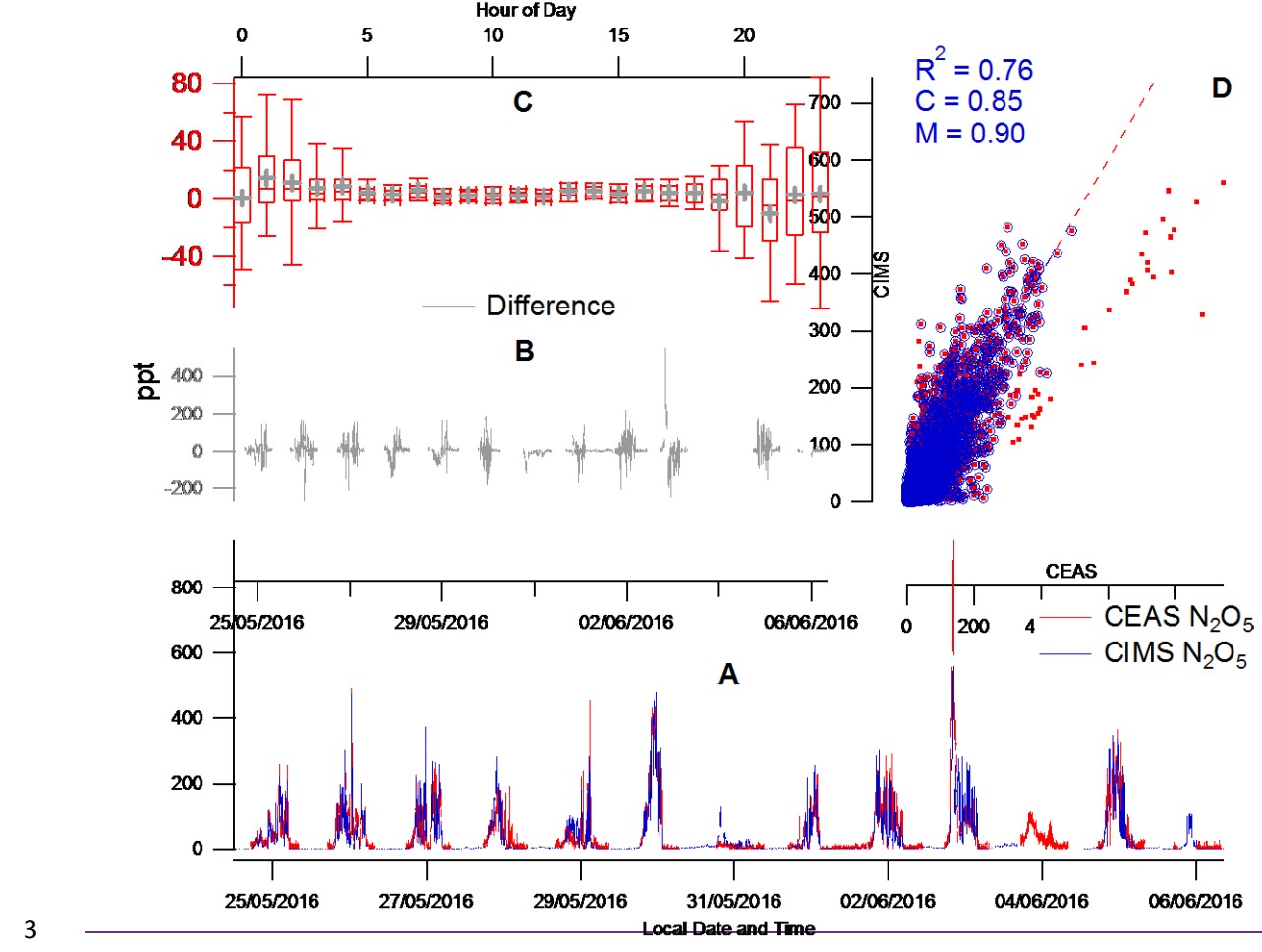

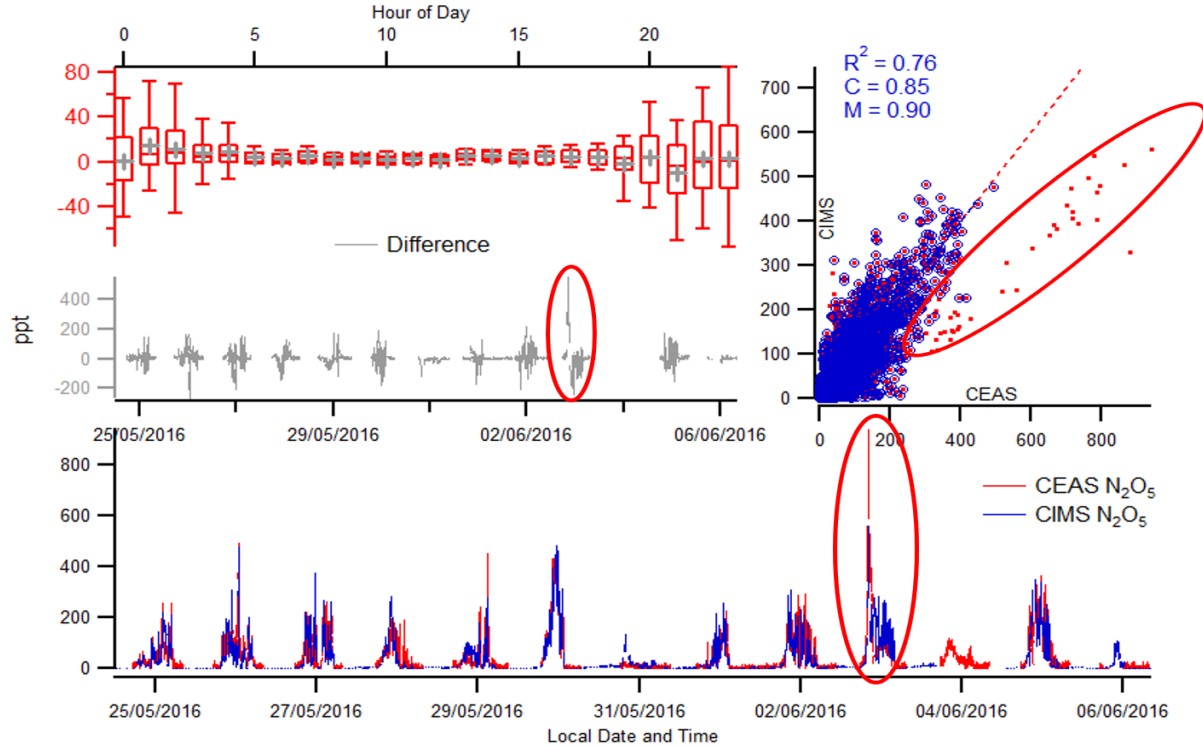

Figure 2. CIMS and CEAS one minute averaged data of $N_2O_5$ with corresponding correlation plot (panel A), campaign and diurnal deviation (panels B and C respectively). The red highlighted periods represent data collected on the 3rd June where a different correlation gradient was observed between CIMS and CEAS. The box and whisker plot represents the diurnal difference for the campaign between the CEAS and CIMS measurements (panel D). C is the y-intercept of the line of best fit and M is the gradient.

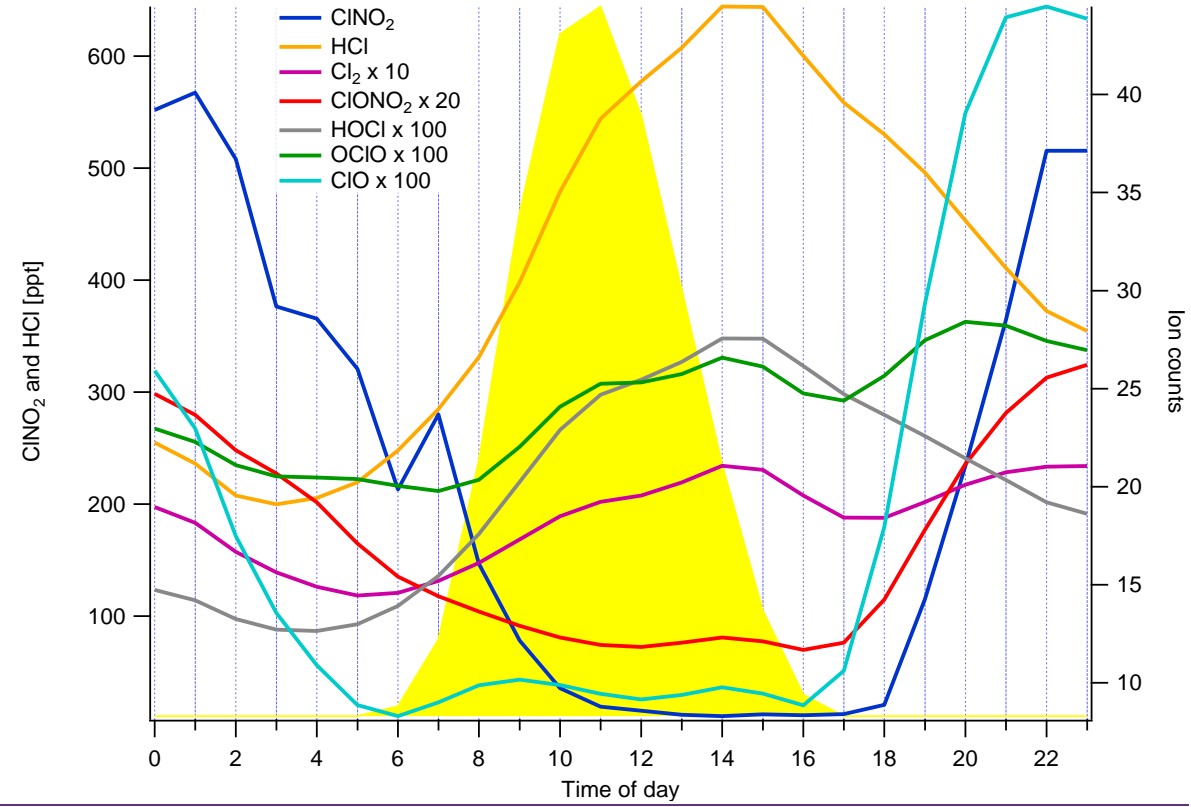

**Figure 3. Mean diurnal profiles of the inorganic halogens detected by the CIMS from the 23rd May to 6th June with average J rate for ClNO₂ as guide for photolysis. ClNO₂ and HCl mixing ratios are on the left y-axis and the other inorganic halogens on the right y-axis displayed in ion counts.**

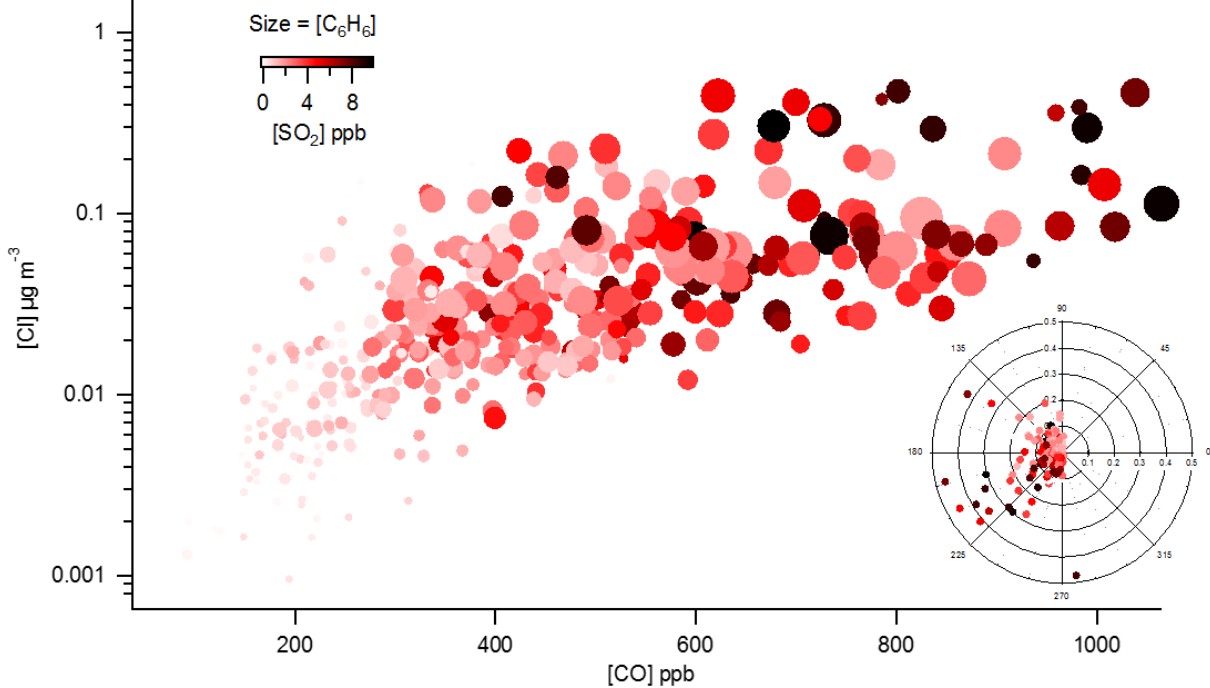

Figure 4. Correlation of particulate Cl⁻ from the AMS measurements and CO colour coded by SO₂ mixing ratio and size binned by increasing benzene mixing ratio. A wind rose plot illustrates the wind direction and particulate Cl⁻ mixing ratio colour coded by SO₂ mixing ratio.

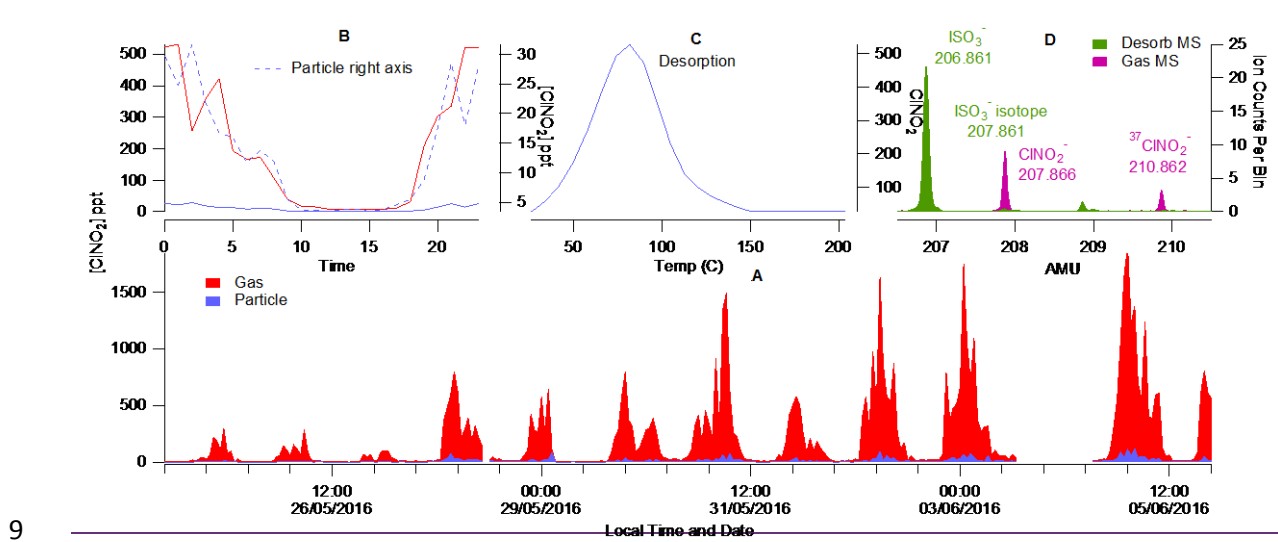

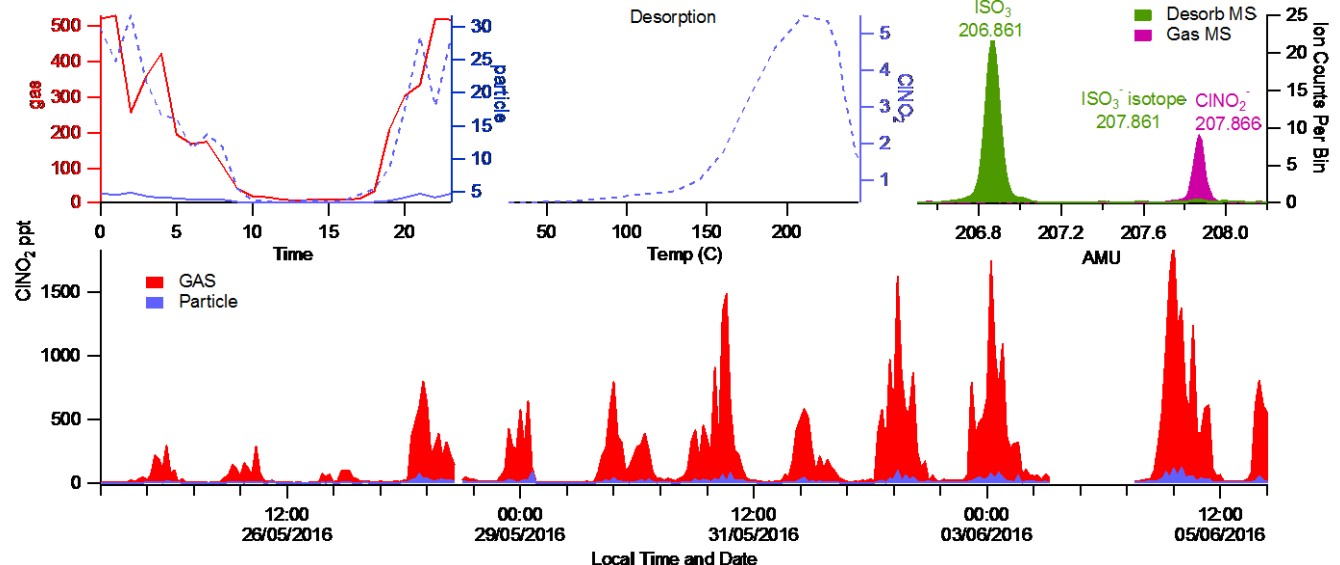

Figure 5. ClNO₂ gas and particle phase campaign time series (1 hour averaged) (panel A) and average diurnal profiles Panel B). The peak fitting for ClNO₂ and the SO₃ interfering mass at 207-208 AMU (panel C) and the desorption profile for the counts attributed to the high resolution ClNO₂ peak (panel D).

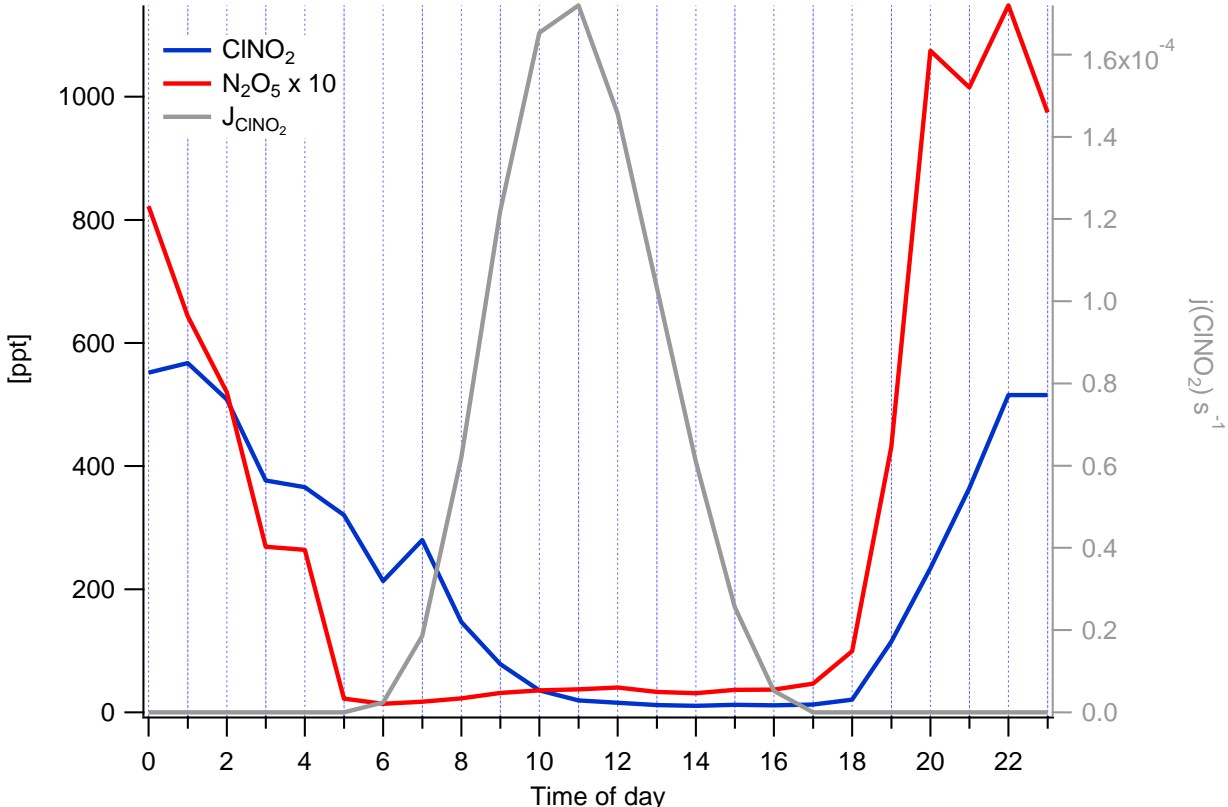

Figure 6. Diurnal profile of N₂O₅, ClNO₂ and j(ClNO₂) for the campaign highlighting the persistence of ClNO₂ passed sunrise and the expected rapid photolysis of N₂O₅.

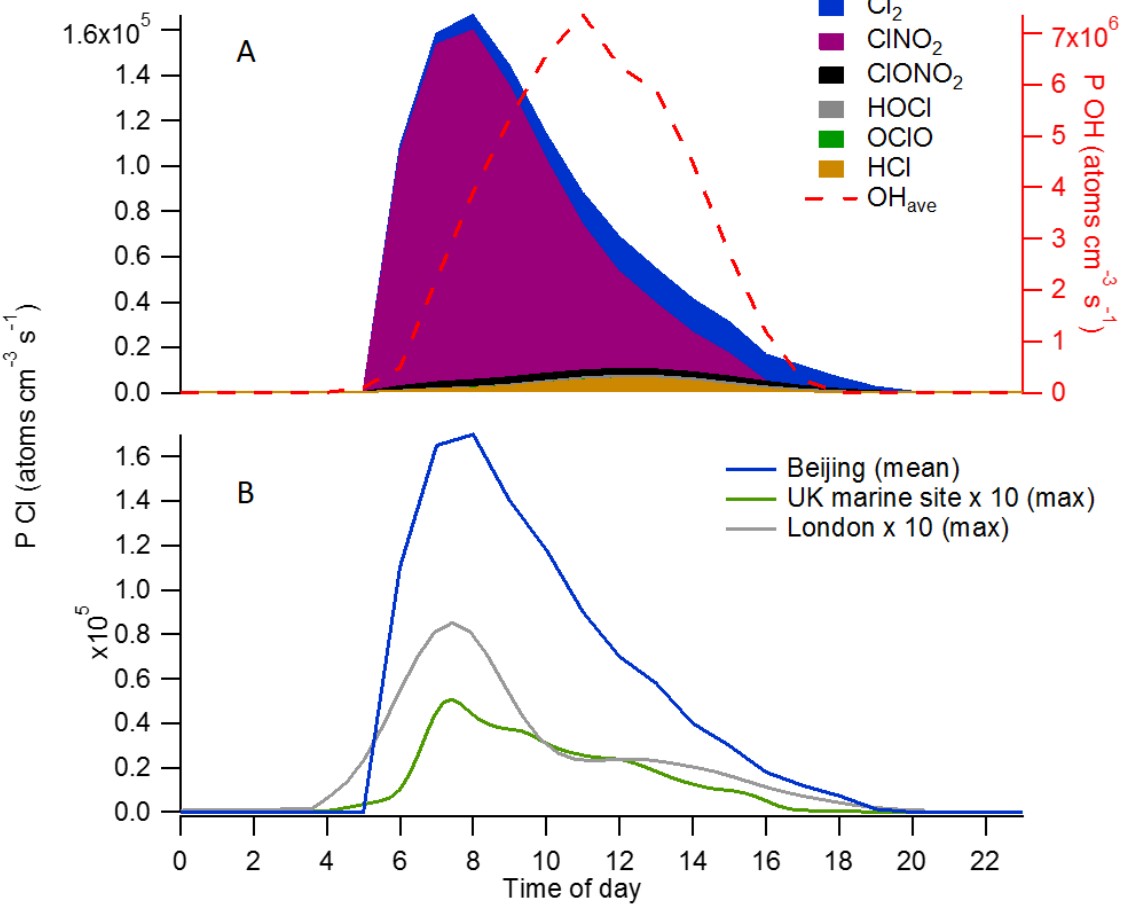

**Figure 7. A) Steady state calculation of inorganic halogens contribution to chlorine atom production. B) Relative mean diurnal profiles of calculated chlorine atom mixing ratio calculation from this work (Beijing) and measurements in the UK (London (Bannan et al., 2015) and a marine site (Weybourne Atmospheric Observatory-Bannan et al., (2017)). The steady state OH production rate from Beijing is also displayed to illustrate relative mixing ratios of oxidants.**

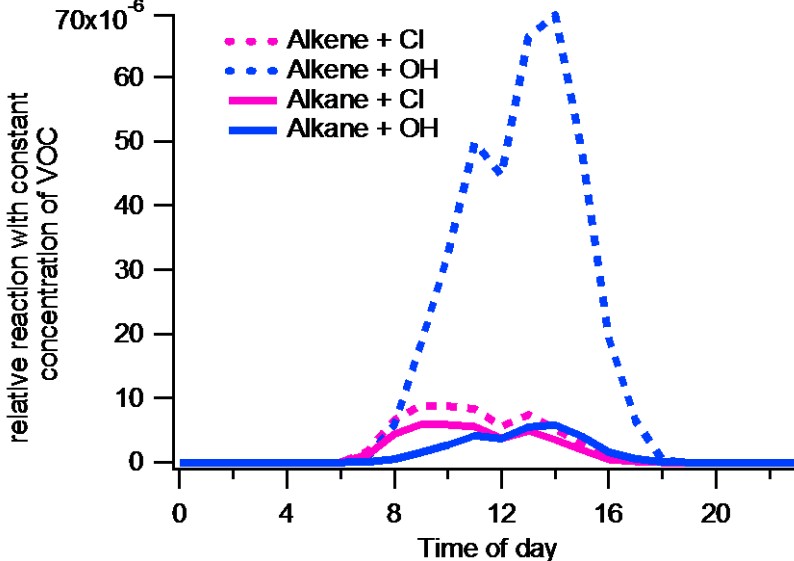

**Figure 8. Mean diurnal time series of alkene (pink) and alkane (blue) relative reaction rate (arbitrary value)**
**with the chlorine atom (dashed) and OH (solid).**

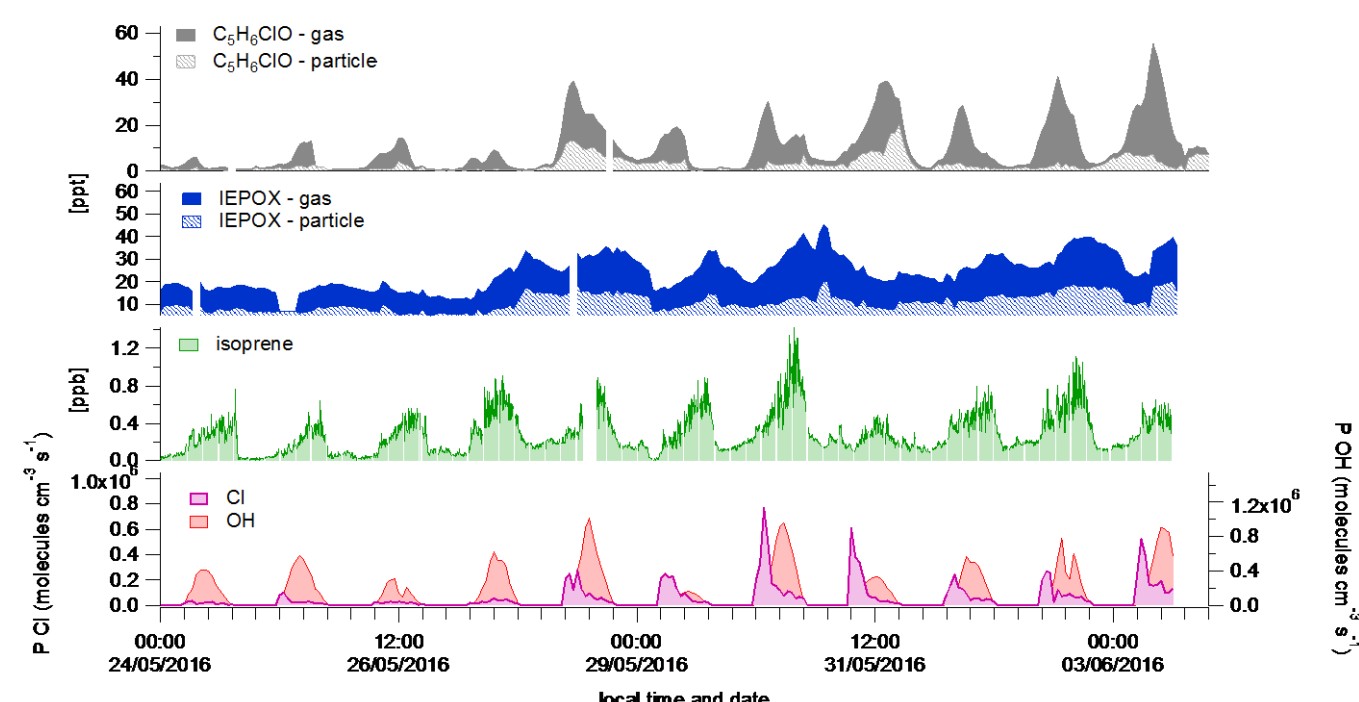

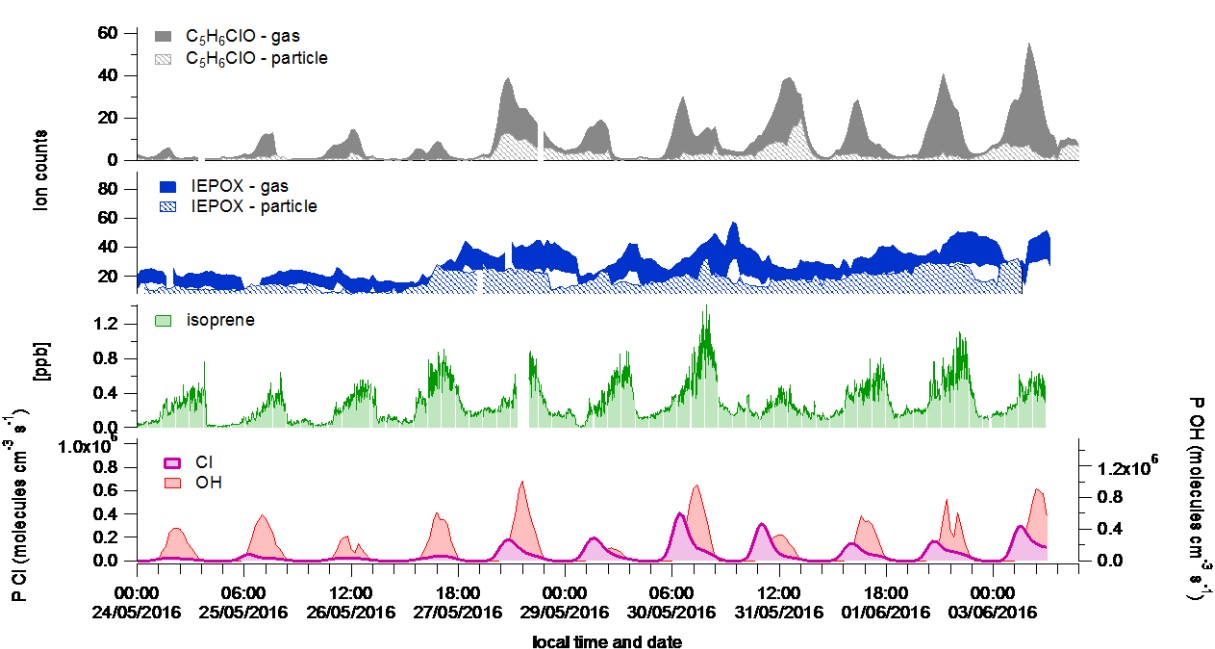

**Figure 9. Campaign time series of isoprene, IEPOX, CMBO and steady state production rate of chlorine**
**atoms and OH**

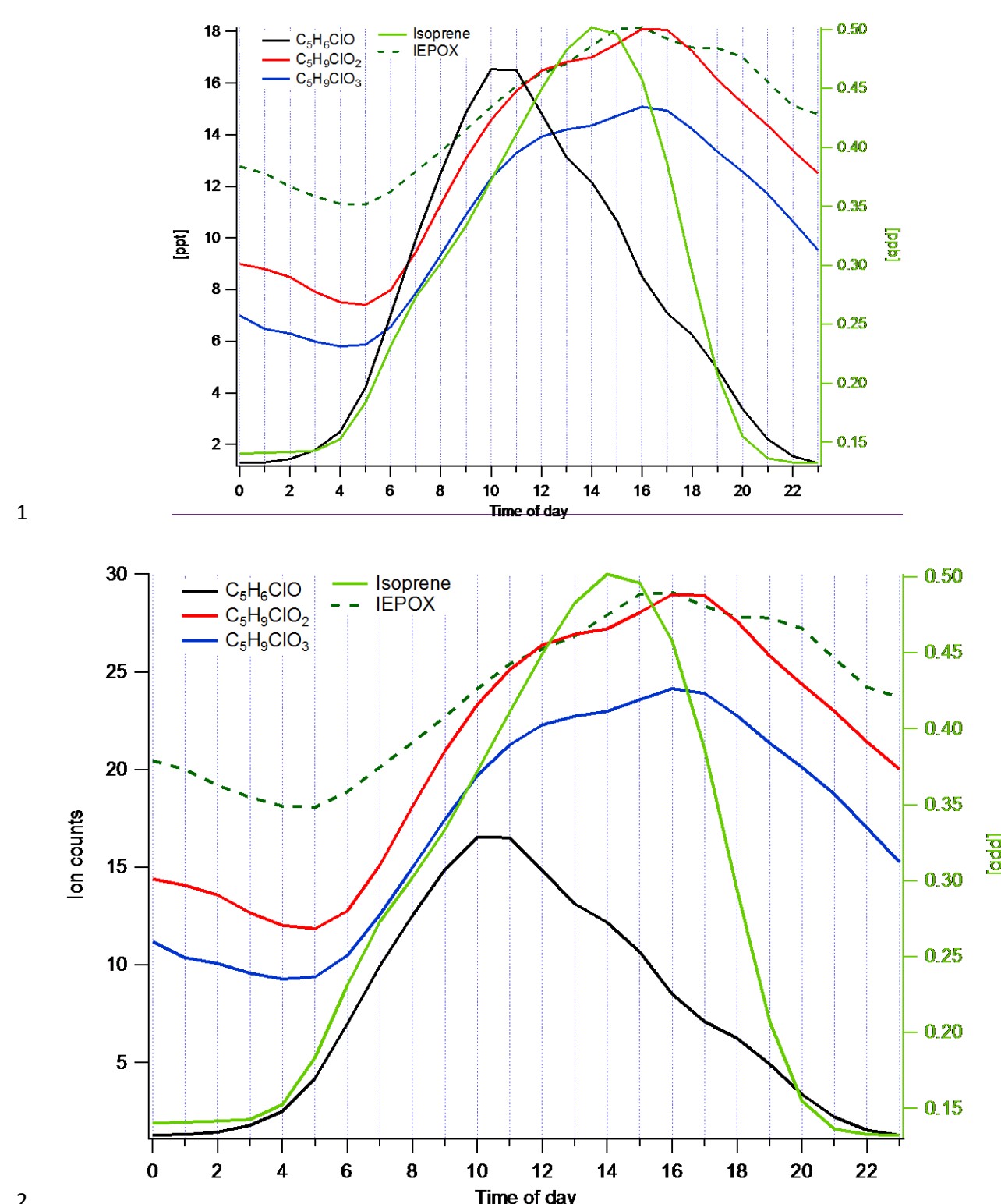

Figure 10. Mean diurnal profiles of isoprene (right y-axis) and its OH oxidation product (IEPOX) and chlorine atom oxidation products CMBO, C₅H₉ClO₂ and C₅H₉ClO₃ (left y-axis

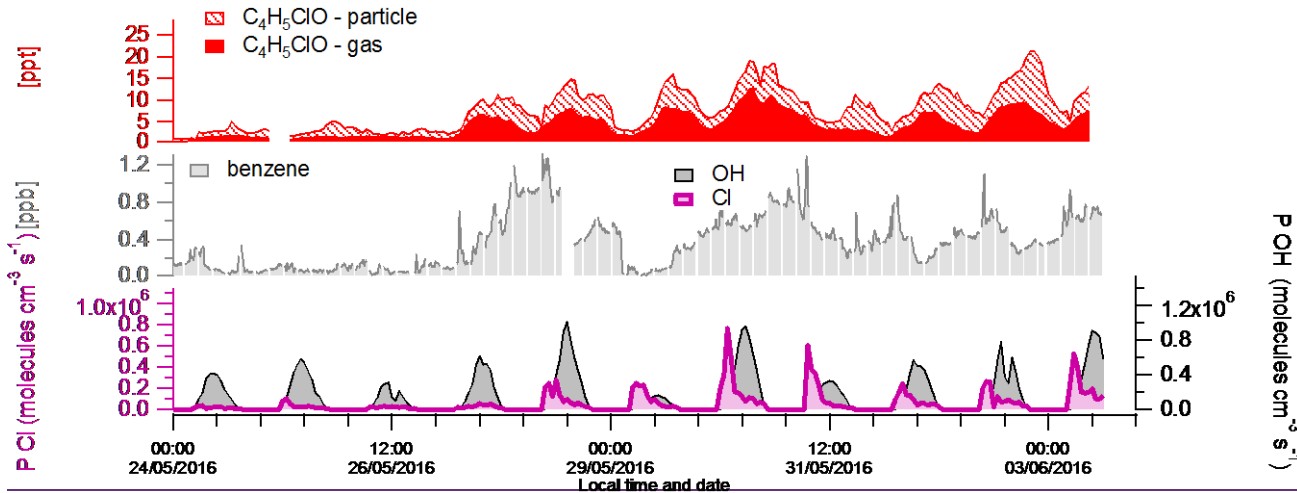

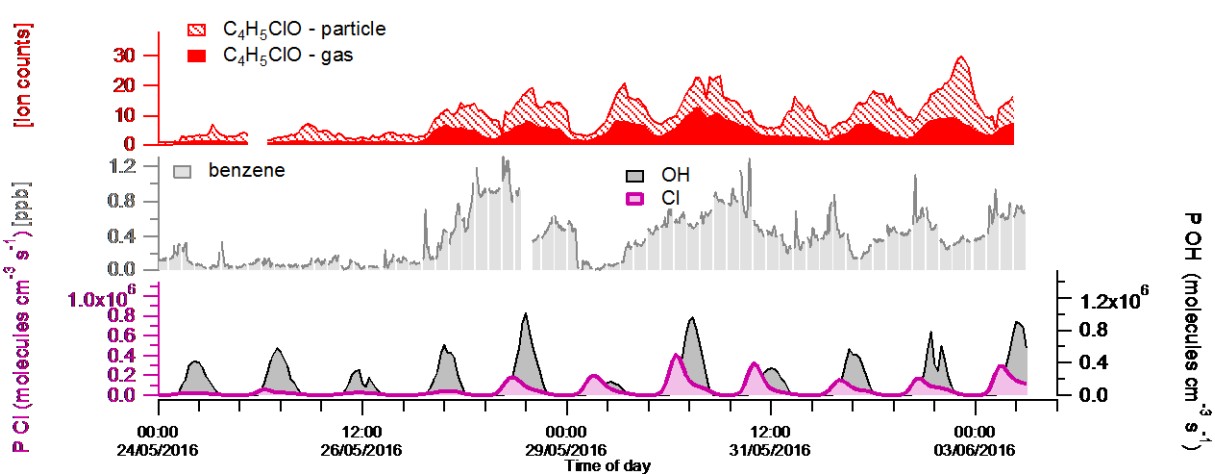

**Figure 11. Campaign time series of benzene and CCA with supporting ~~calcualations~~calculations of OH and the chlorine atom production rates**

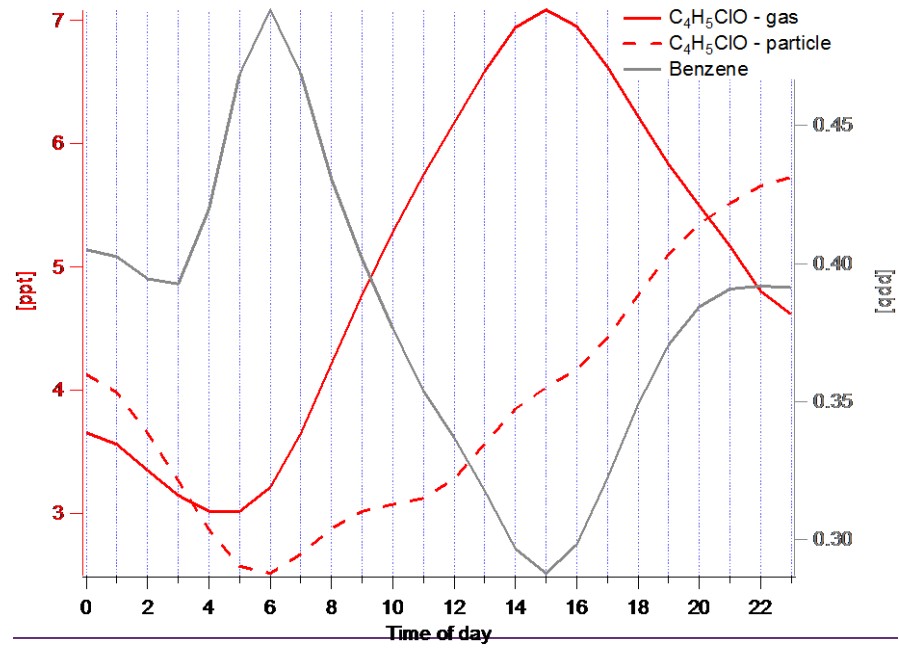

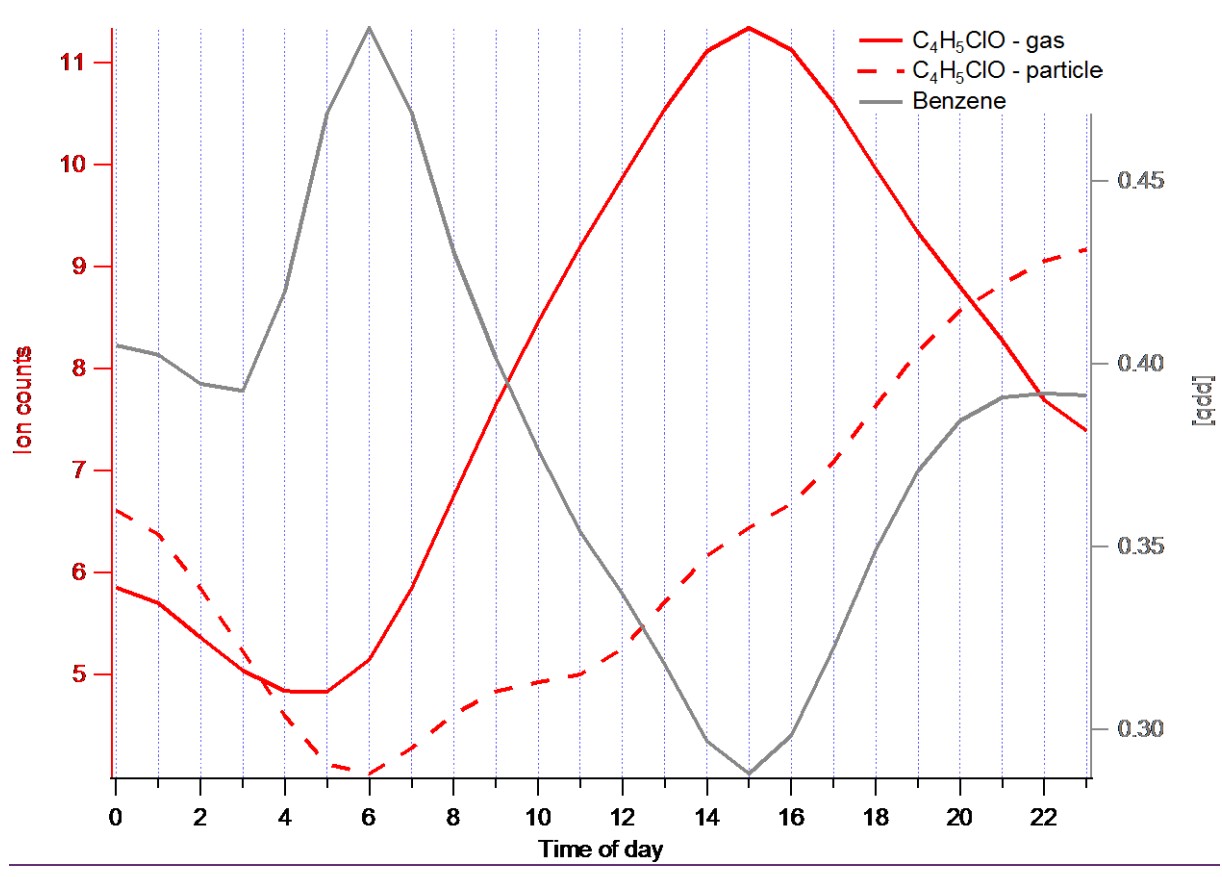

**Figure 12. Mean campaign diurnal profiles of benzene (grey) and CCA in the particle (dashed red) and gas phase (solid red).**