# Peer review of "Chlorine oxidation of VOCs at a semi-rural site in Beijing: Significant chlorine liberation from ClNO₂ and subsequent gas and particle phase Cl-VOC production"

_Atmospheric Chemistry and Physics, 2018_

## Referee Comment (RC1) · Anonymous Referee #1 · 20 Feb 2018

The manuscript by Le Breton et al. describes measurements of reactive chlorine species in the gas and particle phase in Beijing. They use this data to understand the sources of chlorine atoms and constrain the chlorine budget. The manuscript is missing key measurement details that preclude an effective assessment of the quality of the data interpretation. Significant revisions are required before this manuscript can be considered for publication in ACP.

General comments

[Figure]

The authors claim a novel aspect of their findings is the anthropogenic source of reactive chlorine in China (e.g. page 11, line 35). However, this is based on the absence of sodium chloride from an aerosol mass spectrometer (AMS) measurement. The chloride measured by AMS is only non-refractory (i.e. primarily ammonium chloride) and excludes sodium chloride. According to the AMS method cited in this manuscript (Hu et al., 2016), only non-refractory chloride is measured. Discussions including this must be re-considered.

There are important analytical details lacking in the manuscript. Although mixing ratios of HCl, Cl2, ClONO2, HOCl, ClO, and OClO are reported, no information is provided on calibrations for these molecules. These must be included. Calibrations are not reported for any compounds in the particle phase measurements using the FIGAERO inlet. Despite this, and their admission that the observations could be explained by a sampling artifact, a quantification of "particle to gas phase partitioning" for ClNO2 is reported (page 9, lines 26-27). The uncertain nature of this observation is consistent with a statement (page 9, lines 23-24) "this suggests the possible presence of ClNO2 in the particle phase", but inconsistent with a later statement (page 9, lines 30-31) "these data indicates a significant amount of the chlorine associated with ClNO2 is not liberated from the particle phase." In order to report this data in the text and figures, filter spike and recovery tests and gas-phase ClNO2 filter sorption tests must be undertaken. Without this analytical rigor, the data is speculative. Similarly, CMBO in the particle phase is reported in Figure 9 (although not explicitly discussed in the text). Considering these particle observations are listed as a major novel finding of this work, they must be clearly justified.

Throughout the manuscript (see specific examples below), the authors have not properly considered the full literature in their discussion.

The manuscript contains numerous grammatical errors and should be carefully proof-read.
Specific comments

Throughout the manuscript, "mixing ratio" and "concentration" are used interchangeably when discussing gas-phase measurements. All instances of "concentration" should be changed to "mixing ratio" (e.g. page 4, line 3 and page 12, line 26).

Page 2, lines 15-31. The way this is presented, it appears as though the dominant fate of chlorine atoms is reaction with inorganics. In most cases, reactions with organics will be far more important.

Page 4, line 15. IUPAC prefers the term "resolving power" ("resolution" is used to describe another quantity). The m/z must also be defined for the given resolving power. This information should also be reported in Section 2.2.

Page 5, lines 30-34. The sentence starting with "In this calibration..." is repeated.

Page 6, line 20. "Mass" should be "m/z".

Page 6, line 35. Is there any trend in the measurement discrepancies with RH?

Page 6, lines 33-36. These two sentences appear to give different numbers to describe the same results. It is confusing.

Page 9, line 36. Typo in "photolytically"

Page 10, line 1. "Kim et al." is missing from the references section.

Page 10, lines 19-23. Equations 2, 3, 5, and 6 are not balanced.

Page 11, lines 9-12. Budgets of chlorine atoms are also available for Los Angeles (Riedel et al., 2012; Young et al., 2014).

Page 11, lines 23-24. The authors say "a number of studies have deemed chlorine atom chemistry to be insignificant with respect to O3 production and competing VOC oxidation to OH" and cite a single study to justify that a "significantly different approach is needed to assessing oxidation chemistry and photochemical smog in Asia". In fact,

many studies examining this issue globally have shown a demonstrable impact of chlorine atoms on oxidation chemistry (e.g. (Osthoff et al., 2008; Riedel et al., 2014; Sarwar et al., 2014)). This suggests similar techniques can be applied to understand Asian air quality. It is also not clear what the nature of the "significantly different approach" suggested by the authors might be.

Page 11, line 29. The authors mention that steady-state OH was calculated. More details are needed here. What measurements were included in this calculation? How were those measurements made? Page 11, lines 29-33. Calculated chlorine atom to measured OH concentrations are available for Los Angeles (Young et al., 2014).

Page 14, lines 17-20. In the final paragraph of the paper, the authors claim that "chlorine atom chemistry may be under represented within models due to the lack of quantification and identification of particulate Cl-VOC products. This work provides instrumental capability to probe the competition between OH and Cl oxidation chemistry and quantify the SOA yields as a result of both pathways." This paper does not demonstrate quantitative measurement of particulate Cl-VOCs, as no calibrations or desorption efficiencies have been presented or discussed. Furthermore, the authors have not sufficiently shown that particulate Cl-VOCs are necessary to understand the relative impacts of chlorine atoms and OH on air quality.

Figures would be more clear if panels were labeled with (A), (B), etc. Axis labels are often missing or unclear. For example, in Figure 1, presumably all the x-axes should be "m/z" and in Figure 5, the right-hand y-axis of the top left graph is unlabeled.

Page 23, lines 2-6. In the Figure 7 caption, it would be helpful to be explicit that terms are calculated and not measured.

Sources Cited

Hu, W., Hu, M., Hu, W., Jimenez, J. L., Yuan, B., Chen, W., Wang, M., Wu, Y., Chen, C., Wang, Z., Peng, J., Zeng, L. and Shao, M. Journal of Geophysical Research: Atmospheres, 1955–1977, doi:10.1002/2015JD024020, 2016.

Osthoff, H. D., Roberts, J. M., Ravishankara, A. R., Williams, E. J., Lerner, B. M., Sommariva, R., Bates, T. S., Coffman, D., Quinn, P. K., Dibb, J. E., Stark, H., Burkholder, J. B., Talukdar, R. K., Meagher, J., Fehsenfeld, F. C. and Brown, S. S. High levels of nitryl chloride in the polluted subtropical marine boundary layer, Nat. Geosci., 1(5), 324–328, 2008.

Riedel, T. P., Bertram, T. H., Crisp, T. A., Williams, E. J., Lerner, B. M., Vlasenko, A., Li, S.-M., Gilman, J., de Gouw, J., Bon, D. M., Wagner, N. L., Brown, S. S. and Thornton, J. A. Nitryl chloride and molecular chlorine in the coastal marine boundary layer, Environ. Sci. Technol., 46, 10463–10470, 2012.

Riedel, T. P., Wolfe, G. M., Danas, K. T., Gilman, J. B., Kuster, W. C., Bon, D. M., Vlasenko, A., Williams, E. J., Lerner, B. M., Veres, P. R., Roberts, J. M., Holloway, J. S., Lefer, B., Brown, S. S. and Thornton, J. A. An mcm modeling study of nitryl chloride (ClNO2) impacts on oxidation, ozone production and nitrogen oxide partitioning in polluted continental outflow, Atmos. Chem. Phys., 14(8), 3789–3800, doi:10.5194/acp-14-3789-2014, 2014.

Sarwar, G., Simon, H., Xing, J. and Mathur, R. Importance of tropospheric ClNO2 chemistry across the Northern Hemisphere, Geophys. Res. Lett., 41(11), 4050–4058, doi:10.1002/2014GL059962, 2014.

Young, C. J., Washenfelder, R. A., Edwards, P. M., Parrish, D. D., Gilman, J. B., Kuster, W. C., Mielke, L. H., Osthoff, H. D., Tsai, C., Pikelnaya, O., Stutz, J., Veres, P. R., Roberts, J. M., Griffith, S., Dusanter, S., Stevens, P. S., Flynn, J., Grossberg, N., Lefer, B., Holloway, J. S., Peischl, J., Ryerson, T. B., Atlas, E. L., Blake, D. R. and Brown, S. S. Chlorine as a primary radical: evaluation of methods to understand its role in initiation of oxidative cycles, Atmos. Chem. Phys., 14, 3427–3440, 2014.

---

## Referee Comment (RC2) · Anonymous Referee #2 · 2 Mar 2018

Le Breton and coworkers present data acquired using a commercial instrument, an iodide FIGAERO-ToF-CIMS, collected 40 km NW of Beijing in the summer of 2016. Mixing ratios of N2O5 and ClNO2 were quantified and a large number of halogenated molecules, some of which oxidation products of VOCs, were identified for the first time. The authors claim to have quantified particle phase ClNO2. Overall, the manuscript contains results that will be of interest to the broader atmospheric chemistry community and can probably be published after my comments below have been addressed.

Major comments

[Figure]

- The manuscript suffers from organizational issues. Reactions are not consecutively numbered, sections were skipped, etc.

- Mixing ratios of a variety of trace gases, including HCl, Cl2, ClONO2, HOCl, OClO and ClO as well as CMBO, isoprene, IOPEX, and benzene as well photolysis frequencies are presented but it is unclear in many cases how these data were acquired or how instrumental response factors were determined.

- Furthermore, concentrations of Cl and OH were calculated on the basis of steady state assumptions. These calculations are questionable since the only VOC measurements were by PTR-MS, an instrument that quantifies many but not all VOCs. Crucially, a PTR-MS usually does not quantify alkanes, whose abundances are important sinks for Cl atoms.

- Some data (e.g., OClO, CMBO, ClO, ClONO2, IOPEX) are only semiquantitative and should be presented as such.

Specific comments

pg 1 line 27 –replace the comma with "and"

line 29 – ppt is not a concentration unit – please rephrase

pg 2 lines 17 – (O3, HOx, and NOx levels via ... (R1-R9)). Most of the Cl will likely abstract hydrogens from hydrocarbons (R11), in particular at this site. Another important reaction omitted here is OH+HCl->H2O+Cl. Consider reorganizing the introduction to reflect this.

pg 3 lines 2-4. –"This perturbation is currently thought to only be significant in the early hours of the day while OH concentrations are low and chlorine atom production is high through the photolysis of ClNO2."

I don't think this is correct. Reaction of Cl with alkanes produces peroxy radicals, which feed into the "regular" HOx/NOx cycles. Thus, the early morning injection of radicals

impacts (perturbs) radical chemistry for the remainder of the day. Perhaps the authors meant to say "Oxidation of VOCs by Cl is currently thought to be ..."?

lines 11/12 – there are two reactions labeled R11

line 31 – "36%" - please add the value for SOA yield from OH initiated oxidation of isoprene for comparison.

pg 4 line 24 – how were photolysis rates determined?

line 31 – there are two Le Breton et al. 2017 references. Please label them 2017a and 2017b.

pg 5 lines 1-2 Please provide more detail as to how the PTR-MS was calibrated, what molecules were quantified, etc.

line 17 - section 2.3 is absent.

Please describe how the response factors for HCl, Cl2, ClONO2, HOCl, OClO and ClO (Figure 3) were determined

lines 24-25 – "The N2O5 diffusion source was held at a constant temperature (-23 C), and the mass loss rate was characterized gravimetrically for a flow rate of 100 sccm."

N2O5 is quite hygroscopic, such that the diffusion source could "gain weight" simply by absorbing residual moisture. Another potential error with this method is "loss" of NO3 (e.g., through reaction with impurities on the wall) followed by loss of NO2 (toward which the CIMS is probably blind). All this probably doesn't matter since there was a CEAS on site.

How stable/accurate/reproducible is this source? Could it be used as standalone N2O5 calibration method?

Please state if the diffusion source method has been verified using CEAS (which I assume it has).

Line 36 – "these sensitivities" – please state the instrumental response factors here.

pg 6 line 20/21 –"A quadrupole CIMS may not be able to resolve the peak adjacent to ClO at mass 178 and the second dominant peak for the ClNO2 fit would result in a 10% over estimation."

Please clearly state what ions are present at mass 178.

It is unclear what is meant by "second dominant peak for the ClNO2 fit" – is this at m/z 208?

line 32 – please put the N2O5 mixing ratios in context – (temperature, O3 and NO2 levels, NO3 production rate etc.)

line 33 – Were the instruments operated on the same inlet? If not, there may be scatter simply from sampling air at slightly different locations.

line 36 – The offset should have units of ppt

pg 7 line 2 – "although averaging at 4 ppt" perhaps better to give a relative error here

line 3 – please move details on how instruments were operated (heated IMR) to section 2.2

line 21- "Inorganic chlorine abundance and profiles". There is a lot presented in this section, BB, WRF etc, that goes well beyond inorganic chlorine abundances and profiles. This section should be broken up into smaller, more coherent pieces.

line 24 – mixing ratio, not concentration

line 25 – "$\sigma$ 270 ppt" is this standard deviation?

lines 33 – pg 8 line 9. Please comment on the possibility of chlorine nitrate forming on the inner walls of the inlet.

pg 8 line19. –" This suggests the chlorine has an anthropogenic source and not marine" I disagree.

One has to be careful with the interpretation of AMS data. The "standard" AMS chloride product only includes non-refractory aerosol, i.e., does not include sea salt chloride – for one, it does a poor job volatilizing NaCl, and most AMS have a size cut off of 1 micron that filters out most of the larger sea salt aerosol particles. The correlation of AMS chloride with anthropogenic tracers may arise from acid displacement of sea salt chloride in polluted air (that is high in SO2). I'd suggest rewording the entire paragraph (lines 14-27). I don't doubt that anthropogenic Cl sources contribute, but there aren't enough data (shown in this paper) to proof a negligible marine influence.

line 28 - Please describe the WRF model in the methods section, not in the results section.

I'd remove the WRF simulations as they may not account for local BB – chemical tracers would be more robust.

pg 9 line 13. All that is shown is that WRF modeling suggests BB to be a small source of chlorine – it doesn't show industrial emissions. Please rephrase.

line 14 The particle desorption profiles should be discussed in their own section.

lines 14-20. Did you observe the peak at 210? Please expand the AMU axis in Figure 5 to show it.

line 30 " these data indicates a significant amount of the chlorine associated with ClNO2 is not liberated from the particle phase" it should be "these data indicate"

More to the point, you observe that you can drive off ClNO2 if you heat aerosol. Have you considered that additional ClNO2 could be formed by thermally driven reactions? If not, please state that this is a major assumption made here.

line 31-33 "The slope" please show this plot (perhaps as an insert in Figure 5).

Personally, I wouldn't call 5% "significant" considering this is much less than the measurement (calibration) error.

pg 10 line 18 The numbering of the reactions is inconsistent with those on pg 2. Some reactions are unnecessarily duplicated.

line 25 - Please number the steady state expression.

A major source of Cl atom is the reaction OH + HCl -> H2O + Cl, which should not be omitted here. line 26 - And how was "equivalent CH4" determined for this site? It must be massive.

It is very likely that the PTR-MS misses most of it, for example all of the alkanes (Table 2 of de Gouw's Mass Spectrometry Reviews 26, 223 (2007)).

line 32 – How can HCl become a dominant source of Cl atom if it's not part of the steady state expression?

pg 11 line 9 – "The results show that both at the UK marine and urban site max chlorine atom concentrations are more than an order of magnitude lower than the mean of Beijing." Considering the uncertainty of the Cl atom sinks, the authors should only compare Cl atom production rates. Comparing rural and urban sites (Weybourne with Beijing) is like comparing apples and oranges. Many other groups have calculated Cl atom production rates from ClNO2 photolysis, including many polluted urban sites. How do the numbers of this study stack up to these?

line 29 – " Steady state calculations of OH (as described by Whalley et al., 2010)"

Such calculations require comprehensive knowledge of the VOCs, CO, NOx, etc. present., more than is provided by a PTR-MS (whose list of VOCs monitored is not comprehensive).

Imo, the entire section comparing OH and Cl abundances is questionable.

pg 12 line 30 – "longer atmospheric lifetime" how long are the lifetimes of CMBO and of isoprene?

line 34 – "The concentrations of Cl and isoprene were relatively low" How low is relatively low? Please be quantitative.

If CMBO abundances did not follow those of its precursors, does that imply that CMBO can be primary (or originates from other precursors)?

How certain are we that CMBO is a unique marker of chlorine-isoprene chemistry (line 25)?

pg 13 line 36 " ClNO2 was potentially identified in the particle phase "

I agree, but in the preceding text, ClNO2 was not only identified but also quantified, or was it? Either way, the earlier section is inconsistent with the much more conservative conclusion in the end.

pg 14 – many references are incomplete (e.g., Pszenny et al.)  and most are missing their doi.

pg 19 – Figure 1.  Please identify the green, gold/yellow, and magenta lines.  For the second panel, it would be useful to show a blank (zero) measurement also.

pg 20 – please define the "C" and "M" terms

pg 23 – Figure 7A or 7B – one of the "y" axes is mislabeled.

---

## Author Comment (AC1) · 30 Apr 2018

Response to reviewers on "Chlorine oxidation of VOCs at a semi-rural site in Beijing: Significant chlorine liberation from ClNO2 and subsequent gas and particle phase Cl-VOC production" by Michael Le Breton et al.

Reviewer 1 The authors claim a novel aspect of their findings is the anthropogenic source of reactive chlorine in China (e.g. page 11, line 35). However, this is based on the absence of sodium chloride from an aerosol mass spectrometer (AMS) measurement. The chloride measured by AMS is only non-refractory (i.e. primarily ammonium chloride) and excludes sodium chloride. According to the AMS method cited in this manuscript (Hu et al., 2016), only non-refractory chloride is measured. Discussions including this must be re-considered.

Response: We firstly agree that there are limitations on the AMS measurements of particulate chloride due to the refractory component which the AMS will not measure, although there have been several further steps in the manuscript to probe the anthropogenic source of chloride observed, such as the correlation with SO2, benzene and CO. Furthermore, the large distance (200k m) of the site from the coast makes it difficult to transport the seasalt to the site. We have now performed a wind rose analysis in more detail to further probe the source of chloride, whose results are shown in figure S1, in which radial and tangential axes represent the wind direction and speed (km h-1) and the colour bar represents the PM2.5 concentration. We see during the campaign, the severe pollution was from the south and southwest, with little contribution from the east part. Therefore, we could deduce that little contribution of the chloride was from the ocean. In addition, we have now done WRF model simulation to observe how a sea salt model results would correlate with ClNO2 profiles. This resulted in a poor correlation (figure S2) whereas total CO, as stated in the manuscript, had a relatively good correlation with ClNO2, further supporting the anthropogenic source of chloride. Furthermore, the modelled seasalt levels are very low, most likely unable to produce the mixing ratios of ClNO2 observed by the CIMS. Figure S3 has been added to the supplementary.

Action: The following text has been added/amended to the model results analysis and acknowledges the AMS limitations and instead highlighting the further support to anthropogenic Cl from low levels of sea salt and correlation within the model analysis. "The high levels of ClNO2 indicate a local significant source of chlorine to support these observations. The dominant source of chlorine atoms for ClNO2 production within models, such as the Master Chemical Mechanism (MCM), is from sea salt. However, the site is situated 200 km from the Yellow Sea and therefore this origin would have a low probability . The mean AMS chloride mass loading was 0.05 $\mu$g m-3 for the campaign with a maximum of 1.7 $\mu$g m-3. The Cl- from the AMS appears to be correlated strongly with CO and SO2, possibly originating from power plants or combustion sources. It should be noted that the AMS data does not include refractory aerosol and also has a cut off size larger than anticipate size of sea salt particles. Instead, the high Cl- observed appears to originate from mainland areas to the site (Figure 4) rather from the nearest coast, further supporting a strong anthropogenic source. Tham et al., (2016) observed a strong correlation of aerosol chloride with SO2 and potassium from measurements done during the same season in 2014 at Wangdu (semi-rural site 160 km south West of Beijing) and suggested contribution to fine chloride from burning of coal and crop residues. The latter was also supported by satellite fire spot count data (Tham et al., 2016). Riedel et al. (2013) have previously reported high ClNO2 mixing ratios observed from urban and power plant plumes measuring high mixing ratios of gas phase Cl2. The correlation with SO2 indicates coal burning as a potential source of particulate chlorine which is known to be a significant source of PM in the Beijing region (Ma et al., 2017), and the correlation with CO and benzene could be an indicator of biomass burning (Wang et al., 2002). To support this analysis, figure S1 displays a wind rose plot in which radial and tangential axes represent the wind direction and speed (km h-1). The colour bar represents the PM2.5 concentration. We could see that during the campaign, the severe pollution was from the south and southwest, with little contribution from the east part. Therefore, we could deduce that little contribution of the chloride was from the ocean. In order to test the hypothesis of biomass burning as a source of particulate chlorine, biomass burning emissions and transport utilising the EMEP MSC-W chemical transport model driven by meteorology from the WRF-ARW model (Skamarock et al., 2008) were used. Neither of the two biomass burning databases used (FINN and GFAS) contained data on chlorine emissions, so instead the biomass burning emissions of CO (CObb) were tracked and compared to the total mixing ratio of CO (COt) at the Changping site. CO was chosen since the

measurements at Changping had shown strong correlation between CO and ClNO2 and because CO could be expected to be co-emitted with chlorine for both biomass burning and industrial combustion. Figure S2 (supplementary) shows time series of the measured ClNO2 mixing ratios at the Changping site, as well as the modelled mixing ratios of COt and CObb. CObb is shown for calculations using either the FINN or the GFAS data base, while for clarity the COt is only shown using the FINN data base. From this figure it is clear that mixing ratios of CObb are very low compared to COt. The two pollution episodes on May 18-May 23 and May 28-June 5, are to some extent visible in all time series, but for the biomass burning CO series, the second episode is much less pronounced. Night-time averages of the mixing ratios shown in figure S2 were calculated for each night for the time period 18:00 to 08:00 local time (UTC+8), roughly corresponding to the period when ClNO2 is not destroyed by photolysis. Nights with significant amount of missing data for the measurements were excluded. Figure S2 shows scatter plots of these averages of ClNO2 against the averages of the other species including their linear fits. The R2 for these fits were 0.48, 0.04, and 0.21 for COt, CObb FINN, and CObb GFAS respectively. The fact that mixing ratios of CObb is so much smaller than COt according to the model, combined with the much better correlation for COt than for CObb strongly suggests that industrial emissions are the dominant source of chlorine, rather than biomass burning. To further investigate the source of chloride, the model was also run to calculate sea salt levels instead of CO. This resulted in a poor correlation between sea salt and the ClNO2 (figure S4). The absolute levels of sea salt calculated by the model were also very low, unlikely to be able to produce the observed mixing ratios of ClNO2 as observed by CIMS."

There are important analytical details lacking in the manuscript. Although mixing ratios of HCl, Cl2, ClONO2, HOCl, ClO, and OClO are reported, no information is provided on calibrations for these molecules. These must be included. Calibrations are not reported for any compounds in the particle phase measurements using the FIGAERO inlet. Despite this, and their admission that the observations could be explained by a sampling artifact, a quantification of "particle to gas phase partitioning" for ClNO2 is

reported (page 9, lines 26-27). The uncertain nature of this observation is consistent with a statement (page 9, lines 23-24) "this suggests the possible presence of ClNO2 in the particle phase", but inconsistent with a later statement (page 9, lines 30-31) "these data indicates a significant amount of the chlorine associated with ClNO2 is not liberated from the particle phase." In order to report this data in the text and figures, filter spike and recovery tests and gas-phase ClNO2 filter sorption tests must be undertaken. Without this analytical rigor, the data is speculative. Similarly, CMBO in the particle phase is reported in Figure 9 (although not explicitly discussed in the text). Considering these particle observations are listed as a major novel finding of this work, they must be clearly justified.

Response: The inorganic halogens, many of which has no stable production method, are attributed the same sensitivity as of ClNO2 which is a commonly used method. For example, Le Breton et al 2017 showed that inorganic halides have a similar sensitivity. Furthermore, the comparable sensitivity for chloroacetic acid and ClNO2 emphasis a similar sensitivity for chloride containing species when applying the iodide ionisation. As specific sensitivity for compounds being evaporated and subsequently ionised in the gas-phase after entering the same IMR would be redundant. However, one may argue that the efficiency of desorption could be quantified. Here the particle phase ClNO2 is attributed to a possible complex within the ambient aerosol. We claim here that we observe higher concentrations in the particle phase compared to the Henrys Law constant which would suggests that the ClNO2 is trapped within a matrix, rather than unable to partition upon heating, i.e. our conclusion would not be effected by a less than unity effectivity of desorption. This would be the same for CMBO, which we quantify using the sensitivity for chloroacetic acid in the gas phase. This kind of calibration procedures has been performed in previous CIMS work.

Action: A clearer explanation of our application on sensitivities has been added to the manuscript and a validation to the approach with relevant citations. It is now stated that the gas phase sensitivities applies to the particle phase data also. The text regarding particle phase ClNO2 has been made more clear to state that we observe higher concentrations in the particle phase due to a matrix effect rather than limitation of liberation from the particle phase during desorption. CMBO is also described to be quantified using the sensitivity for chloracetic acid which applies for both particle and gas phase. "These sensitivities for N2O5 and ClNO2 (9.8 and 1.6 ion counts per ppt Hz-1 for 1x105 iodide ion counts) were applied relatively to that of formic acid. The other inorganic halogens reported in this work were given the same sensitivity as of ClNO2, as Le Breton et al. (2017) reported many inorganic halogens possess a similar, if not the same, sensitivity. This was further supported by our chloroacetic acid calibration. Other acids identified by CIMS which are reported in the literature are given the sensitivity of N2O5 to provide a minimum concentration so no concentrations are over estimated. A post campaign calibration of chloroacetatic acid (99%, Sigma Aldrich) was utilised to apply a sensitivity factor for all Cl-VOCs measured during the campaign. The calibration was performed using the same method as for formic acid and gave a sensitivity of 1.02 ion counts ppt-1 Hz when normalized to 1x105 I- ion counts. Using relative sensitivities will increase the uncertainties, but is a commonly applied method within the CIMS community, although in this specific case it is very likely that the sensitivity is similar for all inorganic/organic halogens, as demonstrated by Le Breton et al. (2017a)."

Throughout the manuscript (see specific examples below), the authors have not properly considered the full literature in their discussion.

Response: The manuscript has been addressed as proposed and the literature discussion has been expanded

Action: The manuscript has been addressed as proposed and the literature discussion has been expanded

The manuscript contains numerous grammatical errors and should be carefully proofread. Response: The manuscript has now been "cleaned" of grammatical and typographical errors to make it more clear and concise

Action: The manuscript has now been "cleaned" of grammatical and typographical errors to make it more clear and concise

Throughout the manuscript, "mixing ratio" and "concentration" are used interchangeably when discussing gas-phase measurements. All instances of "concentration" should be changed to "mixing ratio" (e.g. page 4, line 3 and page 12, line 26).

Response: This has now been corrected

Action: Concentration has been replaced with "mixing ratio" where necessary

Page 2, lines 15-31. The way this is presented, it appears as though the dominant fate of chlorine atoms is reaction with inorganics. In most cases, reactions with organics will be far more important.

Response: This should be rewritten so the reaction pathways are presented correctly

Action: The following text has been written to clearly state that the dominant loss of the chloride atom is via reaction with VOCs "The liberated chlorine will predominantly react with VOCs, with the above pathways representing alternative routes to loss of the chloride atom, and contribute to daytime photochemical oxidation, competing with OH and perturbing standard organic peroxy radical abundance ($ROx = OH + HO_2 + RO_2$), $O_3$ production rate, $NO_x$ lifetime and partitioning between reactive forms of nitrogen (Riedel et al., 2014)."

Page 4, line 15. IUPAC prefers the term "resolving power" ("resolution" is used to describe another quantity). The m/z must also be defined for the given resolving power. This information should also be reported in Section 2.2.

Response: Resolution has been changed to resolving power and is now reported in section 2.2.

Action: Resolution has been changed to resolving power and is now reported in section

2.2.

Page 5, lines 30-34. The sentence starting with "In this calibration. . ." is repeated

Response: This is correct and should not be repeated

Action: This repetition has now been removed

Page 6, line 20. "Mass" should be "m/z".

Response: This is correct and will be replaced

Action: "Mass" is now "m/z"

Page 6, line 35. Is there any trend in the measurement discrepancies with RH?

Response: There is no trend observed with RH

Action: This observation has been noted in the text

Page 6, lines 33-36. These two sentences appear to give different numbers to describe the same results. It is confusing.

Response: The gradient of the fit is described in the second sentence whereas the R correlation is described in the first

Action: no changes have been made

Page 9, line 36. Typo in "photolytically"

Response: This typographic error is to be changed

Action: It is now spelt "photolytically"

Page 10, line 1. "Kim et al." is missing from the references section.

Response: Kim et al must be added to the References

Action: This reference has been added
Page 10, lines 19-23. Equations 2, 3, 5, and 6 are not balanced.

Response: This is correct and should be amended

Action: The reactions are now balanced

Page 11, lines 9-12. Budgets of chlorine atoms are also available for Los Angeles (Riedel et al., 2012; Young et al., 2014).

Response: These two manuscripts will be discussed in the text Action: These manuscripts have been cited and referenced and it is acknowledged that measurements in urban Los Angeles yield similar production rates of the chloride radical to Beijing, although it is noted that higher concentrations of HCl observed in these measurements contribute significantly to the radical production where as in this paper we see little HCl contribution

Page 11, lines 23-24. The authors say "a number of studies have deemed chlorine atom chemistry to be insignificant with respect to O3 production and competing VOC oxidation to OH" and cite a single study to justify that a "significantly different approach is needed to assessing oxidation chemistry and photochemical smog in Asia". In fact, many studies examining this issue globally have shown a demonstrable impact of chlorine atoms on oxidation chemistry (e.g. (Osthoff et al., 2008; Riedel et al., 2014; Sarwar et al., 2014)). This suggests similar techniques can be applied to understand Asian air quality. It is also not clear what the nature of the "significantly different approach" suggested by the authors might be.

Response: We agree with the reviewer that this section is to be rephrased and the conclusion is not fully supported by the text

Action: The text now reads "Although several studies have demonstrated a non-negligible impact of chlorine oxidation chemistry (e.g. Oshoff et al., 2008, Riedel et al., 2014 and Sarwar et al., 2014), the impact of Cl chemistry varies significantly between various areas and atmospheric conditions, e.g. Bannan et al., 2015, 2017 deemed the

impact from chlorine atom chemistry to be relatively low with respect to O3 production and competing with OH radicals for VOC oxidation"

Page 11, line 29. The authors mention that steady-state OH was calculated. More details are needed here. What measurements were included in this calculation? How were those measurements made? Page 11, lines 29-33. Calculated chlorine atom to measured OH concentrations are available for Los Angeles (Young et al., 2014).

Response: Reviewer 2 also had some comments about the steady state calculations, to which will also be addressed here to provide a complete picture to the reviewer of how this section has been amended and improved. The reviewers are correct that the reaction of OH+ HCl was accidentally omitted from the text in the original manuscript. The section has now been corrected for that omission. The reviewers are also correct that only using the small subset of hydrocarbons measured by the PTRMS would result in significant errors in estimated steady state Cl atom concentrations. However, we did not just use PTRMS measurements. In previous work (Bannan et al., 2015) we have shown that it is possible to calculate Cl atom concentrations, using a simple steady-state expression with the Cl atom production rate estimated from the observed loss rate of ClNO2 and removal of Cl atoms via reaction with the VOC concentrations supplemented with data from the Boston tailpipe study (AQIRP, 1995) and LA VOC study (Fraser et al., 1997), i.e. missing VOC concentrations are estimated simply by using the ratio of measured VOCs and missing VOCs from these urban studies and measured VOC data in this study. The removal of Cl atoms via reaction with VOCs can then be determined using Eq. 1-3, and NIST kinetic data (Manion et al., 2014) (Eq 1) (Eq 2) (Eq 3) Here the simple steady state approach agreed well with the MCM, despite a much more simplistic approach. We used an identical approach in this work utilizing the tailpipe VOC concentrations (AQIRP, 1995). However, we further simplify the approach by using one term CH4 equivalent which accounts for relative concentration and reactivity towards Cl, i.e. if a VOC reacts 1000 times faster it is the equivalent of 1000 CH4 or more formally for each VOC its CH4 equivalent is kCl + VOC [VOC]/ kCl +

CH4. Whilst the approach is a simplification it has been shown that using these emissions it is possible to estimate the Cl atom production in a Megacity environment and produces results that are comparable with much more thorough modelling approach using e.g. the MCM. It also generates a metric, CH4 equivalent, which can be used as a comparative measurement from city to city. Photolysis rates were measured by a spectradiometer for O3, NO2, HCHO, HONO and H2O2. The photolysis rate of any given species was calculated by normalizing to the cross section and quantum yields taken from the recommendations of the Jet Propulsion Laboratory (JPL) kinetic evaluation report (Burkholder et al., 2015). The reviewer is correct that Cl atoms will produce peroxy radicals which can perturb HOx/NOx cycles, and we were only considering the initial oxidation of VOCs. The text has been corrected to reflect that.

Action: the following text has been added to the manuscript and text has been added to the supplementary " "Here, a simple steady state calculation will be used to determine the Cl atom mixing ratio summarised below, but detailed within the supplementary: Cl2 + hv → Cl + Cl (1) ClNO2 + hv → Cl + NO2 (2) ClONO2 + hv → ClO + NO2 (3) HOCl + hv → OH + Cl (4) OClO + hv → O + ClO (5) OH + HCl → Cl + H2O (6) Cl + O3 → ClO + O2 (7) Cl + CH4 equivalent → HCl + products (8)

[Cl]SS = {2J1[Cl2] + J2[ClNO2] + J3[ClONO2] + J4[HOCl] +J5[OClO] +k7 [OH][HCl]} / {k7[O3] + k8[CH4] equivalent} (9) Where [CH4] equivalent represents the reactive VOC present as if it were reacting as CH4 Bannan et al., (2005), were able to use this steady state approach to compare the relative loss via reaction with OH compared with Cl atoms. Although this approach is an estimation, it was shown to produce comparable to results with that of the more rigorous MCM approach."

Page 14, lines 17-20. In the final paragraph of the paper, the authors claim that "chlorine atom chemistry may be under represented within models due to the lack of quantification and identification of particulate Cl-VOC products. This work provides instrumental capability to probe the competition between OH and Cl oxidation chemistry and quantify the SOA yields as a result of both pathways." This paper does not demon-

strate quantitative measurement of particulate Cl-VOCs, as no calibrations or desorption efficiencies have been presented or discussed. Furthermore, the authors have not sufficiently shown that particulate Cl-VOCs are necessary to understand the relative impacts of chlorine atoms and OH on air quality.

Response: The final paragraph has been clarified to state that these results highlight the deficiency in chlorine atom chemistry within models which could be a result of lack of quantification and identification of Cl VOCs in the gas and particle phase. The work here provides instrument capability to both identify and quantify the Cl VOCs in both the particle and gas phase which were shown in the steady state calculations to be important contributors to daytime oxidation. We believe the calibration of chloroacetic acid can be used for indirect quantification of Cl-VOCs and where the ability to measure the precursors to chloride radical production does demonstrate the ability to quantify the chloride radical budget and particulate Cl-VOCs. Action: The text has been amended and reads as follows "The results highlight deficiency in chlorine atom chemistry descriptions within models; possibly due to a lack in quantification and identification of Cl-VOC products in gas and particle phase. This work provides instrumental capability to probe the competition between OH and Cl oxidation chemistry and quantify their effect on ozone and SOA formation."

Figures would be more clear if panels were labelled with (A), (B), etc. Axis labels are often missing or unclear. For example, in Figure 1, presumably all the x-axes should be "m/z" and in Figure 5, the right-hand y-axis of the top left graph is unlabelled.

Response: These plots will be made clearer

Action: Axis have been labelled and the panels are now marked A, B, C etc

Page 23, lines 2-6. In the Figure 7 caption, it would be helpful to be explicit that terms are calculated and not measured.

Response: This is true and will be amended to state they are not measured values

[Figure]

Action: The caption now states these are steady state values

Reviewer 2

The manuscript suffers from organizational issues. Reactions are not consecutively numbered, sections were skipped, etc.

Response: These are to be addressed

Action: Sections are now sequential and reaction numbers are ordered appropriately

Mixing ratios of a variety of trace gases, including HCl, Cl2, ClONO2, HOCl, OClO and ClO as well as CMBO, isoprene, IOPEX, and benzene as well photolysis frequencies are presented but it is unclear in many cases how these data were acquired or how instrumental response factors were determined.

Response: The inorganic halogens have been given the same sensitivity to that of ClNO2. It has been shown by Le Breton et al 2017 that the inorganic halogens have a high and similar, if not the same, sensitivity to iodide ionisation. The acids presented are given the highest sensitivity (that of N2O5) to provide a minimum mixing ratio. As stated in the text, due to reduced availability of Cl-VOCs for laboratory calibration, CMBO and other Cl VOCs are given the same sensitivity as of chloroacetic acid, which is similar to that for ClNO2 which supports literature findings that inorganic halogens/functional groups possess a similar sensitivity. Detailed information about the PTR MS measurements can be found in Yuan et al 2012 and 2013. In brief, 28 masses are measured for the campaign at 1 Hz. Zero air, which was produced by ambient air passing through a platinum catalytic converter at 350 ïĆřC (Shimadzu Inc., Japan), was measured for 15 min every 2.5 hours to determine the background. used to measure background Aromatics masses (m/z 79 for benzene, m/z 93 for toluene, m/z 105 for styrene, m/z 107 for C8 aromatics and m/z 121 for C9 aromatics), oxygenated masses (m/z 33 for methanol, m/z 45 for acetaldehyde, m/z 59 for acetone, m/z 71 for MVK+MACR and m/z 73 for MEK), isoprene (m/z 69) and acetonitrile (m/z 42) were

calibrated by using EPA TO15 standard from Apel-Riemer Environmental Inc., USA. Formic acid (m/z 47), acetic acid (m/z 61), formaldehyde (m/z 31), and monoterpenes (m/z 81 and m/z 137) were calibrated by permeation tubes (VICI, USA). Photolysis rates were measured by a spectradiometer for O3, NO2, HCHO, HONO and H2O2. The photolysis rate of any given species was a given species was calculated by normalizing to the cross section and quantum yields taken from the recommendations of the Jet Propulsion Laboratory (JPL) kinetic evaluation report (Burkholder et al., 2015).

Action: The text regarding inorganic halogen quantification now states "The other inorganic halogens reported in this work are assumed to have the same sensitivity as ClNO2. This is in line with that Le Breton et al. (2017) reported many inorganic halogens possess a similar, if not the same, sensitivity, which is also supported by our chloroacetic acid calibration." The text states for other acids "Other acids identified by CIMS which are reported in the literature are given the sensitivity of N2O5 to provide a minimum concentration so no concentrations are over estimated." The text states for photolysis rates "Photolysis rates were measured by a spectradiometer for O3, NO2, HCHO, HONO and H2O2. The photolysis rate of any given species was calculated by normalizing to the cross section and quantum yields taken from the recommendations of the Jet Propulsion Laboratory (JPL) kinetic evaluation report (Burkholder et al., 2015). The text details the PTR calibration in the calibration section "Detailed information about the PTR MS measurements can be found in Yuan et al 2012 and 2013. In brief, 28 masses are measured for the campaign at 1 Hz. Zero air, which was produced by ambient air passing through a platinum catalytic converter at 350 °C (Shimadzu Inc., Japan), was measured for 15 min every 2.5 hours to determine the background. used to measure background Aromatics masses (m/z 79 for benzene, m/z 93 for toluene, m/z 105 for styrene, m/z 107 for C8 aromatics and m/z 121 for C9 aromatics), oxygenated masses (m/z 33 for methanol, m/z 45 for acetaldehyde, m/z 59 for acetone, m/z 71 for MVK+MACR and m/z 73 for MEK), isoprene (m/z 69) and acetonitrile (m/z 42) were calibrated by using EPA TO15 standard from Apel-Riemer Environmental Inc., USA. Formic acid (m/z 47), acetic acid (m/z 61), formaldehyde (m/z 31), and monoterpenes

(m/z 81 and m/z 137) were calibrated by permeation tubes (VICI, USA).

- Furthermore, concentrations of Cl and OH were calculated on the basis of steady state assumptions. These calculations are questionable since the only VOC measurements were by PTR-MS, an instrument that quantifies many but not all VOCs. Crucially, a PTR-MS usually does not quantify alkanes, whose abundances are important sinks for Cl atoms

Response: Reviewers 1 and 2 comments regarding the steady state calculations have been collected and answered below. The reviewers are correct that the reaction of OH+ HCl was accidentally omitted from the text in the original manuscript. The section has now been corrected for that omission. The reviewers are also correct that only using the small subset of hydrocarbons measured by the PTRMS would result in significant errors in estimated steady state Cl atom concentrations. However, we did not just use PTRMS measurements. In previous work (Bannan et al., 2015) we have shown that it is possible to calculate Cl atom concentrations, using a simple steady-state expression with the Cl atom production rate estimated from the observed loss rate of ClNO2 and removal of Cl atoms via reaction with the VOC concentrations supplemented with data from the Boston tailpipe study (AQIRP, 1995) and LA VOC study (Fraser et al., 1997), i.e. missing VOC concentrations are estimated simply by using the ratio of measured VOCs and missing VOCs from these urban studies and measured VOC data in this study. The removal of Cl atoms via reaction with VOCs can then be determined using Eq. 1-3, and NIST kinetic data (Manion et al., 2014) (Eq 1) (Eq 2) (Eq 3) We were able to show that the simple state approach agreed well with the MCM, despite a much more simplistic approach. We used an identical approach in this work utilizing the tailpipe VOC concentrations (AQIRP, 1995). However, we further simplify the approach by using one term CH4 equivalent which accounts for relative concentration and reactivity towards Cl, i.e. if a VOC reacts 1000 times faster it is the equivalent of 1000 CH4 or more formally for each VOC its CH4 equivalent is kCl + VOC [VOC]/ kCl + CH4. Whilst the approach is a simplification of course, it has been shown that using these

emissions it is possible to estimate the Cl atom production in a Megacity environment and produces results that are comparable with the much more thorough modelling approach of the MCM. It also generates a metric, CH4 equivalent, which can be used as a comparative measurement from city to city. Photolysis rates were measured by a spectradiometer for O3, NO2, HCHO, HONO and H2O2. The photolysis rate of any given species was calculated by normalizing to the cross section and quantum yields taken from the recommendations of the Jet Propulsion Laboratory (JPL) kinetic evaluation report (Burkholder et al., 2015). The reviewer is correct that Cl atoms will produce peroxy radicals which can perturb HOx/NOx cycles, and we were only considering the initial oxidation of VOCs. The text has been corrected to reflect that.

Action: the following text has been added to the manuscript and text has been added to the supplementary "Here, a simple steady state calculation will be used to determine the Cl atom mixing ratio summarised below, but detailed within the supplementary: Cl2 + hv → Cl + Cl (1) ClNO2 + hv → Cl + NO2 (2) ClONO2 + hv → ClO + NO2 (3) HOCl + hv → OH + Cl (4) OClO + hv → O + ClO (5) OH + HCl → Cl + H2O (6) Cl + O3 → ClO + O2 (7) Cl + CH4 equivalent → HCl + products (8)

[Cl]SS = {2J1[Cl2] + J2[ClNO2] + J3[ClONO2] + J4[HOCl] +J5[OClO] +k7 [OH][HCl]} / {k7[O3] + k8[CH4] equivalent} (9) Where [CH4] equivalent represents the reactive VOC present as if it were equivalent CH4 Bannan et al., (2105), were able to use this steady state approach to compare the relative loss via reaction with OH compared with Cl atoms. Although this approach is an estimation, it was shown to produce comparable to results with that of the more rigorous MCM approach."

Some data (e.g., OClO, CMBO, ClO, ClONO2, IOPEX) are only semi-quantitative and should be presented as such.

Response: Many CIMS papers refer to relative calibrations. Here we apply this method to non-calibrated compounds but do apply a maximum sensitivity to limit their impact on model calculations. Their semi quantitative nature will be noted within the text.

[Figure]

Action: The compounds which are not directly are noted in the calibration section to ensure the reader is aware that the values are semi quantitative.

pg 1 line 27 –replace the comma with "and"

Response: This has been altered

Action: the comma has been replaced with "and"

line 29 – ppt is not a concentration unit – please rephrase

Response: This will be changed (we have changed concentrations to mixing ratios)

Action: The units now used are pptV and referred to as mixing ratios and referred to as mixing ratios

pg 2 lines 17 – (O3, HOx, and NOx levels via ... (R1-R9)). Most of the Cl will likely abstract hydrogens from hydrocarbons (R11), in particular at this site. Another important reaction omitted here is OH+HCl->H2O+Cl. Consider reorganizing the introduction to reflect this.

Response: We agree that the OH + HCl reaction must be included within the reactions presented and noted in the text. The hydrogen abstraction to form HCl is within the text and reaction list, although we have no further iterated to the reader that this is the major reaction pathway for chloride-VOC reactions.

Action: The reactions listed (now R1-R11) contain the OH + HCl reaction. We have amended the text to state hydrogen abstraction is the dominant pathway for chloride oxidation "The oxidation mechanism of saturated hydrocarbon (R12-R13) is initiated by reaction with OH or chlorine atom to form an organic peroxy radical (RO2), and H2O or HCl depending on the oxidant, which is the dominant pathway for chloride-VOC reactions."

pg 3 lines 2-4. –"This perturbation is currently thought to only be significant in the early hours of the day while OH concentrations are low and chlorine atom production

is high through the photolysis of ClNO2." I don't think this is correct. Reaction of Cl with alkanes produces peroxy radicals, which feed into the "regular" HOx/NOx cycles. Thus, the early morning injection of radicals impacts (perturbs) radical chemistry for the remainder of the day. Perhaps the authors meant to say "Oxidation of VOCs by Cl is currently thought to be ..."?

Response: This reviewer is correct. Text has been added to clarify this point and that we only consider the initial oxidation of VOCs

Action: The following text has been added "The oxidation of VOCs by chlorine atoms is thought to be significant in the early hours of the day while OH mixing ratio are low and chlorine atom production is high through the photolysis of ClNO2, as well as feeding into the standard HOx/NOx cycles via production of peroxy radicals from reactions with alkanes."

lines 11/12 – there are two reactions labelled R11

Response: This is correct and should be changed

Action: Reactions here are now R11 to R14

line 31 – "36%" - please add the value for SOA yield from OH initiated oxidation of isoprene for comparison.

Response: The Liu et al 2016 has been cited which calculates a yield of 15% for comparison. Action: The Liu et al 2016 has been cited which calculates a yield of 15% for comparison, although this is known to be a factor of 2 higher than used in standard climate models.

pg 4 line 24 – how were photolysis rates determined?

Photolysis rates were measured by a spectradiometer for O3, NO2, HCHO, HONO and H2O2. The photolysis rate of any given species was a given species was calculated by normalizing to the cross section and quantum yields taken from the recommendations of the Jet Propulsion Laboratory (JPL) kinetic evaluation report (Burkholder et al., 2015).

Action: The text now states "Photolysis rates were measured by a spectradiometer for O3, NO2, HCHO, HONO and H2O2. Inorganic halogen photolysis rates were extracted relatively from these J rates."

line 31 – there are two Le Breton et al. 2017 references. Please label them 2017a and 2017b.

Response: These references have been noted as 2017a and 2017b in the text and reference list

Action: These references have been noted as 2017a and 2017b in the text and reference list

pg 5 lines 1-2 Please provide more detail as to how the PTR-MS was calibrated, what molecules were quantified, etc

Response: Detailed information about the PTR MS measurements can be found in Yuan et al 2012 and 2013. In brief, 28 masses are measured for the campaign at 1 Hz. Zero air, which was produced by ambient air passing through a platinum catalytic converter at 350 ïĆřC (Shimadzu Inc., Japan), was measured for 15 min every 2.5 hours to determine the background. used to measure background Aromatics masses (m/z 79 for benzene, m/z 93 for toluene, m/z 105 for styrene, m/z 107 for C8 aromatics and m/z 121 for C9 aromatics), oxygenated masses (m/z 33 for methanol, m/z 45 for acetaldehyde, m/z 59 for acetone, m/z 71 for MVK+MACR and m/z 73 for MEK), isoprene (m/z 69) and acetonitrile (m/z 42) were calibrated by using EPA TO15 standard from Apel-Riemer Environmental Inc., USA. Formic acid (m/z 47), acetic acid (m/z 61), formaldehyde (m/z 31), and monoterpenes (m/z 81 and m/z 137) were calibrated by permeation tubes (VICI, USA).

Action: The text now states in section 2.1 "Detailed information about the PTR MS

measurements can be found in Yuan et al 2012 and 2013. In brief, 28 masses are measured for the campaign at 1 Hz. Zero air, which was produced by ambient air passing through a platinum catalytic converter at 350 °C (Shimadzu Inc., Japan), was measured for 15 min every 2.5 hours to determine the background. used to measure background Aromatics masses (m/z 79 for benzene, m/z 93 for toluene, m/z 105 for styrene, m/z 107 for C8 aromatics and m/z 121 for C9 aromatics), oxygenated masses (m/z 33 for methanol, m/z 45 for acetaldehyde, m/z 59 for acetone, m/z 71 for MVK+MACR and m/z 73 for MEK), isoprene (m/z 69) and acetonitrile (m/z 42) were calibrated by using EPA TO15 standard from Apel-Riemer Environmental Inc., USA. Formic acid (m/z 47), acetic acid (m/z 61), formaldehyde (m/z 31), and monoterpenes (m/z 81 and m/z 137) were calibrated by permeation tubes (VICI, USA)."

line 17 - section 2.3 is absent.

Response: This is correct and will be amended

Action: Calibration is now section 2.3

Please describe how the response factors for HCl, Cl2, ClONO2, HOCl, OClO and ClO (Figure 3) were determined

Response: As described above, these were given the highest sensitivity to limit their impact on the modelling results

Action: This has now been described in the text more clearly

lines 24-25 – "The N2O5 diffusion source was held at a constant temperature (-23 C), and the mass loss rate was characterized gravimetrically for a flow rate of 100 sccm." N2O5 is quite hygroscopic, such that the diffusion source could "gain weight" simply by absorbing residual moisture. Another potential error with this method is "loss" of NO3 (e.g., through reaction with impurities on the wall) followed by loss of NO2 (toward which the CIMS is probably blind). All this probably doesn't matter since there was a CEAS on site.

Response: The technique has been described by Faxon et al 2017 in full detail and the CEAS measurements confirm the accuracy of the calibration technique

Action: The technique has been described by Faxon et al 2017 in full detail and the CEAS measurements confirm the accuracy of the calibration technique

How stable/accurate/reproducible is this source? Could it be used as standalone N2O5 calibration method?

Response: The sources stability over time has not been thoroughly tested for longer than a week, upon stable cooling I believe it can be used as a reproducible source. The CEAS utilised a different calibration technique as the N2O5 CIMS calibration was performed post campaign. The CEAS was separately calibrated as detailed in Wang et al 2017a "Development of a portable cavity-enhanced absorption spectrometer for the measurement of ambient NO3 and N2O5: experimental setup, lab characterizations, and field applications in a polluted urban environment" A brief description of the CEAS calibration has been added, stating that the mirror was calibrated for daily and the filter replaced hourly "The CEAS utilised a dynamic source by mixing NO2 and O3 to generate stable N2O5 for calibration (Wang et al., 2017). The source was used to calibrate the ambient sampling loss of N2O5 in the sampling line, filter, the preheater cavity and optical cavity. This was performed pre and post campaign. During the campaign the reflectivity of the high reflectivity mirror was calibrated daily and filter changed hourly."

Action: The following text was added "The CEAS utilised a dynamic source by mixing NO2 and O3 to generate stable N2O5 for calibration (Wang et al., 2017). The source was used to calibrate the ambient sampling loss of N2O5 in the sampling line, filter, the preheater cavity and optical cavity. This was performed pre and post campaign. During the campaign the reflectivity of the high reflectivity mirror was calibrated daily and filter changed hourly."

Please state if the diffusion source method has been verified using CEAS (which I

assume it has).

Response: It has not been verified with the CEAS, but has with the CIMS in previous literature (Faxon et al., 2017) and the good agreement here between the CEAS and CIMS illustrate its ability.

Action: As stated above, the instruments were not calibrated using the same source, but independently calibrated. The above text was also added.

Line 36 – "these sensitivities" – please state the instrumental response factors here

Response: The sensitivities have been added

Action: The following text has been amended "These sensitivities for N2O5 and ClNO2 (9.8 and 1.6 ion counts per ppt Hz-1 for 1x105 iodide ion counts) were applied relatively to that of formic acid."

pg 6 line 20/21 –"A quadrupole CIMS may not be able to resolve the peak adjacent to ClO at mass 178 and the second dominant peak for the ClNO2 fit would result in a 10% over estimation." Please clearly state what ions are present at mass 178.

Response: The peaks have been stated and are now included in figure 1

Action: The peaks have been stated and are now included in figure 1

It is unclear what is meant by "second dominant peak for the ClNO2 fit" – is this at m/z 208? Response: There is a second peak fitted (as shown in figure 1) which is identified as a cluster of nitric acid with water forming an adduct with iodide that will contribute to up to 10% of the counts at this unit mass

Action: This identification has been stated in the text and provided in figure 1

line 32 – please put the N2O5 mixing ratios in context – (temperature, O3 and NO2 levels, NO3 production rate etc.)

Response: The Wang et al 2018 paper focuses on analysis of N2O5 from the CEAS

utilising the CIMS ClNO2 data. We therefore believe that reporting the typical concentrations and diurnal trends suffice here as the focus is more on the inter-comparison rather than production rates of N2O5 and ClNO2. This information can be added if requested by the editor, although we believe it does not contribute to the manuscript.

Action: NA

line 33 – Were the instruments operated on the same inlet? If not, there may be scatter simply from sampling air at slightly different locations.

Response: They were not on the same inlet and faced different directions

Action: This is now stated in the text

line 36 – The offset should have units of ppt

Response: this has been amended

Action: this has bene amended

pg 7 line 2 – "although averaging at 4 ppt" perhaps better to give a relative error here

Response: The average error has been reported now in the text, 11%.

Action: The text now reads "The largest error between the two measurements occurs at night during the higher levels of N2O5, although averaging at 4 ppt (representing 11% error on the average campaign concentration)."

line 3 – please move details on how instruments were operated (heated IMR) to section 2.2

Response: The details of the heated IMR have been added to section 2.2, although this section still refers to this setup to clarify the possible physical differences resulting in variation of the measured mixing ratio.

Action: Section 2.2 now reads "The ionized gas was then carried out of the ion source and into the Ion-Molecule Reaction (IMR) chamber, which was heated to 40 degrees

Celsius to reduce wall loss, through an orifice (Ø = 1 $\mu$m)."

line 21- "Inorganic chlorine abundance and profiles". There is a lot presented in this section, BB, WRF etc, that goes well beyond inorganic chlorine abundances and profiles. This section should be broken up into smaller, more coherent pieces.

Response: We agree with the reviewer and have now renamed this section and added two sub sections

Action: This section is now named "3.3 Inorganic chlorine: Abundance, profiles and source" We have added section 3.3.1 called "Abundance and profiles" and section 3.3.2 called "Source of chloride"

line 24 – mixing ratio, not concentration

Response: This is to be changed

Action: mixing ratio is now used instead of concentration

line 25 – "$\sigma$ 270 ppt" is this standard deviation?

Response: yes

Action: the text now states "510 ppt (standard deviation ($\sigma$) 270 ppt)"

pg 8 line 9. Please comment on the possibility of chlorine nitrate forming on the inner walls of the inlet.

Response: text has been added to hypothesise this as a possibility to the reader

Action: The following text has been added "IMR chemistry is also not a possible source as these reactions would occur throughout the day, therefore skewing all of the data and not just the night-time levels, although there is a possibility that ClONO2 can be formed in the IMR by reactions between ClO and NO2."

pg 8 line19. –" This suggests the chlorine has an anthropogenic source and not marine" I disagree. One has to be careful with the interpretation of AMS data. The "stan-

dard" AMS chloride product only includes non-refractory aerosol, i.e., does not include sea salt chloride – for one, it does a poor job volatilizing NaCl, and most AMS have a size cut off of 1 micron that filters out most of the larger sea salt aerosol particles. The correlation of AMS chloride with anthropogenic tracers may arise from acid displacement of sea salt chloride in polluted air (that is high in SO2). I'd suggest rewording the entire paragraph (lines 14-27). I don't doubt that anthropogenic Cl sources contribute, but there aren't enough data (shown in this paper) to proof a negligible marine influence.

Response: The response here is similar to the first comment by reviewer 1. We acknowledge the limitations of the AMS data and will provide that information to the reader. We ran the WRF model simulation to observe how the sea salt model results correlate with ClNO2 profiles. This resulted in a poor correlation (figure S4) whereas total CO, as stated in the manuscript, had a relatively good correlation with ClNO2, further supporting the anthropogenic source of chloride. Furthermore, the modelled sea salt levels are very low, most likely unable to produce the mixing ratios of ClNO2 observed by the CIMS. Figure S4 has been added to the supplementary.

Action: Section 3.3.2 has been amended as displayed below to acknowledge the AMS limitations and further show model runs indicating no correlation between sea salt and ClNO2 "The high levels of ClNO2 indicate a local significant source of chlorine to support such high yields. The dominant source of chlorine atoms for ClNO2 production within models, such as the Master Chemical Mechanism (MCM), is from sea salt, although the site is situated 200 km from the Yellow Sea and therefore has low probability that the Cl- has this origin. The mean AMS chloride mass loading was 0.05 $\mu$g m-3 for the campaign with a maximum of 1.7 $\mu$g m-3. The Cl- from the AMS appears to be correlated strongly with CO and SO2, possibly originating from power plants or combustion sources. This could be a result of Cl- originating from anthropogenic sources, although the AMS data does not include refractory aerosol and also has a cut off size larger than most sea salt particles. The high Cl- observed appears to flow into the site from the

mainland (Figure 4) and not from the nearest coast, further supporting a strong anthropogenic source. Tham et al., (2016) observed a strong correlation of aerosol chloride with SO2 and potassium in the same season in 2014 at Wangdu (semi-rural site 160 km south West of Beijing) and suggested contribution to fine chloride from burning of coal and crop residues. The latter was also supported by satellite fire spot count data (Tham et al., 2016). Riedel et al. (2013) have previously reported high ClNO2 mixing ratios observed from urban and power plant plumes measuring high mixing ratios of gas phase Cl2. The correlation with SO2 indicates coal burning as a potential source of particulate chlorine which is known to be a significant source of PM in the Beijing region (Ma et al., 2017), and correlation with CO and benzene is an indicator of biomass burning (Wang et al., 2002). In order to test the hypothesis of biomass burning as a source of particulate chlorine, biomass burning emissions and transport utilising the EMEP MSC-W chemical transport driven by meteorology from the WRF-ARW model (Skamarock et al., 2008) were ran. Neither of these two biomass burning databases available contain data on chlorine emissions, so instead the biomass burning emissions of CO (CObb) were tracked and compared to the total mixing ratio of CO (COt) at the Changping site. CO was chosen since the measurements at Changping had shown strong correlation between CO and ClNO2 and because CO could be expected to be co-emitted with chlorine for both biomass burning and industrial combustion. Figure S2 (supplementary) shows time series of the measured ClNO2 mixing ratios at the Changping site, as well as the modelled mixing ratios of COt and CObb. CObb is shown for both the FINN and GFAS model runs, while COt is only shown for the FINN run since it looks almost completely the same for the GFAS run. From this figure it is clear that mixing ratios of CObb are very low compared to COt. The two episodes of increased mixing ratio, May 18-May 23 and May 28-June 5, are to some extent visible in all time series, but for the biomass burning CO series, the second episode is much less pronounced. Night-time averages of the mixing ratios shown in figure S3 were calculated for each night for the time period 18:00 to 08:00 local time (UTC+8), roughly corresponding to the period when ClNO2 is not destroyed by photolysis. Nights

with significant amount of missing data for the measurements were excluded. Figure S3 shows scatter plots of these averages of ClNO2 against the averages of the other species. Figure S3 also shows a straight line fitted for each of these scatter plots. The R2 for these lines were 0.48, 0.04, and 0.21 for COt, CObb FINN, and CObb GFAS respectively. The fact that mixing ratios of CObb is so much smaller than COt according to the model, combined with the much better correlation for COt than for CObb strongly suggests that industrial emissions are the dominant source of chlorine, rather than biomass burning. To further support probe the source of chloride, the model was run to calculate sea salt levels instead of CO and resulted in a poor correlation with the ClNO2 time series (figure S3). The absolute levels of sea salt in the model were also very low, unlikely able to produce the observed mixing ratios of ClNO2 observed by CIMS." line 28 - Please describe the WRF model in the methods section, not in the results section. Response: The model description has bene moved to section 2.3 "model setup" Action: A new section (2.3) has been populated with the model description.

I'd remove the WRF simulations as they may not account for local BB – chemical tracers would be more robust.

Response: We have now utilised the WRF simulations further to support the hypothesis that the ClNO2 measured cannot be a product of only seasalt particle heterogeneous reactions and anthropogenic chloride must play a significant role

Action: NA

pg 9 line 13. All that is shown is that WRF modeling suggests BB to be a small source of chlorine – it doesn't show industrial emissions. Please rephrase.

Response: We agree this is too strong a claim and have rephrased the sentence

Action: The sentence now reads "The fact that mixing ratios of CObb is so much smaller than COt according to the model, combined with the much better correlation for COt than for CObb strongly suggests that industrial emissions are a more significant

source of chlorine, rather than biomass burning."

line 14 The particle desorption profiles should be discussed in their own section.

Response: This has been placed into its own section

Action: Section 3.4 is now "particle phase ClNO2" and the other sections following have been renumbered

lines 14-20. Did you observe the peak at 210? Please expand the AMU axis in Figure 5 to show it.

Response: Yes we did but again dominating in the particle phase

Action: The m/z axis has been expanded to 210.5

line 30 " these data indicates a significant amount of the chlorine associated with ClNO2 is not liberated from the particle phase" it should be "these data indicate" More to the point, you observe that you can drive off ClNO2 if you heat aerosol. Have you considered that additional ClNO2 could be formed by thermally driven reactions? If not, please state that this is a major assumption made here.

Response: This is a assumption to be stated

Action: This assumption is now stated

line 31-33 "The slope" please show this plot (perhaps as an insert in Figure 5).

Response: This is merely a ratio of the red and blue (gas and particle) time series in panel A of figure 5.

Action: We feel this does not need to be added as it is just utilisation of the data already displayed in the figure

Personally, I wouldn't call 5% "significant" considering this is much less than the measurement (calibration) error.

Response: The term significant has been replaced with "non-negligible"

Action: The term significant has been replaced with "non-negligible"

pg 10 line 18 The numbering of the reactions is inconsistent with those on pg 2. Some reactions are unnecessarily duplicated.

Response: The reactions have been renumbered and balanced

Action: The reactions have been renumbered and balanced

line 25 - Please number the steady state expression.

Response: This has been done

Action: The steady state expression is number (8)

A major source of Cl atom is the reaction OH + HCl -> H2O + Cl, which should not be omitted here. line 26 - And how was "equivalent CH4" determined for this site? It must be massive. It is very likely that the PTR-MS misses most of it, for example all of the alkanes (Table 2 of de Gouw's Mass Spectrometry Reviews 26, 223 (2007)).

Response – the above 2 comments have been answered in the first question regarding the ss as mentioned within the response.

pg 11 line 9 – "The results show that both at the UK marine and urban site max chlorine atom concentrations are more than an order of magnitude lower than the mean of Beijing." Considering the uncertainty of the Cl atom sinks, the authors should only compare Cl atom production rates. Comparing rural and urban sites (Weybourne with Beijing) is like comparing apples and oranges. Many other groups have calculated Cl atom production rates from ClNO2 photolysis, including many polluted urban sites. How do the numbers of this study stack up to these?

Response: We agree with the reviewer that other studies need to be considered in this section. We still believe that comparing urban and rural sites, or indeed urban and

marine sites is an important factor as this paper postulates the impact of anthropogenic chloride and challenges the significance of sea salt as the major source for ClNO2 production. Therefore we would like to keep this comparison within the manuscript.

Action: The following text has bene added to consider previous calculations of chloride radical production. "Studies of chloride atom production in Los Angeles by Riedel et al. (2012) and Young et al. (2014) indicate that the high production rate in Beijing is somewhat typical of urban sites, although HCl and ClNO2 contribution to radical production is the same, whereas here we see very little chloride atom production from HCl in comparison to ClNO2."

Imo, the entire section comparing OH and Cl abundances is questionable

Response: We have now added to this section to present the steady state in a more detailed manor and supported its utilisation referencing the Bannan et al paper which was compared against the MCM for OH radical concentrations. We therefore feel its role within the manuscript has been validated and adds to reliable scientific analysis of the dataset Action: NA

pg 12 line 30 – "longer atmospheric lifetime" how long are the lifetimes of CMBO and of isoprene?

Response: The lifetime of isoprene is 1-2 hours according to Atkinson et al. (2000) , but we do not know the lifetime of CMBO

Action: Although this assumption is made based on the diurnal profiles, we agree we cannot assume the lifetime of CMBO is longer than that of isoprene and therefore have removed this phrase from the manuscript.

line 34 – "The concentrations of Cl and isoprene were relatively low" How low is relatively low? Please be quantitative.

Response: Quantitative values will be added t the text

Action: The exact mean values have been added "1.6x105 molecules cm-3 s-1 Cl and 0.5 ppb isoprene"

If CMBO abundances did not follow those of its precursors, does that imply that CMBO can be primary (or originates from other precursors)?

Response: In response to the below comment, we now acknowledge in the manuscript that CMBO may not be unique to isoprene-chlorine reactions and therefore could have alternative sources

Action: We have added the following text to support the qualitative analysis of the CMBO time series "CMBO may also not be unique to only isoprene-chloride reactions and therefore have alternative sources not represented in this data set."

How certain are we that CMBO is a unique marker of chlorine-isoprene chemistry (line 25)?

Response: We agree with the reviewer that it may be a unique marker of chlorine chemistry, but not proven to be unique to isoprene reactions

Action: The sentence has been rephrased to state "unique marker of chlorine chemistry"

pg 13 line 36 " ClNO2 was potentially identified in the particle phase " I agree, but in the preceding text, ClNO2 was not only identified but also quantified, or was it? Either way, the earlier section is inconsistent with the much more conservative conclusion in the end.

Response: The text states the observation and examines the possible causes which we do not categorically prove is only an instrument artefact therefore we state in the conclusion that it is indeed identified but at unusually high concentrations. We do not state it is quantified as the quantification in the gas phase is sufficient and the same sensitivity as for the particle phase.

Action: The text now reads "ClNO2 was identified in the particle phase at higher ratios with respect to its gas phase component than expected, which may only prove to be significant at such elevated mixing ratios as observed in East Asia."

pg 14 – many references are incomplete (e.g., Pszenny et al.) and most are missing their doi.

Response: The reference has been checked and any inaccuracies or missing doi's have been added

Action: The reference has been checked and any inaccuracies or missing doi's have been added

pg 19 – Figure 1. Please identify the green, gold/yellow, and magenta lines. For the second panel, it would be useful to show a blank (zero) measurement also.

Response: These peaks have bene identified and labelled din the MS plot

Action: The peaks for ions C8H5NO4, C6F3HO3, C9H8NO3, IHNO3H2O and C9H5SO4 are now colour labelled in the MS plot

pg 20 – please define the "C" and "M" terms

Response: They will be defined

Action: C and M are now defined in the caption

pg 23 – Figure 7A or 7B – one of the "y" axes is mislabelled

Response: The labels are correctly labelled

Action: The labels are correctly labelled

References AQIRP, 1995, Effects of gasoline T50, T90 and sulfur on exhaust emissions of current and future technology vehicles. Auto/Oil Air Quality Improvement Research Program, Technical Bulletin No. 18. Bannan T. et al., (2015). The first UK measurements of nitryl chloride using a chemical ionisation mass spectrometer

in central London in the summer of 2012, and an investigation of the role of Cl atom oxidation. J. Geophys. Res., 120(11), 5638-5657. Burkholder, J.B. et al. Chemical Kinetics and Photochemical Data for Use in Atmospheric Studies: Evaluation Number 18. Jet Propulsion Laboratory, California Institute of Technology, Pasadena, CA, 2015. Fraser, M. P., G. R. Cass, B. R. Simoneit, & R. A. Rasmussen (1997). Air quality model evaluation data for organics. 4. C2-C36 non-aromatic hydrocarbons. Environmental science & technology, 31(8), 2356-2367 DOI: 10.1021/es960980g Gao, J., Tian, H., Cheng, K., Lu, L., Zheng, M., Wang, S., Hao, J., Wang, K., Hua, S., Zhu, C., and Wang, Y.: The variation of chemical characteristics of PM2.5 and PM10 and formation causes during two haze pollution events in urban Beijing, China, Atmospheric Environment, 107, 1-8, https://doi.org/10.1016/j.atmosenv.2015.02.022, 2015. Hofzumahaus, Andreas, Franz Rohrer, Keding Lu, Birger Bohn, Theo Brauers, Chih-Chung Chang, Hendrik Fuchs, et al. "Amplified Trace Gas Removal in the Troposphere." Science 324, no. 5935 (2009): 1702. Manion, J. A., R. E. Huie, R. D. Levin, D. R. Burgess Jr, V. L Orkin, W. Tsang, W. S. McGivern, J. W. Hudgens, V. D. Knyazev, D. B Atkinson, E. Chai, A. M. Tereza, C.-Y. Lin, T. C. Allison, W. G. Mallard, F. Westley, J. T. Herron, R. F. Hampson, and D. H Frizzell (2014) NIST Chemical Kinetics Database, NIST Standard Reference Database 17, Version 7.0 (Web Version), Release 1.6.8, Data version 2013.03, National Institute of Standards and Technology, Gaithersburg, Maryland, 20899-8320. Web address: http://kinetics.nist.gov/ Tang, R., Wu, Z., Li, X., Wang, Y., Shang, D., Xiao, Y., Li, M., Zeng, L., Wu, Z., and Hallquist, M.: Primary and secondary organic aerosols in summer 2016 in Beijing, Atmospheric Chemistry and Physics, 18, 4055-4068, 2018. Zhang, R., Jing, J., Tao, J., Hsu, S.-C., Wang, G., Cao, J., Lee, C. S. L., Zhu, L., Chen, Z., and Zhao, Y.: Chemical characterization and source apportionment of PM 2.5 in Beijing: seasonal perspective, Atmospheric Chemistry and Physics, 13, 7053-7074, 2013. Yuan, B., Shao, M., de Gouw, J., Parrish, D. D., Lu, S. H., Wang, M., Zeng, L. M., Zhang, Q., Song, Y., Zhang, J. B., and Hu, M.: Volatile organic compounds (VOCs) in urban air: How chemistry affects the interpretation of positive matrix factorization (PMF) analysis, J. Geophys. Res.-Atmos., 117, Artn

**ACPD**

D24302, 10.1029/2012jd018236, 2012. Yuan, B., Hu, W. W., Shao, M., Wang, M., Chen, W. T., Lu, S. H., Zeng, L. M., and Hu, M.: VOC emissions, evolutions and contributions to SOA formation at a receptor site in eastern China, Atmos. Chem. Phys., 13, 8815-8832, 10.5194/acp-13-8815-2013, 2013.

Please also note the supplement to this comment:
https://www.atmos-chem-phys-discuss.net/acp-2018-9/acp-2018-9-AC1-supplement.pdf

[Figure]

**Fig. 1.**

[Figure]

Fig. 2.

**Supplement:**

1  **Supplementary information for: Chlorine oxidation of VOCs at a semi-rural site in**
2  **Beijing: Significant chlorine liberation from ClNO₂ and subsequent gas and particle**
3  **phase Cl-VOC production**

5  **Steady state calculations**

6  In previous work (Bannan et al., 2015) we have shown that it is possible to calculate Cl atom

7  concentrations, using a simple steady-state expression with the Cl atom production rate estimated from

8  the observed loss rate of ClNO2 and removal of Cl atoms via reaction with the VOC concentrations

9  supplemented with data from the Boston tailpipe study (AQIRP, 1995) and LA VOC study (Fraser et al.,

10  1997), i.e. missing VOC concentrations are estimated simply by using the ratio of measured VOCs and

11  missing VOCs from these urban studies and measured VOC data in this study. The removal of Cl atoms

12  via reaction with VOCs can then be determined using Eq. 1-3, and NIST kinetic data (Manion et al., 2014)

13
$$-d[alkanes]/dt = [X]\sum_i k_{X+alkane,i}[alkane,i]$$
(Eq 1)

14
$$-d[alkenes]/dt = [X]\sum_i k_{X+alkene,i}[alkene,i]$$
(Eq 2)

15
$$-d[alkynes]/dt = [X]\sum_i k_{X+alkyne,i}[alkyne,i]$$
(Eq 3)

16  We were able to show that the simple state approach agreed well with the MCM, despite a much more

17  simplistic approach.   We used an identical approach in this work utilizing the tailpipe VOC

18  concentrations (AQIRP, 1995). However, we further simplify the approach by using one term CH4

19  equivalent which accounts for relative concentration and reactivity towards Cl, i.e. if a VOC reacts 1000

20  times faster it is the equivalent of 1000 CH4 or more formally for each VOC its CH4 equivalent is kCl +

21  VOC [VOC]/ kCl + CH4. Whilst the approach is a simplification of course, it has been shown that using

22  these emissions it is possible to estimate the Cl atom production in a Megacity environment and

23  produces results that are comparable with the much more thorough modelling approach of the MCM.

24  It also generates a metric, CH4 equivalent, which can be used as a comparative measurement from city

25  to city.

26  Photolysis rates were measured by a spectradiometer for O3, NO2, HCHO, HONO and H2O2. The

27  photolysis rate of any given species was calculated by normalizing to the cross section and quantum

28  yields taken from the recommendations of the Jet Propulsion Laboratory (JPL) kinetic evaluation report

29  (Burkholder et al., 2015)."

1 "Consistent with past measurements and the measurements of this study, $ClNO_2$ is expected to provide

2 a significant source of Cl during day time hours, presenting a potentially significant source of the

3 reactive Cl atom during the day. Its rapid photolysis rate and elevated mixing ratios enables Cl to

4 compete with OH oxidation chemistry, the known dominant daytime radical source. Here, a simple

5 steady state calculation will be used to determine the Cl atom mixing ratio as detailed;

6 $$Cl2 + hv \rightarrow Cl + Cl \qquad (1)$$

7 $$ClNO_2 + hv \rightarrow Cl + NO2 \qquad (2)$$

8 $$ClONO2 + hv \rightarrow ClO + NO2 \qquad (3)$$

9 $$HOCl + hv \rightarrow OH + Cl \qquad (4)$$

10 $$OClO + hv \rightarrow O + ClO \qquad (5)$$

11 $$OH + HCl \rightarrow Cl + H2O \qquad (6)$$

12 $$Cl + O3 \rightarrow ClO \quad + O2 \qquad (7)$$

13 $$Cl + CH4 \ equivalent \rightarrow HCl + products \qquad (8)$$

14 $[Cl]SS = \{2J1[Cl2] + J2[ClNO_2] + J3[ClONO2] + J4[HOCl] + J5[OClO] + k7 [OH][HCl]\} / \{k7[O3] + k8[CH4]$

15 equivalent\} (9)

16 Where [CH4] equivalent represents the reactive VOC present as if it were equivalent CH4

17 Bannan et al., (2005), were able to use this steady state approach to compare the relative loss via

18 reaction with OH compared with Cl atoms. The total loss can be estimated using equations 1-3, using

19 the steady state concentrations of Cl (estimated using equation 9)

20 $[Cl]SS = \{2J1[Cl2] + J2[ClNO_2] + J3[ClONO2] + J4[HOCl] + J5[OClO] + k7 [OH][HCl]\} / \{k7[O3] + k8[CH4]$

21 equivalent\} (9)

22 and a mean steady state OH calculated concentration of $7 \times 10^6$ molecules cm3. Again, this approach

23 is an estimation, but was shown to produce comparable to results with that of the more rigorous

24 MCM approach. The main findings of this work, is that as we infer much higher concentrations of Cl

25 atoms, as a result of much higher observed inorganic chlorine species, the impact of Cl atom

26 chemistry is much higher than has been observed in previous work.

27

[Figure]

**Figure S1. Wind rose during the campaign in Changping**

[Figure]

**Figure S2. Time series of measured ClNO$_2$ concentrations, modelled CO$_t$ concentrations, and modelled CO$_{bb}$ at the Changping site. CO$_{bb}$ is shown for both the model using the FINN database and the GFAS database. Periods with missing measurement data are shown in grey.**

[Figure]

2 **Figure S3. Scatter plots of night-time averages of ClNO$_2$ against corresponding averages**

3 **of CO$_t$, CO$_{bb}$ in the FINN model, and CO$_{bb}$ in the GFAS model. Linear regressions for**

4 **each of the comparisons gave the following r2 results: CO$_t$ 0.48, CO$_{bb}$ FINN 0.04, and**

5 **CO$_{bb}$ GFAS 0.21.**

[Figure]

6

7 **Figure S4.  Correlation plots of measured ClNO2 vs modelled COt (green), fine seasalt**

8 **(red) and course seasalt (blue).**

9

---

## Author Response (AR1)

Response to reviewers on "Chlorine oxidation of VOCs at a semi-rural site in Beijing: Significant chlorine liberation from ClNO₂ and subsequent gas and particle phase Cl-VOC production" by Michael Le Breton et al.

Reviewer 1

The authors claim a novel aspect of their findings is the anthropogenic source of reactive chlorine in China (e.g. page 11, line 35). However, this is based on the absence of sodium chloride from an aerosol mass spectrometer (AMS) measurement. The chloride measured by AMS is only non-refractory (i.e. primarily ammonium chloride) and excludes sodium chloride. According to the AMS method cited in this manuscript (Hu et al., 2016), only non-refractory chloride is measured. Discussions including this must be re-considered.

Response: We firstly agree that there are limitations on the AMS measurements of particulate chloride due to the refractory component which the AMS will not measure, although there have been several further steps in the manuscript to probe the anthropogenic source of chloride observed, such as the correlation with $SO_2$, benzene and CO. Furthermore, the large distance (200k m) of the site from the coast makes it difficult to transport the seasalt to the site.

We have now performed a wind rose analysis in more detail to further probe the source of chloride, whose results are shown in figure S1, in which radial and tangential axes represent the wind direction and speed (km $h^{-1}$) and the colour bar represents the PM2.5 concentration. We see during the campaign, the severe pollution was from the south and southwest, with little contribution from the east part. Therefore, we could deduce that little contribution of the chloride was from the ocean.

In addition, we have now done WRF model simulation to observe how a sea salt model results would correlate with $ClNO_2$ profiles. This resulted in a poor correlation (figure S2) whereas total CO, as stated in the manuscript, had a relatively good correlation with $ClNO_2$, further supporting the anthropogenic source of chloride. Furthermore, the modelled seasalt levels are very low, most likely unable to produce the mixing ratios of $ClNO_2$ observed by the CIMS. Figure S3 has been added to the supplementary.

Action: The following text has been added/amended to the model results analysis and acknowledges the AMS limitations and instead highlighting the further support to anthropogenic Cl from low levels of sea salt and correlation within the model analysis.

[revised manuscript text omitted]

There are important analytical details lacking in the manuscript. Although mixing ratios of HCl, Cl2, ClONO2, HOCl, ClO, and OClO are reported, no information is provided on calibrations for these molecules. These must be included. Calibrations are not reported for any compounds in the particle phase measurements using the FIGAERO inlet. Despite this, and their admission that the observations could be explained by a sampling artifact, a quantification of "particle to gas phase partitioning" for $ClNO_2$ is reported (page 9, lines 26-27). The uncertain nature of this observation is consistent with a statement (page 9, lines 23-24) "this suggests the possible presence of $ClNO_2$ in the particle phase", but inconsistent with a later statement (page 9, lines 30-31) "these data indicates a significant amount of the chlorine associated with $ClNO_2$ is not liberated from the particle phase." In order to report this data in the text and figures, filter spike and recovery tests and gas-phase $ClNO_2$ filter sorption tests must be undertaken. Without this analytical rigor, the data is speculative. Similarly, CMBO in the particle phase is reported in Figure 9 (although not explicitly discussed in the text). Considering these particle observations are listed as a major novel finding of this work, they must be clearly justified.

Response: The inorganic halogens, many of which has no stable production method, are attributed the same sensitivity as of $ClNO_2$ which is a commonly used method. For example, Le Breton et al 2017 showed that inorganic halides have a similar sensitivity. Furthermore, the comparable sensitivity for chloroacetic acid and $ClNO_2$ emphasis a similar sensitivity for chloride containing species when applying the iodide ionisation.

As specific sensitivity for compounds being evaporated and subsequently ionised in the gas-phase after entering the same IMR would be redundant. However, one may argue that the efficiency of desorption could be quantified. Here the particle phase $ClNO_2$ is attributed to a possible complex within the ambient aerosol. We claim here that we observe higher concentrations in the particle phase compared to the Henrys Law constant which would suggests that the $ClNO_2$ is trapped within a matrix, rather than unable to partition upon heating, i.e. our conclusion would not be effected by a less than unity

effectivity of desorption. This would be the same for CMBO, which we quantify using the sensitivity for chloroacetic acid in the gas phase. This kind of calibration procedures has been performed in previous CIMS work.

Action: A clearer explanation of our application on sensitivities has been added to the manuscript and a validation to the approach with relevant citations. It is now stated that the gas phase sensitivities applies to the particle phase data also. The text regarding particle phase $ClNO_2$ has been made more clear to state that we observe higher concentrations in the particle phase due to a matrix effect rather than limitation of liberation from the particle phase during desorption. CMBO is also described to be quantified using the sensitivity for chloracetic acid which applies for both particle and gas phase.

"These sensitivities for $N_2O_5$ and $ClNO_2$ (9.8 and 1.6 ion counts per ppt $Hz^{-1}$ for $1x10^5$ iodide ion counts) were applied relatively to that of formic acid. The other inorganic halogens reported in this work were given the same sensitivity as of $ClNO_2$, as Le Breton et al. (2017) reported many inorganic halogens possess a similar, if not the same, sensitivity. This was further supported by our chloroacetic acid calibration. Other acids identified by CIMS which are reported in the literature are given the sensitivity of $N_2O_5$ to provide a minimum concentration so no concentrations are over estimated.

A post campaign calibration of chloroacetatic acid (99%, Sigma Aldrich) was utilised to apply a sensitivity factor for all Cl-VOCs measured during the campaign. The calibration was performed using the same method as for formic acid and gave a sensitivity of 1.02 ion counts $ppt^{-1}$ Hz when normalized to $1x10^5$ I- ion counts. Using relative sensitivities will increase the uncertainties, but is a commonly applied method within the CIMS community, although in this specific case it is very likely that the sensitivity is similar for all inorganic/organic halogens, as demonstrated by Le Breton et al. (2017a)."

Throughout the manuscript (see specific examples below), the authors have not properly considered the full literature in their discussion.

Response: The manuscript has been addressed as proposed and the literature discussion has been expanded

Action: The manuscript has been addressed as proposed and the literature discussion has been expanded

The manuscript contains numerous grammatical errors and should be carefully proofread.

Response: The manuscript has now been "cleaned" of grammatical and typographical errors to make it more clear and concise

Action: The manuscript has now been "cleaned" of grammatical and typographical errors to make it more clear and concise

Throughout the manuscript, "mixing ratio" and "concentration" are used interchangeably when discussing gas-phase measurements. All instances of "concentration" should be changed to "mixing ratio" (e.g. page 4, line 3 and page 12, line 26).

Response: This has now been corrected

Action: Concentration has been replaced with "mixing ratio" where necessary

Page 2, lines 15-31. The way this is presented, it appears as though the dominant fate of chlorine atoms is reaction with inorganics. In most cases, reactions with organics will be far more important.

Response: This should be rewritten so the reaction pathways are presented correctly

Action: The following text has been written to clearly state that the dominant loss of the chloride atom is via reaction with VOCs "The liberated chlorine will predominantly react with VOCs, with the above pathways representing alternative routes to loss of the chloride atom, and contribute to daytime photochemical oxidation, competing with OH and perturbing standard organic peroxy radical abundance (ROx = OH + HO$_2$ + RO$_2$), O$_3$ production rate, NO$_x$ lifetime and partitioning between reactive forms of nitrogen (Riedel et al., 2014)."

Page 4, line 15. IUPAC prefers the term "resolving power" ("resolution" is used to describe another quantity). The m/z must also be defined for the given resolving power. This information should also be reported in Section 2.2.

Response: Resolution has been changed to resolving power and is now reported in section 2.2.

Action: Resolution has been changed to resolving power and is now reported in section 2.2.

Page 5, lines 30-34. The sentence starting with "In this calibration. . ." is repeated

Response: This is correct and should not be repeated

Action: This repetition has now been removed

Page 6, line 20. "Mass" should be "m/z".

Response:  This is correct and will be replaced

Action: "Mass" is now "m/z"

Page 6, line 35. Is there any trend in the measurement discrepancies with RH?

Response: There is no trend observed with RH

Action: This observation has been noted in the text

Page 6, lines 33-36. These two sentences appear to give different numbers to describe the same results. It is confusing.

Response: The gradient of the fit is described in the second sentence whereas the R correlation is described in the first

Action: no changes have been made

Page 9, line 36. Typo in "photolytically"

Response: This typographic error is to be changed

Action: It is now spelt "photolytically"

Page 10, line 1. "Kim et al." is missing from the references section.

Response: Kim et al must be added to the References

Action: This reference has been added

Page 10, lines 19-23. Equations 2, 3, 5, and 6 are not balanced.

Response: This is correct and should be amended

Action: The reactions are now balanced

Page 11, lines 9-12. Budgets of chlorine atoms are also available for Los Angeles (Riedel et al., 2012; Young et al., 2014).

Response: These two manuscripts will be discussed in the text

Action: These manuscripts have been cited and referenced and it is acknowledged that measurements in urban Los Angeles yield similar production rates of the chloride radical to Beijing, although it is noted that higher concentrations of HCl observed in these measurements contribute significantly to the radical production where as in this paper we see little HCl contribution

Page 11, lines 23-24. The authors say "a number of studies have deemed chlorine atom chemistry to be insignificant with respect to O3 production and competing VOC oxidation to OH" and cite a single study to justify that a "significantly different approach is needed to assessing oxidation chemistry and photochemical smog in Asia". In fact, many studies examining this issue globally have shown a demonstrable impact of chlorine atoms on oxidation chemistry (e.g. (Osthoff et al., 2008; Riedel et al.,

2014; Sarwar et al., 2014)). This suggests similar techniques can be applied to understand Asian air quality. It is also not clear what the nature of the "significantly different approach" suggested by the authors might be.

Response: We agree with the reviewer that this section is to be rephrased and the conclusion is not fully supported by the text

Action: The text now reads "Although several studies have demonstrated a non-negligible impact of chlorine oxidation chemistry (e.g. Oshoff et al., 2008, Riedel et al., 2014 and Sarwar et al., 2014), the impact of Cl chemistry varies significantly between various areas and atmospheric conditions, e.g. Bannan et al., 2015, 2017 deemed the impact from chlorine atom chemistry to be relatively low with respect to $O_3$ production and competing with OH radicals for VOC oxidation"

Page 11, line 29. The authors mention that steady-state OH was calculated. More details are needed here. What measurements were included in this calculation? How were those measurements made? Page 11, lines 29-33. Calculated chlorine atom to measured OH concentrations are available for Los Angeles (Young et al., 2014).

Response: Reviewer 2 also had some comments about the steady state calculations, to which will also be addressed here to provide a complete picture to the reviewer of how this section has been amended and improved.

The reviewers are correct that the reaction of OH+ HCl was accidentally omitted from the text in the original manuscript. The section has now been corrected for that omission.

The reviewers are also correct that only using the small subset of hydrocarbons measured by the PTRMS would result in significant errors in estimated steady state Cl atom concentrations. However, we did not just use PTRMS measurements. In previous work (Bannan et al., 2015) we have shown that it is possible to calculate Cl atom concentrations, using a simple steady-state expression with the Cl atom production rate estimated from the observed loss rate of $ClNO_2$ and removal of Cl atoms via reaction with the VOC concentrations supplemented with data from the Boston tailpipe study (AQIRP, 1995) and LA VOC study (Fraser et al., 1997), i.e. missing VOC concentrations are estimated simply by using the ratio of measured VOCs and missing VOCs from these urban studies and measured VOC data in this study. The removal of Cl atoms via reaction with VOCs can then be determined using Eq. 1-3, and NIST kinetic data (Manion et al., 2014)

$$-d[alkanes]/dt = [X]\sum_i k_{X+alkane,i}[alkane,i]$$

(Eq 1)

$$-d[alkenes]/dt = [X]\sum_i k_{X+alkene,i}[alkene,i]$$

(Eq 2)

$$-d[alkynes]/dt = [X]\sum_i k_{X+alkyne,i}[alkyne,i]$$

(Eq 3)

Here the simple steady state approach agreed well with the MCM, despite a much more simplistic approach. We used an identical approach in this work utilizing the tailpipe VOC concentrations (AQIRP, 1995). However, we further simplify the approach by using one term $CH_4$ equivalent which accounts for relative concentration and reactivity towards Cl, i.e. if a VOC reacts 1000 times faster it is the equivalent of 1000 $CH_4$ or more formally for each VOC its $CH_4$ equivalent is kCl + VOC [VOC]/ kCl + CH4. Whilst the approach is a simplification it has been shown that using these emissions it is possible to estimate the Cl atom production in a Megacity environment and produces results that are comparable with much more thorough modelling approach using e.g. the MCM. It also generates a metric, CH4 equivalent, which can be used as a comparative measurement from city to city.

Photolysis rates were measured by a spectradiometer for O3, NO2, HCHO, HONO and H2O2. The photolysis rate of any given species was calculated by normalizing to the cross section and quantum yields taken from the recommendations of the Jet Propulsion Laboratory (JPL) kinetic evaluation report (Burkholder et al., 2015).

The reviewer is correct that Cl atoms will produce peroxy radicals which can perturb HOx/NOx cycles, and we were only considering the initial oxidation of VOCs. The text has been corrected to reflect that.

Action: the following text has been added to the manuscript and text has been added to the supplementary

"

"Here, a simple steady state calculation will be used to determine the Cl atom mixing ratio summarised below, but detailed within the supplementary:

$Cl_2 + h\nu \rightarrow Cl + Cl$       (1)

$ClNO_2 + h\nu \rightarrow Cl + NO_2$       (2)

$ClONO_2 + h\nu \rightarrow ClO + NO_2$       (3)

$HOCl + h\nu \rightarrow OH + Cl$       (4)

$OClO + h\nu \rightarrow O + ClO$       (5)

$OH + HCl \rightarrow Cl + H_2O$       (6)

$Cl + O_3 \rightarrow ClO + O_2$       (7)

$$Cl + CH_4 \text{ equivalent} \rightarrow HCl + \text{products} \qquad (8)$$

$$[Cl]_{SS} = \{2J1[Cl2] + J2[ClNO2] + J3[ClONO2] + J4[HOCl] + J5[OClO] + k7 [OH][HCl]\} / \{k7[O3] + k8[CH4] \text{ equivalent}\} \quad (9)$$

Where [CH4] equivalent represents the reactive VOC present as if it were reacting as CH4

Bannan et al., (2005), were able to use this steady state approach to compare the relative loss via reaction with OH compared with Cl atoms. Although this approach is an estimation, it was shown to produce comparable to results with that of the more rigorous MCM approach."

Page 14, lines 17-20. In the final paragraph of the paper, the authors claim that "chlorine atom chemistry may be under represented within models due to the lack of quantification and identification of particulate Cl-VOC products. This work provides instrumental capability to probe the competition between OH and Cl oxidation chemistry and quantify the SOA yields as a result of both pathways." This paper does not demonstrate quantitative measurement of particulate Cl-VOCs, as no calibrations or desorption efficiencies have been presented or discussed. Furthermore, the authors have not sufficiently shown that particulate Cl-VOCs are necessary to understand the relative impacts of chlorine atoms and OH on air quality.

Response: The final paragraph has been clarified to state that these results highlight the deficiency in chlorine atom chemistry within models which could be a result of lack of quantification and identification of Cl VOCs in the gas and particle phase. The work here provides instrument capability to both identify and quantify the Cl VOCs in both the particle and gas phase which were shown in the steady state calculations to be important contributors to daytime oxidation. We believe the calibration of chloroacetic acid can be used for indirect quantification of Cl-VOCs and where the ability to measure the precursors to chloride radical production does demonstrate the ability to quantify the chloride radical budget and particulate Cl-VOCs.

Action: The text has been amended and reads as follows "The results highlight deficiency in chlorine atom chemistry descriptions within models; possibly due to a lack in quantification and identification of Cl-VOC products in gas and particle phase. This work provides instrumental capability to probe the competition between OH and Cl oxidation chemistry and quantify their effect on ozone and SOA formation."

Figures would be more clear if panels were labelled with (A), (B), etc. Axis labels are often missing or unclear. For example, in Figure 1, presumably all the x-axes should be "m/z" and in Figure 5, the right-hand y-axis of the top left graph is unlabelled.

Response: These plots will be made clearer

Action: Axis have been labelled and the panels are now marked A, B, C etc

Page 23, lines 2-6. In the Figure 7 caption, it would be helpful to be explicit that terms are calculated and not measured.

Response: This is true and will be amended to state they are not measured values

Action: The caption now states these are steady state values

**Reviewer 2**

The manuscript suffers from organizational issues. Reactions are not consecutively numbered, sections were skipped, etc.

Response: These are to be addressed

Action: Sections are now sequential and reaction numbers are ordered appropriately

Mixing ratios of a variety of trace gases, including HCl, Cl2, ClONO2, HOCl, OClO and ClO as well as CMBO, isoprene, IOPEX, and benzene as well photolysis frequencies are presented but it is unclear in many cases how these data were acquired or how instrumental response factors were determined.

Response: The inorganic halogens have been given the same sensitivity to that of $ClNO_2$. It has been shown by Le Breton et al 2017 that the inorganic halogens have a high and similar, if not the same, sensitivity to iodide ionisation. The acids presented are given the highest sensitivity (that of N2O5) to provide a minimum mixing ratio. As stated in the text, due to reduced availability of Cl-VOCs for laboratory calibration, CMBO and other Cl VOCs are given the same sensitivity as of chloroacetic acid, which is similar to that for $ClNO_2$ which supports literature findings that inorganic halogens/functional groups possess a similar sensitivity.

Detailed information about the PTR MS measurements can be found in Yuan et al 2012 and 2013. In brief, 28 masses are measured for the campaign at 1 Hz. Zero air, which was produced by ambient air passing through a platinum catalytic converter at 350 ⬚C (Shimadzu Inc., Japan), was measured for 15 min every 2.5 hours to determine the background. used to measure background Aromatics masses (m/z 79 for benzene, m/z 93 for toluene, m/z 105 for styrene, m/z 107 for C8 aromatics and m/z 121 for C9 aromatics), oxygenated masses (m/z 33 for methanol, m/z 45 for acetaldehyde, m/z 59 for acetone, m/z 71 for MVK+MACR and m/z 73 for MEK), isoprene (m/z 69) and acetonitrile (m/z 42) were calibrated by using EPA TO15 standard from Apel-Riemer Environmental Inc., USA. Formic acid (m/z

47), acetic acid (m/z 61), formaldehyde (m/z 31), and monoterpenes (m/z 81 and m/z 137) were calibrated by permeation tubes (VICI, USA).

Photolysis rates were measured by a spectradiometer for O3, NO2, HCHO, HONO and H2O2. The photolysis rate of any given species was a given species was calculated by normalizing to the cross section and quantum yields taken from the recommendations of the Jet Propulsion Laboratory (JPL) kinetic evaluation report (Burkholder et al., 2015).

Action: The text regarding inorganic halogen quantification now states "The other inorganic halogens reported in this work are assumed to have the same sensitivity as ClNO$_2$. This is in line with that Le Breton et al. (2017) reported many inorganic halogens possess a similar, if not the same, sensitivity, which is also supported by our chloroacetic acid calibration."

The text states for other acids "Other acids identified by CIMS which are reported in the literature are given the sensitivity of N$_2$O$_5$ to provide a minimum concentration so no concentrations are over estimated."

The text states for photolysis rates "Photolysis rates were measured by a spectradiometer for O3, NO2, HCHO, HONO and H2O2. The photolysis rate of any given species was calculated by normalizing to the cross section and quantum yields taken from the recommendations of the Jet Propulsion Laboratory (JPL) kinetic evaluation report (Burkholder et al., 2015).

The text details the PTR calibration in the calibration section

"Detailed information about the PTR MS measurements can be found in Yuan et al 2012 and 2013. In brief, 28 masses are measured for the campaign at 1 Hz. Zero air, which was produced by ambient air passing through a platinum catalytic converter at 350 °C (Shimadzu Inc., Japan), was measured for 15 min every 2.5 hours to determine the background. used to measure background Aromatics masses (m/z 79 for benzene, m/z 93 for toluene, m/z 105 for styrene, m/z 107 for C8 aromatics and m/z 121 for C9 aromatics), oxygenated masses (m/z 33 for methanol, m/z 45 for acetaldehyde, m/z 59 for acetone, m/z 71 for MVK+MACR and m/z 73 for MEK), isoprene (m/z 69) and acetonitrile (m/z 42) were calibrated by using EPA TO15 standard from Apel-Riemer Environmental Inc., USA. Formic acid (m/z 47), acetic acid (m/z 61), formaldehyde (m/z 31), and monoterpenes (m/z 81 and m/z 137) were calibrated by permeation tubes (VICI, USA).

- Furthermore, concentrations of Cl and OH were calculated on the basis of steady state assumptions. These calculations are questionable since the only VOC measurements were by PTR-MS, an instrument that quantifies many but not all VOCs. Crucially, a PTR-MS usually does not quantify alkanes, whose abundances are important sinks for Cl atoms

Response: Reviewers 1 and 2 comments regarding the steady state calculations have been collected and answered below.

The reviewers are correct that the reaction of OH+ HCl was accidentally omitted from the text in the original manuscript. The section has now been corrected for that omission.

The reviewers are also correct that only using the small subset of hydrocarbons measured by the PTRMS would result in significant errors in estimated steady state Cl atom concentrations. However, we did not just use PTRMS measurements. In previous work (Bannan et al., 2015) we have shown that it is possible to calculate Cl atom concentrations, using a simple steady-state expression with the Cl atom production rate estimated from the observed loss rate of ClNO2 and removal of Cl atoms via reaction with the VOC concentrations supplemented with data from the Boston tailpipe study (AQIRP, 1995) and LA VOC study (Fraser et al., 1997), i.e. missing VOC concentrations are estimated simply by using the ratio of measured VOCs and missing VOCs from these urban studies and measured VOC data in this study. The removal of Cl atoms via reaction with VOCs can then be determined using Eq. 1-3, and NIST kinetic data (Manion et al., 2014)

$$-d[alkanes]/dt = [X]\sum_i k_{X+alkane,i}[alkane,i]$$

(Eq 1)

$$-d[alkenes]/dt = [X]\sum_i k_{X+alkene,i}[alkene,i]$$

(Eq 2)

$$-d[alkynes]/dt = [X]\sum_i k_{X+alkyne,i}[alkyne,i]$$

(Eq 3)

We were able to show that the simple state approach agreed well with the MCM, despite a much more simplistic approach. We used an identical approach in this work utilizing the tailpipe VOC concentrations (AQIRP, 1995). However, we further simplify the approach by using one term CH4 equivalent which accounts for relative concentration and reactivity towards Cl, i.e. if a VOC reacts 1000 times faster it is the equivalent of 1000 CH4 or more formally for each VOC its CH4 equivalent is kCl + VOC [VOC]/ kCl + CH4. Whilst the approach is a simplification of course, it has been shown that using these emissions it is possible to estimate the Cl atom production in a Megacity environment and produces results that are comparable with the much more thorough modelling approach of the MCM. It also generates a metric, CH4 equivalent, which can be used as a comparative measurement from city to city.

Photolysis rates were measured by a spectradiometer for O3, NO2, HCHO, HONO and H2O2. The photolysis rate of any given species was calculated by normalizing to the cross section and quantum

yields taken from the recommendations of the Jet Propulsion Laboratory (JPL) kinetic evaluation report (Burkholder et al., 2015).

The reviewer is correct that Cl atoms will produce peroxy radicals which can perturb HOx/NOx cycles, and we were only considering the initial oxidation of VOCs. The text has been corrected to reflect that.

Action: the following text has been added to the manuscript and text has been added to the supplementary

"Here, a simple steady state calculation will be used to determine the Cl atom mixing ratio summarised below, but detailed within the supplementary:

$$Cl2 + hv \rightarrow Cl + Cl \quad\quad\quad\quad (1)$$

$$ClNO2 + hv \rightarrow Cl + NO2 \quad\quad\quad (2)$$

$$ClONO2 + hv \rightarrow ClO + NO2 \quad\quad (3)$$

$$HOCl + hv \rightarrow OH + Cl \quad\quad\quad\quad (4)$$

$$OClO + hv \rightarrow O + ClO \quad\quad\quad\quad (5)$$

$$OH + HCl \rightarrow Cl + H2O \quad\quad\quad\quad (6)$$

$$Cl + O3 \rightarrow ClO \;\; + O2 \quad\quad\quad\quad (7)$$

$$Cl + CH4 \text{ equivalent} \rightarrow HCl + products \quad (8)$$

[Cl]SS = {2J1[Cl2] + J2[ClNO2] + J3[ClONO2] + J4[HOCl] +J5[OClO] +k7 [OH][HCl]} / {k7[O3] + k8[CH4] equivalent} (9)

Where [CH4] equivalent represents the reactive VOC present as if it were equivalent CH4

Bannan et al., (2105), were able to use this steady state approach to compare the relative loss via reaction with OH compared with Cl atoms. Although this approach is an estimation, it was shown to produce comparable to results with that of the more rigorous MCM approach."

Some data (e.g., OClO, CMBO, ClO, ClONO2, IOPEX) are only semi-quantitative and should be presented as such.

Response: Many CIMS papers refer to relative calibrations. Here we apply this method to non-calibrated compounds but do apply a maximum sensitivity to limit their impact on model calculations. Their semi quantitative nature will be noted within the text.

Action: The compounds which are not directly are noted in the calibration section to ensure the reader is aware that the values are semi quantitative.

pg 1 line 27 –replace the comma with "and"

Response: This has been altered

Action: the comma has been replaced with "and"

line 29 – ppt is not a concentration unit – please rephrase

Response: This will be changed (we have changed concentrations to mixing ratios)

Action: The units now used are pptV and referred to as mixing ratios and referred to as mixing ratios

pg 2 lines 17 – (O3, HOx, and NOx levels via ... (R1-R9)). Most of the Cl will likely abstract hydrogens from hydrocarbons (R11), in particular at this site. Another important reaction omitted here is OH+HCl->H2O+Cl. Consider reorganizing the introduction to reflect this.

Response: We agree that the OH + HCl reaction must be included within the reactions presented and noted in the text. The hydrogen abstraction to form HCl is within the text and reaction list, although we have no further iterated to the reader that this is the major reaction pathway for chloride-VOC reactions.

Action: The reactions listed (now R1-R11) contain the OH + HCl reaction. We have amended the text to state hydrogen abstraction is the dominant pathway for chloride oxidation

"The oxidation mechanism of saturated hydrocarbon (R12-R13) is initiated by reaction with OH or chlorine atom to form an organic peroxy radical (RO2), and $H_2O$ or HCl depending on the oxidant, which is the dominant pathway for chloride-VOC reactions."

pg 3 lines 2-4. –"This perturbation is currently thought to only be significant in the early hours of the day while OH concentrations are low and chlorine atom production is high through the photolysis of ClNO2."

I don't think this is correct. Reaction of Cl with alkanes produces peroxy radicals, which feed into the "regular" HOx/NOx cycles. Thus, the early morning injection of radicals impacts (perturbs) radical chemistry for the remainder of the day. Perhaps the authors meant to say "Oxidation of VOCs by Cl is currently thought to be ..."?

Response: This reviewer is correct. Text has been added to clarify this point and that we only consider the initial oxidation of VOCs

Action: The following text has been added "The oxidation of VOCs by chlorine atoms is thought to be significant in the early hours of the day while OH mixing ratio are low and chlorine atom production is

high through the photolysis of ClNO₂, as well as feeding into the standard HOx/NOx cycles via production of peroxy radicals from reactions with alkanes."

lines 11/12 – there are two reactions labelled R11

Response: This is correct and should be changed

Action: Reactions here are now R11 to R14

line 31 – "36%" - please add the value for SOA yield from OH initiated oxidation of isoprene for comparison.

Response: The Liu et al 2016 has been cited which calculates a yield of 15% for comparison.

Action: The Liu et al 2016 has been cited which calculates a yield of 15% for comparison, although this is known to be a factor of 2 higher than used in standard climate models.

pg 4 line 24 – how were photolysis rates determined?

Photolysis rates were measured by a spectradiometer for O3, NO2, HCHO, HONO and H2O2. The photolysis rate of any given species was a given species was calculated by normalizing to the cross section and quantum yields taken from the recommendations of the Jet Propulsion Laboratory (JPL) kinetic evaluation report (Burkholder et al., 2015).

Action: The text now states "Photolysis rates were measured by a spectradiometer for O3, NO2, HCHO, HONO and H2O2. Inorganic halogen photolysis rates were extracted relatively from these J rates."

line 31 – there are two Le Breton et al. 2017 references. Please label them 2017a and 2017b.

Response: These references have been noted as 2017a and 2017b in the text and reference list

Action: These references have been noted as 2017a and 2017b in the text and reference list

pg 5 lines 1-2 Please provide more detail as to how the PTR-MS was calibrated, what molecules were quantified, etc

Response: Detailed information about the PTR MS measurements can be found in Yuan et al 2012 and 2013. In brief, 28 masses are measured for the campaign at 1 Hz. Zero air, which was produced by ambient air passing through a platinum catalytic converter at 350 ⬚C (Shimadzu Inc., Japan), was measured for 15 min every 2.5 hours to determine the background. used to measure background Aromatics masses (m/z 79 for benzene, m/z 93 for toluene, m/z 105 for styrene, m/z 107 for C8 aromatics and m/z 121 for C9 aromatics), oxygenated masses (m/z 33 for methanol, m/z 45 for acetaldehyde, m/z 59 for acetone, m/z 71 for MVK+MACR and m/z 73 for MEK), isoprene (m/z 69) and

acetonitrile (m/z 42) were calibrated by using EPA TO15 standard from Apel-Riemer Environmental Inc., USA. Formic acid (m/z 47), acetic acid (m/z 61), formaldehyde (m/z 31), and monoterpenes (m/z 81 and m/z 137) were calibrated by permeation tubes (VICI, USA).

Action: The text now states in section 2.1 "Detailed information about the PTR MS measurements can be found in Yuan et al 2012 and 2013. In brief, 28 masses are measured for the campaign at 1 Hz. Zero air, which was produced by ambient air passing through a platinum catalytic converter at 350 °C (Shimadzu Inc., Japan), was measured for 15 min every 2.5 hours to determine the background. used to measure background Aromatics masses (m/z 79 for benzene, m/z 93 for toluene, m/z 105 for styrene, m/z 107 for C8 aromatics and m/z 121 for C9 aromatics), oxygenated masses (m/z 33 for methanol, m/z 45 for acetaldehyde, m/z 59 for acetone, m/z 71 for MVK+MACR and m/z 73 for MEK), isoprene (m/z 69) and acetonitrile (m/z 42) were calibrated by using EPA TO15 standard from Apel-Riemer Environmental Inc., USA. Formic acid (m/z 47), acetic acid (m/z 61), formaldehyde (m/z 31), and monoterpenes (m/z 81 and m/z 137) were calibrated by permeation tubes (VICI, USA)."

line 17 - section 2.3 is absent.

Response: This is correct and will be amended

Action: Calibration is now section 2.3

Please describe how the response factors for HCl, Cl2, ClONO2, HOCl, OClO and ClO (Figure 3) were determined

Response: As described above, these were given the highest sensitivity to limit their impact on the modelling results

Action: This has now been described in the text more clearly

lines 24-25 – "The $N_2O_5$ diffusion source was held at a constant temperature (-23 C), and the mass loss rate was characterized gravimetrically for a flow rate of 100 sccm."

$N_2O_5$ is quite hygroscopic, such that the diffusion source could "gain weight" simply by absorbing residual moisture. Another potential error with this method is "loss" of NO3 (e.g., through reaction with impurities on the wall) followed by loss of NO2 (toward which the CIMS is probably blind). All this probably doesn't matter since there was a CEAS on site.

Response: The technique has been described by Faxon *et al* 2017 in full detail and the CEAS measurements confirm the accuracy of the calibration technique

Action: The technique has been described by Faxon et al 2017 in full detail and the CEAS measurements confirm the accuracy of the calibration technique

How stable/accurate/reproducible is this source? Could it be used as standalone $N_2O_5$ calibration method?

Response: The sources stability over time has not been thoroughly tested for longer than a week, upon stable cooling I believe it can be used as a reproducible source. The CEAS utilised a different calibration technique as the N2O5 CIMS calibration was performed post campaign. The CEAS was separately calibrated as detailed in Wang et al 2017a "Development of a portable cavity-enhanced absorption spectrometer for the measurement of ambient NO3 and N2O5: experimental setup, lab characterizations, and field applications in a polluted urban environment" A brief description of the CEAS calibration has been added, stating that the mirror was calibrated for daily and the filter replaced hourly "The CEAS utilised a dynamic source by mixing NO2 and O3 to generate stable N2O5 for calibration (Wang et al., 2017). The source was used to calibrate the ambient sampling loss of N2O5 in the sampling line, filter, the preheater cavity and optical cavity. This was performed pre and post campaign. During the campaign the reflectivity of the high reflectivity mirror was calibrated daily and filter changed hourly."

Action: The following text was added "The CEAS utilised a dynamic source by mixing $NO_2$ and $O_3$ to generate stable $N_2O_5$ for calibration (Wang et al., 2017). The source was used to calibrate the ambient sampling loss of $N_2O_5$ in the sampling line, filter, the preheater cavity and optical cavity. This was performed pre and post campaign. During the campaign the reflectivity of the high reflectivity mirror was calibrated daily and filter changed hourly."

Please state if the diffusion source method has been verified using CEAS (which I assume it has).

Response: It has not been verified with the CEAS, but has with the CIMS in previous literature (Faxon et al., 2017) and the good agreement here between the CEAS and CIMS illustrate its ability.

Action: As stated above, the instruments were not calibrated using the same source, but independently calibrated. The above text was also added.

Line 36 – "these sensitivities" – please state the instrumental response factors here

Response: The sensitivities have been added

Action: The following text has been amended "These sensitivities for $N_2O_5$ and $ClNO_2$ (9.8 and 1.6 ion counts per ppt Hz-1 for $1x10^5$ iodide ion counts) were applied relatively to that of formic acid."

pg 6 line 20/21 – "A quadrupole CIMS may not be able to resolve the peak adjacent to ClO at mass 178 and the second dominant peak for the $ClNO_2$ fit would result in a 10% over estimation."

Please clearly state what ions are present at mass 178.

Response: The peaks have been stated and are now included in figure 1

Action: The peaks have been stated and are now included in figure 1

It is unclear what is meant by "second dominant peak for the $ClNO_2$ fit" – is this at m/z 208?

Response: There is a second peak fitted (as shown in figure 1) which is identified as a cluster of nitric acid with water forming an adduct with iodide that will contribute to up to 10% of the counts at this unit mass

Action: This identification has been stated in the text and provided in figure 1

line 32 – please put the N2O5 mixing ratios in context – (temperature, O3 and NO2 levels, NO3 production rate etc.)

Response: The Wang et al 2018 paper focuses on analysis of N2O5 from the CEAS utilising the CIMS ClNO2 data. We therefore believe that reporting the typical concentrations and diurnal trends suffice here as the focus is more on the inter-comparison rather than production rates of N2O5 and ClNO2. This information can be added if requested by the editor, although we believe it does not contribute to the manuscript.

Action: NA

line 33 – Were the instruments operated on the same inlet? If not, there may be scatter simply from sampling air at slightly different locations.

Response: They were not on the same inlet and faced different directions

Action: This is now stated in the text

line 36 – The offset should have units of ppt

Response: this has been amended

Action: this has bene amended

pg 7 line 2 – "although averaging at 4 ppt" perhaps better to give a relative error here

Response: The average error has been reported now in the text, 11%.

Action: The text now reads "The largest error between the two measurements occurs at night during the higher levels of $N_2O_5$, although averaging at 4 ppt (representing 11% error on the average campaign concentration)."

line 3 – please move details on how instruments were operated (heated IMR) to section 2.2

Response: The details of the heated IMR have been added to section 2.2, although this section still refers to this setup to clarify the possible physical differences resulting in variation of the measured mixing ratio.

Action: Section 2.2 now reads "The ionized gas was then carried out of the ion source and into the Ion-Molecule Reaction (IMR) chamber, which was heated to 40 degrees Celsius to reduce wall loss, through an orifice ($\varnothing$ = 1 μm)."

line 21- "Inorganic chlorine abundance and profiles". There is a lot presented in this section, BB, WRF etc, that goes well beyond inorganic chlorine abundances and profiles. This section should be broken up into smaller, more coherent pieces.

Response: We agree with the reviewer and have now renamed this section and added two sub sections

Action: This section is now named "3.3 Inorganic chlorine: Abundance, profiles and source"

We have added section 3.3.1 called "Abundance and profiles" and section 3.3.2 called "Source of chloride"

line 24 – mixing ratio, not concentration

Response: This is to be changed

Action: mixing ratio is now used instead of concentration

line 25 – "σ 270 ppt" is this standard deviation?

Response: yes

Action: the text now states "510 ppt (standard deviation (σ) 270 ppt)"

pg 8 line 9. Please comment on the possibility of chlorine nitrate forming on the inner walls of the inlet.

Response: text has been added to hypothesise this as a possibility to the reader

Action: The following text has been added "IMR chemistry is also not a possible source as these reactions would occur throughout the day, therefore skewing all of the data and not just the night-time

levels, although there is a possibility that ClONO2 can be formed in the IMR by reactions between ClO and NO2."

pg 8 line19. –" This suggests the chlorine has an anthropogenic source and not marine" I disagree.

One has to be careful with the interpretation of AMS data. The "standard" AMS chloride product only includes non-refractory aerosol, i.e., does not include sea salt chloride – for one, it does a poor job volatilizing NaCl, and most AMS have a size cut off of 1 micron that filters out most of the larger sea salt aerosol particles. The correlation of AMS chloride with anthropogenic tracers may arise from acid displacement of sea salt chloride in polluted air (that is high in SO2). I'd suggest rewording the entire paragraph (lines 14-27). I don't doubt that anthropogenic Cl sources contribute, but there aren't enough data (shown in this paper) to proof a negligible marine influence.

Response: The response here is similar to the first comment by reviewer 1. We acknowledge the limitations of the AMS data and will provide that information to the reader. We ran the WRF model simulation to observe how the sea salt model results correlate with ClNO2 profiles. This resulted in a poor correlation (figure S4) whereas total CO, as stated in the manuscript, had a relatively good correlation with ClNO2, further supporting the anthropogenic source of chloride. Furthermore, the modelled sea salt levels are very low, most likely unable to produce the mixing ratios of ClNO2 observed by the CIMS. Figure S4 has been added to the supplementary.

Action: Section 3.3.2 has been amended as displayed below to acknowledge the AMS limitations and further show model runs indicating no correlation between sea salt and ClNO$_2$

[revised manuscript text omitted]

line 28 - Please describe the WRF model in the methods section, not in the results section.

Response: The model description has bene moved to section 2.3 "model setup"

Action: A new section (2.3) has been populated with the model description.

I'd remove the WRF simulations as they may not account for local BB – chemical tracers would be more robust.

Response: We have now utilised the WRF simulations further to support the hypothesis that the $ClNO_2$ measured cannot be a product of only seasalt particle heterogeneous reactions and anthropogenic chloride must play a significant role

Action: NA

pg 9 line 13. All that is shown is that WRF modeling suggests BB to be a small source of chlorine – it doesn't show industrial emissions. Please rephrase.

Response: We agree this is too strong a claim and have rephrased the sentence

Action: The sentence now reads "The fact that mixing ratios of CObb is so much smaller than COt according to the model, combined with the much better correlation for COt than for CObb strongly suggests that industrial emissions are a more significant source of chlorine, rather than biomass burning."

line 14 The particle desorption profiles should be discussed in their own section.

Response: This has been placed into its own section

Action: Section 3.4 is now "particle phase $ClNO_2$" and the other sections following have been renumbered

lines 14-20. Did you observe the peak at 210? Please expand the AMU axis in Figure 5 to show it.

Response: Yes we did but again dominating in the particle phase

Action: The m/z axis has been expanded to 210.5

line 30 " these data indicates a significant amount of the chlorine associated with $ClNO_2$ is not liberated from the particle phase" it should be "these data indicate"

More to the point, you observe that you can drive off $ClNO_2$ if you heat aerosol. Have you considered that additional $ClNO_2$ could be formed by thermally driven reactions? If not, please state that this is a major assumption made here.

Response: This is a assumption to be stated

Action: This assumption is now stated

line 31-33 "The slope" please show this plot (perhaps as an insert in Figure 5).

Response: This is merely a ratio of the red and blue (gas and particle) time series in panel A of figure 5.

Action: We feel this does not need to be added as it is just utilisation of the data already displayed in the figure

Personally, I wouldn't call 5% "significant" considering this is much less than the measurement (calibration) error.

Response: The term significant has been replaced with "non-negligible"

Action: The term significant has been replaced with "non-negligible"

pg 10 line 18 The numbering of the reactions is inconsistent with those on pg 2. Some reactions are unnecessarily duplicated.

Response: The reactions have been renumbered and balanced

Action: The reactions have been renumbered and balanced

line 25 - Please number the steady state expression.

Response: This has been done

Action: The steady state expression is number (8)

A major source of Cl atom is the reaction OH + HCl -> H2O + Cl, which should not be omitted here. line 26 - And how was "equivalent CH4" determined for this site? It must be massive.

It is very likely that the PTR-MS misses most of it, for example all of the alkanes (Table 2 of de Gouw's Mass Spectrometry Reviews 26, 223 (2007)).

Response – the above 2 comments have been answered in the first question regarding the ss as mentioned within the response.

pg 11 line 9 – "The results show that both at the UK marine and urban site max chlorine atom concentrations are more than an order of magnitude lower than the mean of Beijing." Considering the uncertainty of the Cl atom sinks, the authors should only compare Cl atom production rates. Comparing rural and urban sites (Weybourne with Beijing) is like comparing apples and oranges. Many other groups have calculated Cl atom production rates from $ClNO_2$ photolysis, including many polluted urban sites. How do the numbers of this study stack up to these?

Response: We agree with the reviewer that other studies need to be considered in this section. We still believe that comparing urban and rural sites, or indeed urban and marine sites is an important factor as this paper postulates the impact of anthropogenic chloride and challenges the significance of sea salt as the major source for ClNO2 production. Therefore we would like to keep this comparison within the manuscript.

Action: The following text has bene added to consider previous calculations of chloride radical production.

"Studies of chloride atom production in Los Angeles by Riedel et al. (2012) and Young et al. (2014) indicate that the high production rate in Beijing is somewhat typical of urban sites, although HCl and $ClNO_2$ contribution to radical production is the same, whereas here we see very little chloride atom production from HCl in comparison to $ClNO_2$."

Imo, the entire section comparing OH and Cl abundances is questionable

Response: We have now added to this section to present the steady state in a more detailed manor and supported its utilisation referencing the Bannan et al paper which was compared against the MCM for OH radical concentrations. We therefore feel its role within the manuscript has been validated and adds to reliable scientific analysis of the dataset

Action: NA

pg 12 line 30 – "longer atmospheric lifetime" how long are the lifetimes of CMBO and of isoprene?

Response: The lifetime of isoprene is 1-2 hours according to Atkinson et al. (2000) , but we do not know the lifetime of CMBO

Action: Although this assumption is made based on the diurnal profiles, we agree we cannot assume the lifetime of CMBO is longer than that of isoprene and therefore have removed this phrase from the manuscript.

line 34 – "The concentrations of Cl and isoprene were relatively low" How low is relatively low? Please be quantitative.

Response: Quantitative values will be added t the text

Action: The exact mean values have been added "$1.6 \times 10^5$ molecules cm-3 s-1 Cl and 0.5 ppb isoprene"

If CMBO abundances did not follow those of its precursors, does that imply that CMBO can be primary (or originates from other precursors)?

Response: In response to the below comment, we now acknowledge in the manuscript that CMBO may not be unique to isoprene-chlorine reactions and therefore could have alternative sources

Action: We have added the following text to support the qualitative analysis of the CMBO time series "CMBO may also not be unique to only isoprene-chloride reactions and therefore have alternative sources not represented in this data set."

How certain are we that CMBO is a unique marker of chlorine-isoprene chemistry (line 25)?

Response: We agree with the reviewer that it may be a unique marker of chlorine chemistry, but not proven to be unique to isoprene reactions

Action: The sentence has been rephrased to state "unique marker of chlorine chemistry"

pg 13 line 36 " $ClNO_2$ was potentially identified in the particle phase "

I agree, but in the preceding text, $ClNO_2$ was not only identified but also quantified, or was it? Either way, the earlier section is inconsistent with the much more conservative conclusion in the end.

Response: The text states the observation and examines the possible causes which we do not categorically prove is only an instrument artefact therefore we state in the conclusion that it is indeed identified but at unusually high concentrations. We do not state it is quantified as the quantification in the gas phase is sufficient and the same sensitivity as for the particle phase.

Action: The text now reads "$ClNO_2$ was identified in the particle phase at higher ratios with respect to its gas phase component than expected, which may only prove to be significant at such elevated mixing ratios as observed in East Asia."

pg 14 – many references are incomplete (e.g., Pszenny et al.) and most are missing their doi.

Response: The reference has been checked and any inaccuracies or missing doi's have been added

Action: The reference has been checked and any inaccuracies or missing doi's have been added

pg 19 – Figure 1. Please identify the green, gold/yellow, and magenta lines. For the second panel, it would be useful to show a blank (zero) measurement also.

Response: These peaks have bene identified and labelled din the MS plot

Action: The peaks for ions C8H5NO4, C6F3HO3, C9H8NO3, IHNO3H2O and C9H5SO4 are now colour labelled in the MS plot

pg 20 – please define the "C" and "M" terms

Response: They will be defined

Action: C and M are now defined in the caption

pg 23 – Figure 7A or 7B – one of the "y" axes is mislabelled

Response: The labels are correctly labelled

Action: The labels are correctly labelled

*Correspondence to:* M. Le Breton (Michael.le.breton@gu.se) and S. Guo (guosong@pku.edu.cn)

**Abstract**. A Time of Flight Chemical Ionisation Mass spectrometer (CIMS) utilizing the Filter Inlet for Gas and AEROsol (FIGAERO) was deployed at a regional site 40 km north west of Beijing and successfully identified and measured 17 sulfur containing organics (SCOs = organo/nitrooxyorgano sulfates and sulfonates) with biogenic and anthropogenic precursors. The SCOs were quantified using laboratory synthesized standards of lactic acid sulfate and nitrophenol organosulfate (NP OS):). The variation in field observations werewas confirmed by comparison to offline measurement techniques (orbitrap and High performance Liquid Chromotography (HPLC)) using daily averages. The mean total (of the 17 identified by CIMS) SCO particle mass concentration was $210 \pm 110$ ng m$^{-3}$ and had a maximaum of 540 ng m$^{-3}$, although contributed to only $2 \pm 1$% of the organic aerosol (OA). The CIMS identified a persistent gas phase presence of SCOs in the ambient air, which was further supported by separate vapour pressure measurements of NP OS by a Knudsen Effusions Mass Spectrometer (KEMS). An increase in relative humidity (RH) promoted partitioning of SCO to the particle phase whereas higher temperatures favored higher gas phase concentrations. Biogenic emissions contributed to only 19% of total SCOs analysedmeasured in this study. Here $C_{10}H_{16}NSO_7$, a monoterpene derived SCO, representinged the highest fraction (10%) followed by an isoprene-derived SCO. The Anthropogenic SCOs with polycyclic aromatic hydrocarbon (PAH) and aromatic precursors dominated the SCO mass

loading (51%) with $C_{11}H_{11}SO_7$, derived from methyl napthalene oxidation, contributing to 40 ng m$^{-3}$ and 0.3% of the OA mass. Anthropogenic related SCOs correlated well with benzene, although their abundance depended highly on the photochemical age of the air mass, tracked using the ratio between pinonic acid and its oxidation product, acting as a qualitative photochemical clock. In addition to typical anthropogenic and biogenic precursor the biomass burning precursor nitrophenol (NP) provided a significant level of NP OS. It must be noted that the contribution analysis here is only representative of the SCOs detected, where there are likely to be many more SCOs present which the CIMS has not identified.

Gas and particle phase measurements of glycolic acid suggest that partitioning towards the particle phase promotes glycolic acid sulfate production, contrary to the current formation mechanism suggested in the literature. Furthermore, the $HSO_4.H_2SO_4^-$ cluster measured by the CIMS was utilized as a qualitative marker for acidity and indicates that the production of total SCOs is efficient in highly acidic aerosols with high $SO_4^{2-}$ and organic content. This dependency becomes more complex when observing individual SCOs due to variability of specific VOC precursors.

**1. Introduction**

Atmospheric particulate matter (PM) is known to play a major role in affecting climate and air quality leading to severe health issues, such as respiratory and cardiovascular degradation (Pope *et al.*, 2002; 2011; Kim *et al*., 2015). Secondary organic aerosols (SOA), formed through reactions of volatile organic compounds (VOCs) yielding semi volatile products that partition into the aerosol phase, represents a significant fraction of PM (Hallquist *et al*., 2009) and remains the most poorly understood PM source (Foley *et al*., 2010) due to the complexity of itstheir chemical nature, resulting in discrepancies between observations and models (Heald *et al*., 2005). Annual average PM$_1$ (particulate matter of diameter less than 1 micron) concentrations in Beijing reached 89.5 µg m$^{-3}$ in 2013 and, although recently dropped to 80.6 µg m$^3$, is still significantly above the Chinese National Ambient air quality Standard (CNAAQS, 35 µg m$^3$ annual average). The knowledge gap of PM primary emissions and secondary production limits scientifically based abatement strategies targeting effects of secondary pollution in highly polluted regions (Hallquist *et al.*, 2016; Zhang *et al.,* 2012a). Therefore, Beijing is an ideal case study region for intense measurement campaigns to increase our understanding of the sources and processes involved in atmospheric aerosol chemistry in megacities. A growing number of field studies in this region have been performed in recent years, specifically focused on the haze events investigating the composition of primary and secondary particle aerosols and their formation mechanisms (Guo *et al.,* 2012, 2014, 2013; Huang *et al*., 2010; Hu *et al.,* 2016, 2017; Li *et al.,* 2017).

Organosulfates (OSs), here part of sulfur containing organics (SCOs), are known important SOA components formed by reactions between reactive organic compounds and sulfate (Iinuma *et al.,* 2007; Surrattt *et al*., 2007, 2008), which is generated by the oxidation of SO$_2$, primarily emitted by fossil fuel combustion (Wuebbles and Jain, 2001). OSs have previously been measured in ambient aerosols at a number of geographical locations, from remote regions to highly

populated urban environments (Surratt *et al.*, 2007, 2008; Kristensen *et al.,* 2011; Stone *et al.*, 2012; Zhang *et al.*, 2012b; Worton *et al.*, 2013; Shalamzari *et al.*, 2013; Hansen *et al.*, 2014) although their composition and contribution to organic mass can vary significantly (Huang *et al.*, 2015). To date, several of their precursors are not known (e.g., Hansen *et al.*, 2014). Mechanistic studies reveal multiple possible pathways for SCO formation, which depend on availability of reactants in the atmosphere (Hettiyadura *et al.*, 2015), increasing the complexity of understanding their occurrence and descriptions within models. Measurements of specific SCOs have shown they may individually contribute up to 1% of the total organics (Olson *et al.*, 2011; Liao *et al.,* 2015).

Isoprene SCOs are hypothesized to be the most abundant precursor in the ambient atmosphere (Surratt *et al.*, 2007; Liao *et al.,* 2015) and are often used as markers of isoprene-derived SOA in field campaigns (Zhang *et al.*, 2012b). Aromatic SCOs, considered to originate from anthropogenic sources, have been recently observed in Lahore, Pakistan (Stone *et al.*, 2012) and in urban sites in East Asia (Lin *et al.*, 2012). Riva *et al.* (2015 and 2016) have also previously probed the SCO formation potential from PAH and alkane oxidation in the presence of acidic sulfate aerosols. Glycolic acid sulfate (GAS) is considered another potentially important SCO due to its common abundance and possible sources (Olson *et al.*, 2011; Liao *et al.*, 2015). It is thought to form via a gas phase precursor reaction with an acidic aerosol sulfate or from the particle phase reaction of methyl vinyl ketone with a sulfate particle, although both of these mechanisms are yet to be proven (Liao *et al.*, 2015). GAS is also the only SCO to date, which has been detected in the gas phase (Ehn *et al.*, 2010), providing possible importance of gas to particle phase partitioning of some SCOs.

SCOs are thought to be good tracers for heterogeneous aerosol phase chemistry and SOA formation since the known formation mechanisms involve reactive uptake of gas phase organic species onto aerosol (Surratt *et al.,* 2010). Due to their hydrophilic nature, polarity and relatively low volatility, they may significantly help nanoparticle growth and increase their potential to become cloud condensation nuclei (Smith *et al.*, 2008). Therefore, it is imperative to improve our knowledge of SCO abundance, formation, distribution, precursors and fate to help develop our understanding of SOA formation.

Mass spectrometry coupled with electrospray ionization is a common method to detect SCOs (Iinuma *et al.*, 2007; Reemtsma *et al.*, 2006; Surratt *et al.*, 2007; Gomez-Gonzalez *et al.*, 2008). Liquid chromatography can efficiently separate aromatic and monoterpene derived organic sulfates containing aromatic rings or long alkyl chains and is used for speciation of SCOs (Stone *et al.*, 2012). Furthermore, hydrophilic interaction liquid chromatography has been utilized as a very selective technique due to its ability to allow the SCO to retain a carboxyl group, enabling detection of a larger suite of compounds (Gao *et al.,* 2006). The methods above often rely on sampling filters taken in the field and therefore provide a relatively low measurement frequency. This can limit the ability to evaluate production pathways when concentrations are often integrated over a period of hours or more. Further reactions on filters between the organics and sulfates hasve also been postulated to add a bias to the SCO concentration measured with respect to initial deposition onto the filter (Hettiyadura *et al.*, 2017; Kristensen *et al.*, 2016). Recently, a Particle Analysis Laser

Mass Spectrometer (PALMS) was utilized to measure a number of OSs over the United States highlighting the ability of time of flight mass spectrometers to measure several SCOs at high time frequencies (Liao *et al*., 2015).

This study utilizes a Filter Inlet for Gas and AEROsol (FIGAERO) Time of Flight Chemical Ionisation Mass Spectrometer (ToF-CIMS) for the measurement of ambient SCOs at a semi-rural site 40 km from Beijing, China. This instrument enables measurements of either the gas-phase components or thermally desorbed particles by a high resolution mass spectrometer via a multi-port inlet, as described in detail by Lopez-Hilfiker *et al.,.* (2014). The soft and selective ionization technique and high time resolution coupled with the FIGAERO enables the simultaneous detection and measurement of SCOs in the gas and particle phase at ng m$^{-3}$ concentrations. This work aims to identify dominant SCOs in Beijing and their precursors. The high time resolution measurements are utilized to probe their abundance under different chemical and environmental regimes providing insight into their formation.

**2. Experimental**

**2.1 Site description**

The data presented here was collected during the measurement campaign "Photochemical smog in China" with an initiative to enhance our understanding of SOA formation via photochemical smog in China (Hallquist *et al.*, 2016). The campaign was coordinated by Peking University (PKU) and the University of Gothenburg with focus on spring/summertime episodic pollution episodes in Northeast China through gas and particle phase measurements. The setup was situated at a semi-rural site 40 km North East of downtown Beijing close to Changping town (40.2207° N, 116.2312° E). All on-line instruments sampled from inlets on the 4$^{th}$ floor laboratory (12 metres above ground) at Peking University Changping Campus from the 13$^{th}$ May to 23$^{rd}$ June 2016, while filter measurements took place on the roof. The average temperature and relative humidity throughout the campaign were 23°C and 44% respectively. The wind speed averaged at 2 ms$^{-1}$ from the South-South West. A total of 4 pollution episodes were observed during the campaign period, which are classified as sustained periods of high aerosol loading reaching a maximum of 115 micrograms per cubic meter (μg m$^{-3}$).) for PM1. The episodes were dominated by organic and nitrate aerosols although episode 3 contained high sulfate loading, equal to that of nitrate. The mass loading for the semi-rural site showed good correlation with the PKU campus measurement site (30 km South-South-West of the Changping site and 12 km North West of downtown Beijing) throughout the campaign allowing for extrapolation of the semi-rural site results to inner city conditions. HYSPLIT back trajectory results showed the pollution episodes often correlated with air masses coming from the direction of Beijing (South-South-East). Clean air days were mostly with North Westerly winds with clean air coming from the rural mountain regions North West of Beijing and the measurement site.

A high resolution Time of Flight Aerosol Mass Spectrometer (ToF-AMS) was utilized to measure the mass concentrations and size distributions of non-refractory species in submicron aerosols, including organics, sulfate,

ammonium and chloride (DeCarlo *et al., 2006*; Hu *et al.,* 2013). The setup of this instrument has been previously described by Hu *et al.* (2016). An Ionicon Analytik high sensitivity PTR-MS (Proton TRansfer Mass Spectrometer) as described by de Gouw and Warneke *et al.,. (2007) provided supporting precursor VOC measurements.

**2.3 ToF-CIMS setup**

Gas and particle phase species were measured using an iodide ToF-CIMS coupled to the FIGAERO inlet (Lopez-Hilfiker *et al*., 2014). The ToF-CIMS can be operated in either negative or positive ionization modes, and a variety of reagent ion sources can be used. In this work the ToF-CIMS was operated in single reflection mode. The negative Iodide ion (I⁻) was used as the reagent in all experiments. Dry UHPultra high purity $N_2$ was passed over a permeation tube containing liquid $CH_3I$ (Alfa Aesar, 99%), and the flow was passing a Tofwerk X-Ray Ion Source type P (operated at 9.5 kV and 150 μA) to produce the ionization ions. The ionized gas was then directed to the IMRIon-Molecule Reaction (IMR) through an orifice (Ø = 1 μm). Reaction products (e.g., compound X) were identified by their corresponding cluster ions, XI⁻ or the deprotonated ion, allowing for the collection of whole-molecule data. The nominal reagent and sample flow rates into the Ion-Molecule Reaction (IMR)IMR chamber of the instrument were 3.5 liters per minute (LPM) and 2 LPM respectively. The IMR itself was temperature controlled at 40˚C and operated at a nominal pressure of 500 mbar. The ToF-CIMS was configured to measure singularly charged ions with a mass to charge ratio (*m/z*) of 7 – 620, a reduced mass range in order to compensate for the lower count rate emitted by the soft X-ray source with respect to the Polonium-235 radioactive source as commonly deployed. The tuning was optimized to increase sensitivity, which resulted in a spectral resolution of 3500. The mass range was at some instances during the campaign changed to a higher mass range (1000 *m/z*) to ensure no major contributing peaks were being unaccounted for. Perfluoropentanoic acid was utilised as a mass calibrant up to *m/z* 527 through its dimer and trimer. This range of mass calibration peaks also limited accurate peak identification above *m/z* 620.

**2.4 FIGAERO inlet**

The FIGAERO inlet collected particles on a Zefluor® PTFE membrane filter. The aerosol sample line was composed of 12 mm copper tubing, while 12 mm Teflon tubing was used for the gas sample line. The FIGAERO was operated in a cyclic pattern; 25 minute of gas phase sampling and simultaneous particle collection, followed by a 20 minute period during which the filter was shifted into positioned over the IMR inlet and the collected particle mass was desorbed. Desorption was facilitated by a 2 LPM flow of heated UHP $N_2$ over the filter. The temperature of the $N_2$ was increased from 20 to 250˚C in 15 minutes (3.5˚C min⁻¹), followed by a 5 minute temperature soak time to ensure that all remaining mass that volatilizes at 250˚C was removed from the filter. The resulting desorption time series profiles allowed for a distinct separation of measured species as a function of their thermal properties.

**2.5 SCO Measurement**

**2.5.1 Identification**

Spectral analysis was performed using Towfware V2.5.11. the average peak shape for the tuning utilised for this campaign was used to calculate the mass resolution and optimization of the baseline fit. The mass spectrum was mass calibrated (allowing for accurate centroid peak position to be estimated, improving on a Gaussian assumption) using 4 ions up to mass 527 (the dimer of perfluorpentanoic acid) and applying a custom peak shape to achieve accurately peak identification below 5 ppm error across the mass range (0-620 AMU). A time series of the mass calibrant error for the entire campaign (figure SI1) illustrates how the error deviated by only +/-1 throughout the measurement period. This provides confidence that variation in signal and peak positioning did not result in large errors of identification and quantification of the analyzed peaks. *A priori* unknown peaks were added to resolve overlapping peaks on the spectra until the residual was less than 5%. Each unknown peak was assigned a chemical formula using the peaks exact mass maxima to 5 decimal places and also isotopic ratios of subsequent minor peaks. An accurate fitting was characterized by a ppm error of less than 5 and subsequent accurate fitting of isotopic peaks. An example of the spectra and peak fitting can be found in Figure 1, highlighting the mass spectral fit for GAS and $C_9H_9SO_5$.. Although the structure cannot be determined with CIMS, it is assumed that no fragmentation of larger SCO species contribute to the SCO identified due to the soft ionization technique employed. The SCOs were identified in the spectra as negative ions assumed to be formed by hydrogen removal. Here, we present 17 SCOs that were identified in the mass spectra, which are displayed in Table 1 with their respective exact mass, formula, literature nomenclature and possible precursors. The peak fittings for all 17 SCOs is presented in the supplementary (SI2). All 17 SCOs represented a significant signal in the average desorption spectra from the particle phase analysis. It must be noted that gas phase spectra at times contained other ions at a similar mass to the SCOs that contributed to higher counts than the SCO. This may result in a variable error to the measurement, although this should be at a minimum due to the use of a custom peak shape and low general mass calibration error of the spectra. The SCOs detected ions ranged from 154.96 *m/z* (GAS) to 294.06 *m/z* ($C_{10}H_{11}NSO_7$). The number of oxygen in the SCO ranged from $O_3$ ($C_7H_7SO_3$) to $O_7$ ($C_5H_8SO_7$). It is acknowledged that the CIMS may not detect all SCOs in the ambient air due to peak fitting resolution limitations and limits of detection, therefore enabling the possibility for misrepresentation of the dominant SCO and an underestimation of total abundance. However, no physical features of the SCO (structure, O:C ratio, mass etc.) should inhibit the CIMS identifying the major SCO in the Beijing ambient air. Consequently, we here, and to facilitate descriptions of the relationship between individual SCOs and the total SCO measure, assumed that the measured SCOs do represent a significant fraction.

**2.5.2 Quantification of SCOs**

The OS and nitroxy organo sulfates (NOS) calibrations normalized to formic acid calibrations (as described in Le Breton *et al*., 2012, 2013) to account for any drift in sensitivity throughout the campaign. This relative sensitivity technique has been previously utilized for $N_2O_5$ and $ClNO_2$ and has been verified with laboratory experiments (Le Breton *et al*., 2014). As a result of low mass range of the SCOs, common functionality, relatively small change in polarity and lack of available stable SCO standards, we calibrated for 2 SCOs (lactic acid sulfate (LAS) and NP OS) and applied an average sensitivity for all the SCOs detected in Beijing. The ToF-CIMS sensitivity utilizing iodide as a reagent ion is known to vary by up to 3 orders of magnitude; therefore, further work is necessary to develop SCO standards and assess possible variations in sensitivity. NP OS is available commercially from Sigma Aldrich and was utilised to calibrate for the NOS²s. L(+) Lactic acid from Sigma Aldrich (95%) was utilised as the preliminary agent for lactic acid sulfate synthesis and was produced using the same technique by Olson *et al*. (2011). Briefly, a solution of 76.1 mg, 1.29 mmol, lactic acid in 2 mL di-methyl-formamide (DMF) was added dropwise to sulphur trioxide pyridine (0.96 g, 7.75 mmol) in 2mL DMF at 0 ºC. The solution is then stirred for 1 hour at 0 ºC and 40 minutes at room temperature, the solution is re-cooled to 0 ºC and trimethylamine (0.23 mL, 1.66 mmol) was added for quenching and the mixture wad further stirred for 1 hour. The solvent is then evaporated under vacuum and NMR is directly utilised to calculate the purity which was found to be 8.2%.

A known mass of the solid calibrant (NP OS and Lactic acid sulfate) was added to 3 different volumes of milliQ water to produce different concentration standards. A known volume of each solution was then placed onto the FIGAERO filter and a desorption cycle was performed. The total ion counts for the high resolution (HR) SCO peak relates directly to the sensitivity of the system with respect to total ion counts per molecule reaching the detector. Figure 2 shows a 3-point calibration curve for NP OS and the corresponding thermogram, mass spectra and peak fit. The sensitivity of LAS and NP OS calibrations was calculated to be 2.0 and 1.6 ion counts per ppt $Hz^{-1}$ respectively. All SCO were calibrated using the LAS sensitivity and all NOS using the NP OS sensitivity.

During desorption of both SCOs, fragmentation of the organic core and sulphate group was observed resulting in a desorption profile at *m/z* 97 (the bisulphate ion) and the deprotonated organic mass, i.e., $C_3H_5O_3$ for lactic acid. A number of different temperature ramping rates was performed with the FIGAERO to further probe the fragmentation and it was found that an increase in ramp rate (ºC/minute) decreased the calculated sensitivity due to an increase in fragmentation. This not only serves to highlight how the calibration tests of a species must mimic the exact measurement conditions, but also suggests potential interferences from fragmentation on the organic *m/z´*s. The relatively low concentration of the organic precursor with respect to the SCO results in little error in quantification, although this ratio may significantly change in different air masses and a number of products of organic oxidation may fragment resulting in a significant error. This fragmentation can also be observed within the high resolution thermograms of the FIGAERO as a double desorption and further highlights the necessity for detailed thermogram

analysis to accurately deconvolve desorption's relevant only to particle loss from the filter and not fragmentation or ion chemistry in the IMR. The fragmentation is considered to be constant throughout the campaign. The error for the SCO measurements may vary for each individual SCO possibly due to structure, volatility and fragmentation. It is commonly accepted within the literature for compounds lacking calibration that a functional group sensitivity can be applied (e.g. Lee *et al.* (2016) for organic nitrates (ONs)). Here we calculate an average error of 52 % for the SCOs, calculated using the standard deviation of the NP OS calibration time series data.

The limitation of FIGAERO temperature ramps to 250 ˚C may result in further error as some SCOs may not be fully desorbed from the filter due to their low vapour pressures. To evaluate the mass left on a filter, several double desorption cycles were performed where mass is collected and desorbed such as in standard use. This is performed by re-heating the same filter once cooled to attain a second thermogram of the same filter. The second thermogram exhibited an average of 90% reduction of counts for the SCO, although the NOSs had an average decrease of 82% counts. This indicates that most, but not all mass, are removed from the filters when desorbing. For the interpretation of the results of the field campaign this effect will induce a small distortion on the time evolution of SCOs when comparing to other parameters, e.g. 9% of NOSs will remain on the filter and being subjected to the subsequent desorption cycle.

**2.5.3 Offline and online measurement comparison of SCOs**

Filter measurements, for orbitrap and HPLC MS analysis, were taken diurnally at the same sampling site, although from a different inlet and location in the building. The CIMS hourly desorption data was averaged over the corresponding collection period to attain a day and nighttime CIMS data point. The period between the 23rd May and 1st June was selected due to all instrument measurements being undisturbed during this period. It must be noted that CIMS data is lacking one data point daily while the background filter measurements were taken. The CIMS, orbitrap and HPLC do not measure all of the same species. Here a comparison of total 5 ions is presented with the HPLC and 2 from the orbitrap, where a further extensive comparison is to be performed in an accompanying manuscript. Figure SI3 illustrates the time series of CIMS (hourly and diurnal) measurements of GAS and IEPOX sulfate alongside the HPLC measurements. The diurnal data agrees well with an R value of 0.78 and 0.82 for GAS and IEPOX sulfate respectively. The sum of the time series we have multi instrument data for (GAS; IEPOX sulfate, LAS, $C_4H_7SO_7$, $C_5H_{11}SO_7$ and $C_5H_7SO_7$) for HPLC and $C_9H_{11}SO_5$ and $C_9H_8SO_5$ for orbitrap is displayed in the top panel. In general, the time series agree well and also have a good correlation (R = 0.7 and 0.81 for HPLC and orbitrap respectively) illustrating the ability for CIMS to agree with the offline methods and measure the SCOs accurately at low and high time resolution.

**2.7 Knudsen Effusion Mass Spectrometer (KEMS)**

The KEMS technique was utilized to measure the vapour pressure of SCOs observed in the gas phase measurements by the CIMS. The KEMS technique is able to measure vapour pressures from $10^{-1}$ to $10^{-8}$ Pascals (Pa) ranging from volatile organic compounds to extremely low volatility organic compounds. A full description of the technique can be found in Booth *et al*. (2009, 2010) and the measurements of a series of compounds over a large vapour pressure (VP) range, in a recent inter-comparison study from this instrument, can be found in Krieger *et al*. (2017). Briefly, the instrument consists of a temperature controlled Knudsen effusion cell, suitable for controlled generation of a molecular beam of the sample organic compounds in a vacuum chamber, coupled to a quadrupole mass spectrometer. The cell has a chamfered effusing orifice with a size ≤1/10 the mean free path of the gas molecules in the cell. This ensures the orifice does not significantly disturb the thermodynamic equilibrium of the samples in the cell (Hilpert, 2001). The system is calibrated using the mass spectrometer signal from a sample of known vapour pressure, in this case malonic acid (vapour pressure at 298K = $5.25 \times 10^{-4}$ Pa, (Booth *et al*., 2012)). A load-lock allows the ioniser filament to be left on, then a new sample of unknown vapour pressure can be measured. Solid state vapour pressures measured in the KEMS can then be converted to sub-cooled liquid vapour pressures using the melting point, enthalpy and entropy of fusion, which are obtained by using a Differential Scanning Calorimeter (DSC) (TA instruments Q200).

**3. Concentrations and partitioning of atmospheric SCOs**

**3.1 SCO contribution to $PM_1$ at Changping**

The SCOs measured at the Changping site had a mean campaign concentration of 210± 110 ng m$^{-3}$ (Table 1). The highest concentration of total SCOs during the campaign was 540 ng m$^{-3}$ and the lowest 40 ng m$^{-3}$, thus they are omnipresent and have significant sources during most atmospheric conditions. These concentrations are consistent with Stone *et al*. (2012) reporting an average SCO concentration of 700 ng m$^{-3}$ in a number of rural and urban sites in Asia. A mean SCO contribution to organic aerosol (OA) in the work presented here was calculated to be 2.0 ± 1% (Table 1), within the range of values calculated by Stone *et al*. (2012) (0.8% to 4.5%), further supporting evidence that the SCO contribution to $PM_1$ mass is relatively low in Asia. The CIMS cannot claim to measure total SCO, rather than singularly identify and measure SCOs contributing to the total mass loading. Therefore, the SCO contribution reported in this work should be considered as a lower limit. The Liao *et al*. (2015) study also supports the idea that the SCO contribution to $PM_1$ mass in anthropogenically dominated regions is less significant than that from biogenically dominating air masses by observing a significantly higher contribution of IEPOX sulfate to $PM_1$ mass on the East coast of the United States (1.4%) than the West Coast (0.2%).

The observation of higher relative contribution of SCOs to total organics in more remote regions compared to a densely populated urban area, supports the idea that SCOs provide a higher contribution to PM in aged air due to their secondary

production pathways. Similar to Lahore (as studied by Stone *et al*., 2012), Beijing has many strong primary anthropogenic sources which will dominate the mass loading and therefore, initially, will contribute to a lower fraction of the total concentration from secondary production due to limited processing near the source. Throughout the campaign, a good correlation ($R^2 = 0.66$) was observed between an increase in ΔSCO mass and $PM_1$ mass, although the SCO contribution to $PM_1$ decreased exponentially (Figure 3) indicating that the pollution episodes contain a lower fraction of SCOs with respect to total $PM_1$. This result suggests that SCO do not play as large a role as expected even though their precursors (organics and sulfate) are abundant within the episodes, indicating the conditions of their formation may be more vital than the absolute concentrations of precursors.

**3.2 Gas to particle phase partitioning of SCOs**

The FIGAERO ToF-CIMS data exhibited indication of SCOs in both the particle and gas phase. Previous studies have supported the existence of e.g., gas phase GAS in ambient air (e.g., Ehn *et al*., 2010), although some work has attributed other measurement techniques detection of gas phase SCO to result from measurement artefacts (Hettiyadura *et al*., 2017; Kristensen *et al*., 2016). Once all HR peaks have been identified, the batch fitting and HR time series for the whole data set is processed and then separated into gas phase measurements and particle phase desorption profile time series. The data is background corrected, i.e., subtraction of both the gas phase background periods and blank filter desorption's. Upon analysis of the resultant data, significant concentrations of gas phase SCOs were observed. Figure 4 depicts the overall sum SCO mass concentration time series in the gas and particle phase. The mean contribution from gas phase SCO to total SCO was found to be 11.6%, $23\pm8$ ng m$^{-3}$. This suggests a significant amount of SCO is always present in the gas phase and factors that influence gas-to-particle partitioning influence the level of this contribution. These changes in contribution also reduce the possibilities for memory effect, e.g., one possibility is the deposition of SCOs onto the IMR walls during the temperature ramp of the desorption which in time may de-gas and be observed in the gas phase. This would likely result in a constant ratio of particle to gas phase concentrations and would likely cause a hysteresis in the observed gas phase measurements with respect to the particle phase, which was not observed.

The vapour pressure of NP OS was measured using the KEMS instrument in the laboratory to establish the existence of gas phase SCOs. This technique has recently been employed to measure the vapour pressure of NP (Bannan *et al*., 2017). The KEMS experiments found the solid state vapour pressure of NP OS to be $5.07\times10^{-5}$ Pa at 298 K. Assuming an average subcooled liquid correction for all compounds measured in the Bannan *et al*. (2017) study, as no DSC data is available, the subcooled liquid vapour pressure of NP OS is $2.32\times10^{-4}$ Pa. This vapour pressure lies within the semi volatile organic compound range, therefore supporting the potential partitioning of SCOs to the gas phase under ambient conditions. To further validate the CIMS and KEMS findings, one can evaluate different compounds VPs from the FIGAERO data utilizing the $T_{max}$ and compare to literature values. The CIMS, using $T_{max}$, estimated VPs of malonic, succinic and glutaric acid to be $2\times10^{-3}$, $1.85\times10^{-3}$, $1\times10^{-3}$ Pa which compare well to values presented by Bilde *et al*.

(2015) VPs; $6.2 \times 10^{-3}$, $1.3 \times 10^{-3}$, $1 \times 10^{-3}$ Pa. Using this agreement for well-known substances we notice the $T_{max}$ of SCO to be in the range where it can provide significant gas-phase concentration. Still, the observed presence of gas phase GAS and IEPOX-OS does not agree with previous studies of these compounds (Stone *et al.,* 2012, Hettiyadura *et al.*, 2017). Therefore, one needs to be cautious and deeper analysis into exact VPs and partitioning from the present work must be performed to assess whether their gas phase presence could be fully confirmed. So far we note that fragmentation of organic species (oligomers) during desorption could lead to a potential artefact and a lower $T_{max}$ at a monomer peak (Stark *et al.*, 2017 and Lopez-Hilfiker *et al.*, 2016). However, here we identify and expect no dimers or oligomers that could fragment to form the SCOs identified. Furthermore, the higher mass organics are likely to have a much higher VP than the lower mass SCO and provide a second $T_{max}$ which would produce a lower VP value due to the greater energy required to break the bonds. Analysis of $T_{max}$ throughout the campaign shows no double peak thermograms and an acceptable stability of $T_{max}$ (SI4). $T_{max}$ varied by up to 14 degrees Celsius and appeared to correlate well with particulate loading, similar to that observed by Huang *et al.*, (2018), who suggested that this is a result of diffusion limitations within the particle matrix. If the data is tentatively analyzed to assess the mechanism regarding their partitioning, aerosol liquid water content would affect the partitioning of gas phase compounds to aerosols (Zhang *et al.*, 2007). Data point size coding the correlation of the gas and particle phase SCO concentrations indicates partitioning towards the aerosol phase at lower relative humidities (Figure 4). Conversely, as the temperature increases (as indicated by red colour shading) the SCOs partition further towards the gas phase, as thermodynamically expected. Further work is necessary to validate these findings and determine the mechanisms and importance of gas phase SCO abundance in ambient air. For example, the high contribution in the gas phase could be perturbed if equilibrium between condensation to particle phase and gas phase formation has not been established. It must be noted that the a correct calibration of $T_{max}$ with VP would be necessary to extract such information, but qualitatively the relative VP compared to NP OS could be utilized as a reliable scale due its independent calibration by KEMS.

**4. Sources and secondary formation of SCOs**

**4.1 SCO sources at the Changping site**

SCOs are known to have biogenic and anthropogenic sources and some which have multiple sources from both, e.g., GAS (Hettiyadura *et al.,* 2017; Hansen *et al.*, 2014). Burning events are known to emit high levels of organics and nitrates and potentially sulfur, depending on the type of fuel used. This enables biomass burning to be a potential anthropogenic and biogenic source of SCOs. The site at Changping was influenced by both regional anthropogenic pollution from the Beijing area and localized anthropogenic activity (industry, biomass burning and traffic) but also emissions from biogenic sources, as it is situated in a semi-rural area, with forest, vegetation and plantations. This was

evident from the benzene and isoprene PTR-MS measurements which have mean campaign concentrations of 0.55 $\pm$ 0.4 and 0.27 $\pm$ 0.19 ppb respectively with maxima of 5 and 1.5 ppb respectively. Thus, as shown in Figure 5, PTR-MS measurements mean daily concentrations were utilized to evaluate if the ratio between benzene and isoprene can indicate a higher mass loading and contribution of aromatic and biogenic SCOs measured in this work. Data on days with incomplete time series have been removed to ensure the data presented represents a full mean of the day concentration. A good correlation between the benzene:isoprene ratio and sum of SCOs is observed. It suggests an increase in relative anthropogenic emissions promotes an increase in total SCO loading. It should be noted that $C_6H_{10}SO_7$ has no known precursor in the literature, although it contributes significantly to the SCO mass loading in this work (16%).

**4.1.1 *Biogenic and anthropogenic SCOs**

Biogenic SCOs are known to be comprised of monoterpene, sesquiterpene and isoprene derived SCOs which have been identified in rural, sub-urban and urban areas around the world, and have been shown to be a major constituent of SOA (Surrattt *et al*., 2008; Shalamzari *et al.,* 2014, Liao *et al.,* 2015). IEPOX sulfate is commonly found to be the most dominant SCO at many locations and was identified also at the Changping site. The IEPOX sulfate mean concentration represented 0.11% of the OA mass, agreeing well with concentrations found in Western USA (significant anthropogenic emissions) and lower than the Eastern USA as expected due to higher biogenic and isoprene emissions (Liao *et al.,* 2015). Although IEPOX sulfate is considered one of the most abundant individual organic molecules in aerosols (Chan *et al*., 2010), here our results show it only contributed to 2% of the SCO mass and was the 8[th] most abundant SCO in the particle phase. Additionally, two other isoprene derived SCOs, $C_5H_8SO_7$ and $C_4H_8SO_7$, were measured by the CIMS with mean campaign concentrations of 2 and 3 ng m$^{-3}$ respectively and a contribution of 0.02% to OA mass. The highest contributing biogenic SCO to the ambient air was a NOS, $C_{10}H_{16}NSO_7$, a known NOS derived from alpha-pinene oxidation. This NOS had a mean campaign concentration of 21 ng m$^{-3}$ and a 0.2% contribution to OA mass.

Anthropogenic SCOs, including polyaromatic hydrocarbon (PAH) derived SCOs have received more attention in recent studies due to their identification (Nozière *et al*., 2010; Hansen *et al*., 2015). Aromatic SCOs and sulfonates have only recently been identified as atmospherically abundant SCOs (Riva *et al.,* 2015). In this work we find that the PAH derived SCO $C_{11}H_{11}SO_7$ is the most dominant SCO in Beijing with a mean concentration of 40 ng m$^{-3}$, contributing to 20% of the total SCO mass and 0.4% of the OA mass. This SCO has been identified in laboratory studies as an SCO forming from the photo-oxidation of 2-methyl napthalene, one of the most abundant gas phase PAHs and is thought to represent a missing source of urban SOA (Riva *et al.,* 2015). This work presents the possible significance of PAH SCOs in Beijing and further evidence that photo-oxidation of PAHs represents a greater SOA potential than currently recognized. A further 8 anthropogenic aromatic derived SCOs were identified as common components of the PM$_1$

representing more than half of the total SCOs with $C_7H_5SO_4$ contributing to 24 ng m$^{-3}$ and 0.23% OA mass. The total anthropogenic related SCOs had a mean mass of 120 ng m$^{-3}$ and contributed to 1.2% of the OA mass.

**4.1.2 *Biomass burning source of SCOs**

NP (a product of benzene oxidation and nitration) has previously been detected in the gas and aerosol phase (Harrison *et al.,* 2005) and is an important component of brown carbon (Mohr *et al.,* 2013). NP has primary sources, such as vehicle exhausts and biomass burning (Inomata *et al.,* 2013 and Mohr *et al.,* 2013) and secondary sources via the photo-oxidation of aromatic hydrocarbons in the atmosphere (Harrison *et al.,* 2005). High levels on anthropogenic activity, biomass burning and strong photochemistry in Beijing therefore enable this region to be a strong potential source of NP. Both NP and NP OS diurnal profiles exhibit an increase in the morning (6 am onwards) although NP OS appears to increase in concentration more rapidly. The early morning biomass burning and anthropogenic activity are likely to contribute to production of both species, although the higher sulphate content of the biomass burning emissions may promote a faster production of NP OS and conversion of NP to NP OS. Both compounds continue to increase with a photochemical profile with one peak at midday but also a peak around 4 pm, likely to be a second source of the day from anthropogenic activity. The NP OS continues to increase until sunset, which could result from further photochemical production from the NP emitted throughout the day whereas NP falls off after the 4 pm peak. The campaign time series for NP and NP OS can be seen in Figure 6. Unlike its precursor and most other pollutant markers measured in this work, including all other SCOs, NP OS exhibits higher concentrations between 17th to the 22nd May compared to the 28th May to 3rd June. The only compound with a similar campaign profile is acetonitrile (a marker for biomass burning), which has significantly enhanced concentrations between 6 and 8 am from the 17th to the 22nd May. Back trajectories of these two time periods show the air mass during the first period comes from the west, a more rural region of China and known to be influenced heavily by biomass burning, whereas the second time period has wind directions mainly bringing in air masses that have gone through the Tianjin and Beijing area. It is therefore hypothesized that the NP OS, which peaks later in the day than the NP and acetonitrile, is a secondary product formed from the biomass burning and has aged after being emitted from air masses further away. Here NP can have more local sources of biomass burning and traffic which then can contribute to NP OS production, but at a slower time scale, which in this data set, appears as lower production due to the limited oxidation of local air masses.

**4.2 SCO production mechanisms**

**4.2.1 *Precursor analysis**

The availability of the organic precursors of SCOs is a limiting factor for the SCO production rate. The measurement of the precursors in the gas and particle phase by CIMS enables a more descriptive mechanism to be outlined as the

partitioning of the precursor will vary the distribution between gas and particle production pathways and therefore rate of corresponding SCO production. Glycolic acid (GA) has on average 75% of its mass in the gas phase for the measurement period whereas GAS is dominantly in the particle phase (Figure 7). The GAS particle phase concentration is observed to increase as the $SO_4^{2-}$ mass loading increases and the GA gas and particle concentrations increase, although the partitioning of the GA towards the gas phase restricts the SCO production. This can be seen in Figure 7 as for a given $SO_4^{2-}$ concentrations, the data with warm colors (red), representing a high fraction of precursor GA in the particle phase, generally provides a higher concentration of particulate GAS.

The main formation mechanism of GAS is thought to be via the reaction of GA in the gas phase with an acidic aerosol sulfate (Liao *et al*., 20165), contrary to what is observed here. Although an increase in GAS is observed to correlate with the GA, it appears that a partitioning towards the particle phase promotes GAS production. An $R^2$ correlation of 0.68 is observed between GAp and GASp whereas an $R^2$ of 0.4 is observed between GAg and GASp.

The sum of benzene SCOs exhibits a good correlation to the gas phase benzene time series (Figure 8), although their abundance should also rely on the availability of sulfur in the particle phase and the age of the air mass, if it is assumed that they are formed via secondary reactions of primary pollutants. In order to assess how the SCO production rates may vary due to these factors, two distinct high benzene SCO events with similar benzene concentrations were scrutinized, i.e., the 29th May and the 1st June. The first period has lower $SO_4^{2-}$ concentrations, higher $H_2SO_4$ levels and a higher total benzene SCO concentration. The exact age (or time for oxidation) of compounds in an air mass are without an extensive modelling study complicated to derive. However, as proxies to attain an approximation about oxidation state one may use some trace compounds. Monoterpene oxidation by the hydroxyl radical (OH) or $O_3$ results in the formation of multifunctional organic acids such as pinonic acid which can then be further oxidised by OH to form 3-methyl-1,2,3-butane-tricarboxylic acid (MBTCA), both of which are measured by CIMS. Therefore, in an air mass containing monoterpene emissions, as known here through the identification of their products such as $C_{10}H_{16}NO_7S$, we can utilize the ratio of pinonic acid and MBTCA; as tracers of monoterpene SOA processing as detected in ambient aerosols in Europe, USA and the Amazon (Gao *et al*., 2006) as a relative photochemical clock. During the high benzene event on the 1st June, according to the pinonic acid: MBTCA ratio, the air mass is less oxidized relative to the air mass on the 29th May (Figure 8). This would allow less time for secondary production and explain the relatively lower concentration of SCOs, irrespective of higher $SO_4^{2-}$ concentrations and similar benzene concentrations.

To further elaborate The Gothenburg Potential Aerosol Mass (Go:PAM) reactor (Watne et al, 2018) was tested and utilized to simulate aging of the air mass during periods of the campaign. Here the ratio between pinonic acid and MBTCA was observed to increase by an average of 3 during aging within the Go:PAM which has been calculated to be the OH exposure equivalent of 2 days in the ambient atmosphere. As this ratio increased with aging, the SCO concentration also increased exponentially, further supporting the secondary production of SCO in photochemically

aged air mass. Although limited data is available here for simultaneous Go:PAM and CIMS measurements, the results indicate the potential utilization of the chamber to probe secondary production processes.

**4.2.2 Aerosol acidity**

The molecular ion $H_3S_2O_7^-$ was identified in the mass spectra throughout the campaign, which has previously been detected by Liao *et al*. (2015) using a Particle Analysis Laser Mass Spectrometer to measure SCOs. They attribute this mass to be a cluster of $HSO_4^-$ with sulfuric acid ($H_2SO_4$). Particles in the presence of $H_2SO_4$, and therefore high acidity, form this cluster whereas neutralized ions are likely to favour the unclustered $HSO_4^-$ form. Therefore, the ratio between the cluster and the bisulfate ion increases with increasing aerosol acidity (Murphy *et al*., 2007; Carn *et al*., 2011). Liao *et al*. (2015) validate the appropriateness of this cluster as a marker for aerosol acidity thorough comparisons to a thermodynamic model with gas and aerosol phase measurement inputs. Acidity was also calculated utilizing the gas and particle phase $H_2SO_4$ and liquid $H^+$ ion concentration analysed using an offline technique, as described by Guo *et al.* (2010), from diurnal samples taken at the site. This method showed good agreement with the integrated diurnal counts of the $H_3S_2O_7^-$ ion. Therefore, we employ the $HSO_4^-.H_2SO_4^-$ cluster in this work as a qualitative scale for particle acidity utilizing similar assumptions. Figure 9 shows how total SCO mass concentration generally increased as total organic mass from the AMS increased. The correlation indicates that higher acidity (darker colours) tends to promote formation of SCOs when in the presence of high levels of organics and $SO_4^{2-}$ (larger symbol sizes), supporting the growing consensus that aerosol acidity plays an important role in ambient SCO formation. This importance of acidity agrees well with both the acid-catalyzed epoxydiol ring opening formation mechanism (Surrattt *et al*., 2010) and the sulfate radical initiated SCO formation because efficient formation of sulfate radicals also requires acidity (Schindelka *et al*., 2013).

**5. Conclusions**

The FIGAERO ToF-CIMS was successfully utilized for the ambient detection of 17 SCOs in Beijing in the gas and aerosol phase with limits of detection in the ng $m^3$ range. Good agreement with offline filter measurements further supports the robustness of this method for high and low time resolution measurements of SCOs. Further calibrations and comparisons to total SCO measurements are required to evaluate its performance limitation with regards to sensitivity application and peak identification. The SCOs measured by CIMS contributed to 2% of the OA at the semi-rural site, highlighting the relatively low contribution of SCOs in Beijing, an anthropogenically dominated environment.

This calculation from CIMS may only be valid to infer each individual SCO contribution to total SOC mass as limitations in SCO identification and quantification limit the CIMS ability for total SCO measurements. Significance of their secondary production pathway prevailed, although still present in relatively fresh air masses. Contributions of SCO to total organics (2.0±1%), sulfate (15±19%) and PM (1.0±1.4%) indicate the concentrations observed in Beijing result from highly processed ambient air masses.

Gas phase SCOs were identified for all the SCOs measured at the site, contributing to in average to 12% of the total SCO mass. The possibility of gas phase SCOs in ambient air was supported by KEMS vapour pressure measurements of NP OS and derived $T_{max}$ values which suggest a vapour pressure in the semi volatile range. The partitioning towards the gas phase was more efficient at high atmospheric temperatures, while lower relative humidities promoted partitioning to the particle phase.

Biogenic SCOs contributed to a small fraction of the total SCO mass at Changping and waswere dominated by an α-pinene derived OS with 0.2% contribution to the OA mass. IEPOX sulfate was only the $8^{th}$ most abundant SCO measured, contrary to common reports that it is one of the most abundant SCOs. Anthropogenic precursors contributed to more than half of the SCO mass loading with a PAH derived SCO contributing to as much as 1.2% of the OA mass. Benzene derived SCOs correlated well with gas phase benzene levels and were heavily influenced by photochemical aging. The contribution of each benzene derived SCO to total benzene derived SCO mass varied daily and throughout the campaign highlighting the complexity of the atmospheric processing and composition of SCO. Significant contributions from aromatic SCOs highlight the importance of anthropogenically emitted organics in the Beijing region and their contribution to the Beijing outflow and subsequent photochemistry. NP OS was attributed to biomass burning emissions due to co-occurrence with high levels of acetonitrile. This highlights the importance of anthropogenic emissions and their contribution to SOA from the urban Beijing outflow.

A qualitative CIMS marker for aerosol acidity highlighted the increase in SCO production rate in acidic aerosols in the presence of high $SO_4^{2-}$ and organics. The correlation of SCO production and RH becomes more complex for individual SCOs, which cannot be resolved within this studies framework.

**Acknowledgement:**

The work was done under the framework research program on 'Photochemical smog in China" financed by the Swedish Research Council (639-2013-6917). The National Natural Science Foundation of China (21677002) and the National Key Research and Development Program of China (2016YFC0202003) also helped fund this work.

[Figure]

**Figure 1. The bottom panel displays the average campaign mass spectrum for the whole mass range of the ToF (3-620) which is further expanded to show small regions in the middle panel and specific and HR fitting for individual peaks in the top panel.**

**Table 1. SCOs identified at the Changping site with their respective mass, chemical name and potential precursors.**

| m/z ion | Molecular formula | Reference | OS name | Precursor | [mean] µgm-3 | mean % PM | mean % OA | mean % OS |
|---|---|---|---|---|---|---|---|---|
| 154.965582 | $C_2H_3SO_6^-$ | Surrat 2007 | Glycolic acid sulphate | Glycolic acid | 2.97 | 0.02 | 0.03 | 1.6 |
| 168.981232 | $C_3H_5SO_6^-$ | Olson 2011 | Lactic acid sulphate | Lactic acid | 13.00 | 0.07 | 0.14 | 5.9 |
| 171.012139 | $C_7H_7SO_3^-$ | Riva 2015 | | Aromatics (Benzene and PAHs) | 6.00 | 0.03 | 0.06 | 2.7 |
| 172.019964 | $C_7H_8SO_3^-$ | Riva 2015 | | Aromatics (Benzene and PAHs) | 4.00 | 0.02 | 0.04 | 0.9 |
| 184.991403 | $C_7H_5SO_4^-$ | Riva 2015 | | Aromatics (Benzene and PAHs) | 24.00 | 0.12 | 0.25 | 12.1 |
| 187.007053 | $C_7H_7SO_4^-$ | Staudt 2014 | Methyl phenyl sulphate | benzene | 14.00 | 0.07 | 0.15 | 6.7 |
| 199.007053 | $C_8H_7SO_4^-$ | Riva 2015 | | Aromatics (Benzene and PAHs) | 5.00 | 0.03 | 0.05 | 2.4 |
| 199.999622 | $C_4H_8SO_7^-$ | Surrat 2007 | | 2-methylglyceric acid (isoprene) | 3.00 | 0.02 | 0.03 | 0.8 |
| 201.022703 | $C_8H_9SO_4^-$ | Staudt 2014 | 4 methyl benzyl sulphate | benzene | 6.00 | 0.03 | 0.06 | 3.7 |
| 211.999622 | $C_5H_8SO_7^-$ | Surrat 2008 | | isoprene | 2.00 | 0.01 | 0.02 | 1.3 |
| 215.023097 | $C_5H_{11}SO_7^-$ | Surrat 2010 | IEPOX sulphate | IEPOX | 11.00 | 0.06 | 0.12 | 4.5 |
| 217.9759 | $C_6H_4NSO_6^-$ | - | Nitrophenol sulphate | Nitrophenol | 1.00 | 0.01 | 0.01 | 0.4 |
| 226.015272 | $C_6H_{10}SO_7^-$ | Boris 2016 | unknown | unknown | 30.00 | 0.16 | 0.32 | 15.1 |
| 229.017618 | $C_9H_9SO_5^-$ | Riva 2015 | | Aromatics (Benzene and PAHs) | 10.00 | 0.05 | 0.11 | 5.0 |
| 231.033268 | $C_9H_{11}SO_5^-$ | Riva 2015 | | Aromatics (Benzene and PAHs) | 13.00 | 0.07 | 0.14 | 6.7 |
| 287.023097 | $C_{11}H_{11}SO_7^-$ | Riva 2015 | | Aromatics (Benzene and PAHs) | 40.00 | 0.21 | 0.42 | 20.2 |
| 294.065296 | $C_{10}H_{16}NSO_7^-$ | Surrat 2008 | | alpha pinene | 21.00 | 0.00 | 0.22 | 9.9 |

[Figure]

**Figure 2. The desorption profile of NP OS 3 step calibrations for 0.1 μl, 0.2 μl and 0.3 μl 1000 ppm solution is displayed in the bottom panel and its corresponding average stick spectra (top left) and sum of counts per molecule loading for each calibration (top right)**

[Figure]

[Figure]

**Figure 3.** The time series of total SCO (colour coded with time) is displayed in the bottom panel. The time series of the AMS organic (green) and sulfate (red) is displayed in the middle panel and the correlation of SCO to PM$_1$, mass fraction of PM$_1$ and organics are displayed in the upper panel. The colour coding represents time throughout the campaign.

[Figure]

**Figure 4.** The time series of total SCOs in the gas and particle phase (bottom panel) and their correlation colour coded by temperature and size binned by relative humidity (top panel).

[Figure]

**Figure 5. A time series of the mean daily benzene to isoprene ratio as a marker for anthropogenic and biogenic influence (black) is displayed in the top panel. The CIMS data was also binned to provide mean daily SCO concentrations for aromatic (blue) and biogenic (green) precursor SCOs (bottom panel). The red bars represent SCOs with an unknown source or SCO produced via both biogenic and anthropogenic pathways. The AMS $SO_4^{2-}$ concentration is also presented to indicate availability of sulfur in the particle phase.**

[Figure]

**Figure 6.** The time series of gas and particle phase NP (red), NP OS (blue) and gas phase acetonitrile (grey) and CO (black) between the 16th and 3rd June

[Figure]

**Figure 7. Campaign time series of glycolic acid (red) and GAS (blue) in the particle and gas phase. The top panel illustrates the correlation between GAS in the particle phase and SO$_4^{2-}$ colour coded by GAp/GAg.**

[Figure]

**(A)**

[Figure]

**(B)**

Figure 8. (A) Total benzene/PAH derived SCO (SCO_anthro) time series and respective $SO_4^{2-}$ and benzene concentrations. The indicator of photochemical aging (pinonic acid: MBTCA) is plotted in green. (B) illustrates the mass spectral difference between fresh and aged air masses through Go:PAM and respective time series for pinonic acid and MBTCA.

[Figure]

**Figure 9. Correlation plot of total SCOs vs total particle phase organics as a function of acidity (colour coding counts of HSO₄.H₂SO₄: HSO₄) and SO₄²⁻ (data point size spanning concentrations from 0.2 to 16 µg m⁻³).**

---

## Author Response (AR2)

Response to reviewers on "Chlorine oxidation of VOCs at a semi-rural site in Beijing: Significant chlorine liberation from $ClNO_2$ and subsequent gas and particle phase Cl-VOC production" by Michael Le Breton et al.

**Report 1**

I thank the authors for having taken many of my suggestions in consideration and having made many appropriate changes. This manuscript will require further edits before it can be considered for publication as the present version seemed to have been submitted in a bit of a rush. It would have been nice to see a marked-up version of the paper that indicated what precise changes were made (there was a marked-up version but of another paper attached to the author-response-version2.pdf file).

**Major comments**

1 - Many of the "measurements" (HCl, Cl2, ClONO2, HOCl, ClO, OClO, CMBO, IOPEX, ...) continue to be presented with units of mixing ratios, even though the instruments making these measurements have not been properly calibrated for these compounds.
For example, the authors stated in their response letter that "Le Breton et al 2017 showed that inorganic halides have a similar sensitivity. Furthermore, the comparable sensitivity for chloroacetic acid and ClNO2 emphasis a similar sensitivity for chloride containing species when applying the iodide ionisation". It is unclear if this paragraph refers to the 2017a or 2017b paper and what is meant by the terms "similar" and "comparable", since these are qualitative, not quantitative descriptors.

It'd be OK if the data were presented as raw data with units of Hz or counts and the subsequent discussion of the data (including that of Cl VOCs) adjusted accordingly. The use of concentration or mixing ratio units, however, is not justified.

The manuscript has now been amended to present uncalibrated species with units of counts.

2 - The calculations of Cl and OH concentrations using simplistic steady state equations remain questionable. For starters, some of the input parameters are highly uncertain (see point #1 above). There are also some pretty coarse assumptions being made about the Cl and OH hydrocarbon reactivity at this site.

Yes, one can calculate concentrations using rough approximations and steady state assumptions ("In previous work (Bannan et al., 2015) we have shown that it is possible to calculate Cl atom concentrations using simply state expressions"). However, simply having these type of calculations before does not mean that this approach gives accurate results (as the text in the supplemental suggests, see specific comment below).

Furthermore, just because a past steady state calculation "agreed well with the MCM" does not mean it will agree for other data sets (not sure what "agreeing well" even means in this context). Besides, chlorine chemistry is a recent addition to the MCM and is likely still incomplete, such that it hardly constitutes a gold standard.

We agree with the referee and we hope that we have clarified the uncertainty in the steady state calculations through the modified text that follows in the main text and the SI section

Main text:

Although this approach is an estimation, it was shown to produce results comparable with that of the more rigorous MCM approach although we do acknowledge large errors will be present in the radical species calculations, which is detailed in the supporting information.

SI section:

There is a large uncertainty in the radical concentrations estimated in this work using the steady state method and this is fully acknowledged. The production rates for Cl are calculated directly from measurements of species, e.g. ClNO2 concentrations and their photolysis rates, estimated by a photochemical model and so that aspect of the calculation has a relatively small uncertainty. There will be missing sources of Cl production that are not measured but the main known precursors are represented in these calculations. However, the loss rate carries the bulk of the uncertainty in these calculations; first it is known that not all VOCs are measured and even after estimation of missing VOCs there will be some missing loss. Second, calibration of some VOCs in this campaign will be uncertain, hence the reporting of counts per second rather than absolute numbers. Third, the rate coefficients associated with the loss processes will carry some uncertainty too. Finally, the distribution of missing VOCs is based on USA and European emission profiles and these maybe different from those from Asia. Therefore, the uncertainty in radical concentration using this method will be at least 50%, where 20% is from rate coefficient uncertainty, taking into consideration temperature

dependences, and an estimated 30% from uncertainty in concentration measurements. However, OH reactivity measurements (e.g. Yang et al., 2016) provide a direct measurement of total loss rates and although not a direct comparison with Cl loss rates they can serve as a guide. Using 20s-1 as an estimate for the total loss rate for OH the steady state calculations are predicting loss rates in the range 10-15 s-1. Therefore, a 50% uncertainty and recognising that the concentrations generated are almost certainly an upper limit provides some further context to these calculations.

3 - some "action" items identified by the authors themselves have not been completed - for example, reactions are still not numbered consecutively (see specific comments below), entire paragraphs appear without sub- and superscripts, the references continue to have doi's missing, plus there are still numerous grammatical errors that need be corrected.

The reactions are now numbered consecutively and all sub and superscripts are now applied appropriately

**Specific comments**

pg 1 / lines 29 and 30. Use of pptV and ppt is inconsistent. The acronym ppt should be defined as it could be misunderstood as part per thousand.

ppt and ppb is now consistently used throughout the text

pg 2 / line 29. "The liberated chlorine will predominantly react with VOCs"
This should be one of the reactions listed above. Cl will more quickly abstract a hydrogen from a hydrocarbon (or add to a double bond) than react with O3.

We state below the equations that OH "will predominantly react with VOCs with the pathways listed (R2-R11) representing alternative routes to loss of the chloride radical. The description of VOC oxidation then follows in the next paragraph.

pg 2 / line 31 $HO_2$, $RO_2$ etc. should have subscripts

These formulas have now been subscripted

pg 3 / line 7 - "The oxidation mechanism of saturated hydrocarbon (R12-R13) is"

Please rephrase.

This has been rephrased to read

"Saturated hydrocarbons are usually oxidised by reaction with OH or chlorine atom to form an organic peroxy radical (RO2), and H2O or HCl depending on the oxidant (R12 and R13), which is the dominant pathway for chloride-VOC reactions."

pg 4 / lines 7-12 A major factor (which should be acknowledged here) is the availability of aerosol chloride, which varies considerably between measurement locations.

The following sentence has been added

"A major factor in the variation of ClNO2 mixing ratios is the availability and abundance of aerosol chloride which can vary significantly, although is predominantly  present as sodium chloride from sea salt."

pg 5 / lines 6-9. Thanks for adding more detail on the photolysis rate measurements. Please state whether this a commercial instrument, how and when last it was calibrated, and clarify the statement "The photolysis rate of any given species was calculated by normalizing to the cross section and quantum yields" as this doesn't sound right.

The instrument is a commercial Metcon UF CCD, which was calibrated by high power halogen lamp after the field campaign. The statement is rewritten as: "Photolysis rates were measured by a commercial spectradiometer for O3, NO2, HCHO, HONO and H2O2 (Metcon UF CCD), the instrument was calibrated by high power halogen lamp after the field campaign. The photolysis rate of other related species were scaled by the recommendation of the Jet Propulsion Laboratory (JPL) kinetic evaluation report (Burkholder et al., 2015). Before the campaign the was instrument  calibrated through comparison with a chemical actinomter in 2014 (Zou et al., 2016)."

pg 5 / lines 10-20 Numerous grammatical errors in this paragraph. Please correct.

The grammatical errors in this paragraph have been addressed

It now reads

"An Ionicon Analytik high sensitivity PTR-MS (Proton TRansfer Mass Spectrometer) as described by de Gouw and Warneke et al, (2007) provided supporting precursor VOC measurements. Detailed information about the PTR MS measurements can be found in Yuan et al 2012 and 2013. In brief, 28 masses are measured throughout the campaign at 1 Hz. Zero air, which was produced by ambient air passing through a platinum catalytic converter at 350 °C (Shimadzu Inc., Japan), was measured for 15 min every 2.5 hours to determine the background. Aromatic masses (m/z 79 for benzene, m/z 93 for toluene, m/z 105 for styrene, m/z 107 for C8 aromatics and m/z 121 for C9 aromatics), oxygenated masses (m/z 33 for methanol, m/z 45 for acetaldehyde, m/z 59 for acetone, m/z 71 for MVK+MACR and m/z 73 for MEK), isoprene (m/z 69) and acetonitrile (m/z 42) were calibrated by using EPA TO15 standard from Apel-Riemer Environmental Inc., USA. Formic acid (m/z 47), acetic acid (m/z 61), formaldehyde (m/z 31), and monoterpenes (m/z 81 and m/z 137) were calibrated by permeation tubes (VICI, USA)."

Please state the uncertainty of the PTR-MS data.

The uncertainties of most species are below 10%, which is detailed in the previous work (Liu, Y. 2015, ACP). This is now stated in the text.

line 38. N2 should have a subscript (twice).

N2 now is subscripted

pg 6 / lines 3-21 Numerous sub- and superscripts missing in this paragraph

Sub and superscripts are now properly utilised in the entire manuscript

line 25-26 " but is a commonly applied method within the CIMS community "
This is bad practice and done only by certain groups - most in the community strive to calibrate their instruments properly. Please remove this phrase.

This sentence has been removed as the data shown for uncalibrated species is now displayed in counts

pg 8 / line 5 "The high level of agreement". Be quantitative and state the level of agreement

The following sentence has been added

"The high level of agreement (R2 of 0.76) from low mixing ratio measurements and a species with a short lifetime from different inlets confirms the accuracy and reliability of the CIMS measurements for this campaign."

pg 8 / lines 8-15. A comparison of N2O5 mixing ratios would be more meaningful if the production rate of NO3 (i.e., NO2 & O3 concentration) and the steady state lifetime of N2O5 were provided for context.

The following text was added to the end of the paragraph directing the reader to a manuscript (Wang et al 2018) which evaluated the production rates and fate of N2O5 from the same campaign and data set

"Further analysis of N2O5 nighttime chemistry was performed by Wang et al (2018) who calculated an average steady state lifetime of 310 + 240 s and mean uptake coefficient of 0.034 ± 0.018."

pg 8 / lines 19-. Seems like the authors gave up on sub- and superscripts entirely at this point.

Sub and superscripts are now properly utilised in the entire manuscript

pg 11/ line 28-35. There are 15 reactions (numbered R1-R15) listed in the introduction on page 2 and 3. Some of the same reactions stated again but numbered differently here.

The equations for the steady state calculation are now labelled "ss" to reduce confusion to the reader on reaction numbers and imply that these reactions are relevant to the steady state calculation

pg 12 / line 7 " Steady state calculations reveal a sharp rise of chlorine atoms produced at sunrise peaking at 1.6x105 molecules cm3 around 7 am which then gradually decreases, contributing to Cl atom production until 2 pm (Figure 7a). "

The text gives a chlorine atom concentration, whereas Fig. 7a only gives the production rate. Consider showing the magnitude of the assumed sink (chlorine reactivity in 1/s, next to OH reactivity in 1/s) in the supplemental and adding a disclaimer, such as "If a Cl sink as shown in Fig. X and a steady state w.r.t. production and loss are assumed, ..."

The referee is correct that we assume a steady state approximation, as shown in equation 9. We have added more text to clarify the uncertainty in the steady state assumption concentration, see answer to previous question. However, we feel that the figure is more informative if production rate is shown rather than steady state, as this reflects the measurements that were taken during the campaign.

pg 15 / line 21 "enabling an average daytime peak mixing ratio of chlorine atoms of $1.6 \times 10^5$ molecules cm-3." Stating a number for actual Cl atom concentration is not justified here given the large uncertainty with the magnitude of the hydrocarbon sink for Cl.

The concentration stated of chlorine atoms has been removed

Supplemental / pg 1 / line 20 " it has been shown that using these emissions it is possible to estimate the Cl atom production" The statement, as written, suggests that such calculations give accurate results. This has not been shown. Please rephrase.

This has been rephrased to read

". Whilst the approach is a simplification of course, it has been shown that using these emissions it is possible to estimate the Cl atom production, albeit it with some significant error due to significant number of estimations made, in a Megacity environment and produces results that are comparable with the much more thorough modelling approach of the MCM. It also generates a metric, CH4 equivalent, which can be used as a comparative measurement from city to city. "

Reviewer 2

The manuscript by Le Breton et al. describes measurements of reactive chlorine species in the gas and particle phase in Beijing. They use this data to understand the sources of chlorine atoms and constrain the chlorine budget. The revised paper is greatly improved and, subject to addressing the comments below, can be considered for publication in ACP.

The authors have improved details regarding experiments, calibrations, and models, including much more information about relevant caveats and assumptions. This greatly improves the paper. Given the uncertainties the authors have described, their word choice throughout should reflect these uncertainties. There are two cases where word change is required. Firstly, the source of aerosol chloride does have some uncertainty, because no measurement of sea salt was made. Thus, the end of the first paragraph in Section 3.3.2 should reflect this. Instead of "we could deduce that", use "it is

likely that" or equivalent. Secondly, the second paragraph of Section 3.4 begins with an assumption about the accuracy of methods/assumptions, and is followed (page 11, lines 1-2) with "this data indicates". Again, terminology that incorporates the stated uncertainty (e.g. suggests) should be used.

We have considered the phrasing and terminology of the data throughout the manuscript and also specifically changed the two examples as follows

The phrase "it is likely that" has bene used to replace "we deduce that" and "this data suggests that" has replaced "this data indicates" as suggested.

The manuscript still contains several grammatical errors and should be carefully proofread (e.g. page 5, lines 13-16).

The grammatical errors within the manuscript have been addressed

In several places throughout the manuscript the incorrect terms "chloride atom" or "chloride radical" are used (e.g. page 2, line 30).

The term chloride radical is now used consistently within the text

Mixing ratios are still given as ppt, ppb, etc. In the response file, the authors indicate they have changed all to pptv, ppbv, etc. This still needs to be completed (including in figures).

All units are now in ppt and ppb

Figure 2: there is overlap between panels B, C, and D.

This overlap of panels has been altered

Figure 5: the axes could still be more clear here (e.g. the difference between the right and left axes of panel B are unclear; some of the axes also appear to be floating and it isn't clear to which panel they belong). The axes should not use [ClNO2] because this indicates a concentration and the graphs depict a mixing ratio.

Figure 5 has been amended as recommended

Figures 9-11: the axes use [ppt], which incorrectly implies concentration instead of mixing ratio.

The axis have now been changed to ion counts

---

## Author Response (AR3)

A few minor comments/suggestions prior to uploading the final version for copy-editing:

pg 5 / line 11 "Photolysis rates were measured by a commercial spectradiometer for O3, NO2, HCHO, HONO and H2O2. (Metcon UF CCD), the instrument was calibrated by high power halogen lamp after the field campaign."
Please subscript molecular formulae and insert an "a" between "by" and "high".

These have been subscripted

pg 5 / line 15 "Before the campaign the was instrument calibrated through comparison with a chemical actinomter in 2014 (Zou et al., 2016)."
Spelling: actinometer. The Zou et al. paper does not describe this comparison. Please give an indication what the comparison showed (e.g., the instruments agreed within +/-10%).
I am assuming this instrument integrates light over one hemisphere. How was the upwelling radiation accounted for?

The spelling has now been amended

The following text has been added

"Before the campaign the was instrument calibrated through comparison with a chemical actinometer utilised in 2014 (Zou et al., 2016), agreeing within 10%. The surface albedo is normally 0.05 at the ground near the site. Upwelling radiation is neglected as is represents an insignificant fraction of the downwelling values."

pg 11 / line 11 "Data" are plural, so it should be "these data suggest"

This has been changed

pg 20/ line 35 "nitryl chlorine" should be nitryl chloride

This has been changed